# A Graphop Analysis of Graph Neural Networks on Sparse Graphs: Generalization and Universal Approximation

Ofek Amran[1]   Tom Gilat[1]   Ron Levie[1]

## Abstract

Generalization and approximation capabilities of message passing graph neural networks (MPNNs) are often studied by defining a compact metric on a space of input graphs under which MPNNs are equicontinuous. Such analyses are of two varieties: 1) when the metric space includes graphs of unbounded sizes, the theory is only appropriate for dense graphs, and, 2) when studying sparse graphs, the metric space only includes graphs of uniformly bounded size. In this work, we present a unified approach, defining a compact metric on the space of graphs of all sizes, both sparse and dense, under which MPNNs are equicontinuous. This leads to more powerful universal approximation theorems and generalization bounds than previous works. The theory is based on, and extends, a recent approach to graph limit theory called graphop analysis.

## 1. Introduction

In recent years, Graph Neural Networks (GNNs) have emerged as a powerful class of models in machine learning, advancing the ability to represent and analyze relationship within data (Hamilton et al., 2017). Unlike traditional neural networks, that are designed for grid-structured data such as images (Lecun et al., 1998; Jmour et al., 2018), GNNs capture complex patterns and dependencies in graph-structured data (Kipf & Welling, 2017). Graphs are found in various domains, ranging from social networks and recommendation systems to molecular biology and transportation networks (Fan et al., 2019; Wang et al., 2022; Gao et al., 2023). *Message Passing Neural Networks* (MPNNs) are a special subclass of GNNs employing a message passing paradigm, where nodes aggregate and update information from their neighbors, allowing for the effective capture of local and global graph structures (Gilmer et al., 2017; Chami et al., 2022).

Despite their successes, several theoretical challenges remain in the study of MPNNs. Some of these challenges revolve around generalization, expressivity, and the universal approximation capability of GNNs. Generalization refers to the ability of GNNs to perform well on unseen data (Garg et al., 2020; Liao et al., 2021; Maskey et al., 2022; Morris et al., 2023; Rauchwerger et al., 2025; Vasileiou et al., 2025a;b). Expressivity, on the other hand, deals with the capability of GNNs to distinguish graph structures (Morris et al., 2019; Xu et al., 2019; Böker et al., 2024; Rauchwerger et al., 2025; Vasileiou et al., 2025a). Lastly, universal approximation theorems for GNNs concern whether GNNs can approximate any continuous function on graphs with respect to some compact topology, given enough layers and parameters (Chen et al., 2019; Keriven & Peyré, 2019; Azizian & Lelarge, 2021; Lu et al., 2021; Chen et al., 2022; Böker et al., 2024; Rauchwerger et al., 2025)

**A basic aparoach to generalization and universality.** In this paper we focus on the following approach for deriving universal approximation theorems and generalization bounds, based on endowing the space of graph $\mathcal{G}$ with a metric $d$. The two approaches can be summarized by the following two informal statements. Denote the set of MPNNs of interest by $\mathcal{M}$.

- **Stone–Weierstrass theorem.** Let $(\tilde{\mathcal{G}}, \tilde{d})$ be the completion of the metric space $(\mathcal{G}, d)$. If $(\tilde{\mathcal{G}}, \tilde{d})$ is compact, every $\theta \in \mathcal{M}$ is continuous, and $\mathcal{M}$ is an algebra that separates points, then $\mathcal{M}$ has universal approximation: every continuous function over $(\tilde{\mathcal{G}}, \tilde{d})$, restricted to $(\mathcal{G}, d)$, can be uniformly approximated by some $\theta \in \mathcal{M}$.

- **Covering number generalization bound.** If $(\mathcal{G}, d)$ has finite covering number and every $\theta \in \mathcal{M}$ is Lipschitz continuous (or more generally uniformly equicontinuous), then $\mathcal{M}$ admits a generalization bound: the error between the training and test set's risks is guaranteed to decay to zero as the size of the training set goes to infinity.

For more details on covering number generalization bounds

[1]Faculty of Mathematics, Technion – Israel Institute of Technology. Correspondence to: Ofek Amran <ofek.amran@campus.technion.ac.il>, Tom Gilat <tom.gilat@gmail.com>, Ron Levie <levieron@technion.ac.il>.

*Proceedings of the 43rd International Conference on Machine Learning*, Seoul, South Korea. PMLR 306, 2026. Copyright 2026 by the author(s).

see Appendix C.4 and for Stone-Weierstrass see Theorem C.4 or Theorem 7.32 in (Rudin, 1976).

**Limitations in previous results.** In order to base a universal approximation theorem on the Stone–Weierstrass theorem, one must endow the space of graphs with a compact pseudo-metric under which MPNNs are both continuous and capable of separating points. These requirements naturally pull in opposite directions. On the one hand, compactness forces all graphs to lie "close" to one another. However, if graphs are too nearby each other, then MPNNs must have discontinuities. Moreover, if in order to facilitate the continuity of MPNNs the metric is chosen too fine, MPNNs lose the capability to separate points. Similarly, the finite covering and Lipschitz continuity are competing requirements in covering number generalization bounds. This makes utilizing the above two theorems challenging for GNNs.

As a result of these challenges, existing analyses have the following limitations. In order to include graphs of all sizes in $\mathcal{G}$, generalization and approximation capabilities of MPNNs are commonly studied through graph limit theory[1]. To be able to process graphs of all sizes, current approaches that fall under this umbrella use normalized sum aggregation as the message passing mechanism, which limits the analysis to dense graphs (Levie, 2024; Böker et al., 2024; Rauchwerger et al., 2025; Rauchwerger & Levie, 2025). On the other hand, approaches that can account for sparse graphs, and use sum aggregation, must limit the space $\mathcal{G}$ to only include graphs up to some size (Chen et al., 2019; Azizian & Lelarge, 2021; Chen et al., 2022; Vasileiou et al., 2025a).

**Our contribution.** In this work, we present a unified approach, defining a compact metric on the space of all graphs, both sparse and dense, of all sizes, that fulfills the conditions of the above two theorems. This is done by replacing the graphon-based (dense) graph limit theory of previous approaches by a graph limit theory appropriate for sparse graphs. For this, we adopt and extend *graphop-analysis* (Backhausz & Szegedy, 2022), a graph limit theory appropriate for sparse graphs which unifies many previously proposed theories.

We extend graphop theory to include vertex attributes, and introduce a subclass of graphops called *bounded fiber operators (bofops)*. We show that bofops can describe a rich family of sparse connectivity patterns, and are the limit objects of both sparse and dense graphs under the standard metric used in graphop theory – the action metric. MPNNs are canonically extended to accept bofops as inputs, and we show that one can model both sum and normalized sum

aggregation naturally through the lens of bofop theory, as well as additional novel aggregations. Hence, in our setting, MPNNs can accepts both sparse and dense graphs of any size.

To describe the expressivity of MPNNs on bofops, we extend the 1-WL graph isomorphism test to bofops. We moreover adopt a pseudo-metric on the space of bofops, called DIDM-mover's distance (Rauchwerger et al., 2025), which is coarser than the action metric. The DIDM-mover's metric describes similarity between bofops through their computational trees underlying the 1-WL algorithm. While the end goal is to use the DIDM-mover's distance in the analysis, as this is the metric that describes the separation power of MPNNs, the properties of this metric are only revealed once first analyzing MPNNs with respect to the action metric.

We then prove that bofops satisfy all of the requirements of the above approaches to generalization and universal approximation: 1) the space of bofops is a compact metric space both under the action and the DIDM-mover's metrics (and hance bofops also admit a finite covering), 2) the space of sparse and dense graphs of all sizes is dense in the space of bofops (and hence bofops are the completion of the space of all graphs), 3) MPNNs are Lipschitz continuous with respect to the action metric and uniformly equicontinuous with respect to the DIDM-mover's pseudo-metric, and 4) MPNNs separate points with respect to the DIDM-mover's pseudo metric. This leads to powerful universal approximation and generalization theorems.

### 1.1. Related Work

Appendices C through E provide extensive background and related work on concepts relevant to this paper. In appendix C we review background on machine learning, MPNNs, and discuss generalization and universal approximation theorems. Appendix D covers background in graph limit theory, including classical limit objects for dense graphs such as graphons, as well as exciting analyses of GNNs based on graphon theory. We also give several approaches to sparse graph limit theories extended by graphops in Appendix O. Finally, appendix E is devoted to iterated degree measures.

## 2. Background

### 2.1. Basic Definitions

We begin with background and notation. For additional background on analysis we refer to Appendices A and B.

**Basic Notations.** The set of non-negative integers is denoted by $\mathbb{N}_0$, and $[n] = \{1, \ldots, n\}$. For a subset $E \subset \mathcal{X}$, $E^c$ is its complement. For a function $f : \Omega \to \mathbb{R}$, we write $f(x)$ or $f_x$ for the value of $f$ at a point $x \in \Omega$. We use $f(-)$ or $f_-$ to denote the function $x \mapsto f(x)$ or $f_x$, i.e., simply $f$.

---

[1]Graph limit theory is the study of the limit objects in the completion of graph metric spaces, i.e., it is the characterization of limits of graph Cauchy sequences. Such limit objects are typically interpreted as graphs with infinitely many vertices.

**Graphs.** A graph is a triple $G = (V, E, \mathbf{A})$ where $V = [n]$ is the node set, and $|V|$ denotes the umber of nodes. The edge set is given by $E \subset V \times V$, where $(i,j) \in E$ if and only if $i$ and $j$ are connected by an edge, and $\mathbf{A} \in \mathbb{R}^{n \times n}$ is the adjacency matrix of $G$. We restrict the analysis to undirected graphs, i.e., $\mathbf{A} = \mathbf{A}^\top$. For an unweighted graphs, the entries of $\mathbf{A}$ are given by $e_{i,j} = 1$ if $(i,j) \in E$ and $e_{i,j} = 0$ otherwise. An *attributed graph*, also called a *graph-signal*, is a graph $G$, where each $i \in V$ is equipped with a feature vector $\mathbf{f}(i) = \mathbf{f}_i \in \mathbb{R}^d$, and $d$ is called the feature dimension. For a node $i \in V$, let $\mathcal{N}(i) = \{j \in V \mid (i,j) \in E\}$ be the *neighborhood* of $i$.

**Analysis.** For a Borel measurable space $\Omega$, let $\mathscr{M}_{\leq 1}(\Omega)$ denote the space of Borel measures $\mu$ on $\Omega$ with total mass at most 1, i.e., $\|\mu\| := \mu(\Omega) \leq 1$. Let $\mathcal{P}(\Omega)$ denote the space of Borel probability measures on $\Omega$. For a subset $E \subset \Omega$, $\mathbb{1}_E$ denotes its indicator function. We say that $f$ is an element of $\mathcal{L}^p(\Omega)$, $1 \leq p < \infty$, if $\|f\|_p^p := \int_\Omega |f|^p \, d\mu$ is finite and $f \in \mathcal{L}^\infty(\Omega)$ if $\|f\|_\infty := \inf\{C > 0 : |f(x)| < C \text{ for a.e. } x \in \Omega\}$ is finite. Here, *a.e.* stands for *almost every*. Given a Borel measure space $(\Omega_1, \Sigma_1, \mu)$ and a measurable function $f : \Omega_1 \to \Omega_2$, the *pushforward* $f_* \mu$ of $\mu$ via $f$ is the Borel measure on $\Omega_2$ that satisfies $f_* \mu(E) = \mu(f^{-1}(E))$ for every $E \subset \Omega_2$.

## 2.2. MPNNs on Graphs

Message passing neural networks (MPNNs) (Gilmer et al., 2017), are a class of neural networks designed for graph structured data. MPNNs takes graphs, or attributed graphs as an input, and iteratively update node features through message passing.

We define the parameters of an MPNN as follows. Let $L \in \mathbb{N}_0$ and let $p, d_0, \ldots, d_L, d \in \mathbb{N}_0$. An *L-layer MPNN model* is a collection $\varphi = (\varphi^{(l)})_{l=0}^L$ of Lipschitz continuous functions, where

$$\varphi^{(0)} : \mathbb{R}^d \to \mathbb{R}^{d_0}, \qquad \varphi^{(l)} : \mathbb{R}^{2d_{l-1}} \to \mathbb{R}^{d_l}, \quad 1 \leq l \leq L,$$

and where the functions $\varphi^{(l)}$ are called *update functions*. Given a Lipschitz continuous function $\psi : \mathbb{R}^{d_L} \to \mathbb{R}^p$, the tuple $(\varphi, \psi)$ is called an *MPNN model with readout*, where $\psi$ is referred to as the readout function. We call $L$ the *depth* of the MPNN, $d$ the *input feature dimension*, $d_0, \ldots, d_L$ the *hidden feature dimensions*, and $p$ the *output feature dimension*. We denote the space of all MPNN models with Lipschitz constants of the update functions and $\psi$ bounded by $D$ by $\mathcal{MP}_D(d, d_0, \ldots, d_L, p)$.

Applying an MPNN model as a function on graph-signals is defined as follows. Let $(\varphi, \psi)$ be an L-layer MPNN model with readout, and $(G, \mathbf{A}, \mathbf{f})$ be a weighted graph-signal with adjacency matrix $\mathbf{A}$, where $\mathbf{f} : V(G) \mapsto \mathbb{R}^d$. The application of the MPNN on $(G, \mathbf{f})$ is defined as follows:

initialize $\mathfrak{g}_-^{(0)} := \varphi^{(0)}(\mathbf{f}(-))$ and compute the *hidden node representations* $\mathfrak{g}_-^{(l)} : V(G) \to \mathbb{R}^{d_l}$ at layer $l$, with $1 \leq l \leq L$, and the *graph-level output* $\mathfrak{G} \in \mathbb{R}^p$, by

$$\mathfrak{g}_v^{(l)} := \varphi^{(l)}\left(\mathfrak{g}_v^{(l-1)}, (\mathbf{A}\mathfrak{g}_-^{(l-1)})_v\right),$$

$$\mathfrak{G} := \psi\left(\sum_{v \in V(G)} \mathfrak{g}_v^{(L)}/|V|\right).$$

The expression $(\mathbf{A}\mathfrak{g}_-^{(l-1)})_v$, i.e., coordinate $v$ of $\mathbf{A}\mathfrak{g}_-^{(l-1)}$, is called the *aggregated feature* of the node $v \in V(G)$.

Given an unweighted graph, one can normalize its adjacency matrix in various ways. For example, when $\mathbf{A}_{i,j} = e_{i,j}$, the aggregation is called *sum aggregation*. When $\mathbf{A}_{i,j} = e_{i,j}/|V|$, the aggregation is called *normalized sum aggregation*, and takes the form $\sum_{u \in \mathcal{N}(v)} \mathfrak{g}_u^{(l-1)}/|V|$ for unweighted graphs. Here, we note that MPNN with normalized sum aggregation are only appropriate for dense graphs. Indeed, if the graph is sparse, then $|V| \gg |\mathcal{N}(v)|$, and the aggregation will always give a close-to-zero outcome. Hence, such an MPNN will always give roughly the same output for all sparse graphs. Moreover, sum aggregation is only asymptotically appropriate for sparse graphs, otherwise, the aggregation diverges to infinity as the size of the graph increases.

One can also consider novel aggregations. For example, the following matrix can be seen as symmetric average aggregation: $\mathbf{D}^{-1/2}\mathbf{A}\mathbf{D}^{-1/2}$, where $\mathbf{D}$ is the diagonal *degree matrix*, with diagonal elements $d_i = \sum_j a_{i,j}$. Symmetric average aggregation can be appropriate both for sparse and dense graphs.

The architecture defined above corresponds to the Graph Isomorphism Network (GIN) (Xu et al., 2019). While there are more general formulations of MPNNs, which use message functions in addition to update functions, we show in Appendix F that such MPNNs can be reduced to our model.

## 2.3. DIDM-Mover's Distance

It is well-known that MPNNs have the same separation power as the 1-WL test (Morris et al., 2019; Xu et al., 2019). The 1-WL algorithm is an iterative color refinement algorithm, where at each step each node is given a distinct color with respect to the distribution of colors of its neighbors in the previous step. At the last step, the histogram of colors over all nodes is extracted. These colors can be used to define a metric called DIDM-Mover's distance.

**IDMs and DIDMs.** By the above description, colors at step $j$ can be mathematically defined as histograms, or measures, of the colors of step $j - 1$, which are themselves measures of colors from step $j - 2$, and so on. This leads to the notion of *iterated degree measures (IDMs)* for the node

colors in the 1-WL algorithm, and *distributions of iterated degree measures (DIDMs)* for the histogram of the colors of all nodes. Formally, the spaces of IDMs and DIDMs are defined as follows (Rauchwerger et al., 2025).

**Definition 2.1** (Spaces of IDMs and DIDMs). The *space of IDMs* $\mathcal{H}^L$ of order $L \in \mathbb{N}$ is defined inductively as follows. First, $\mathcal{H}^0 = [-1, 1]^d$, for some $d \in \mathbb{N}_0$. Then, for every $L > 0$, define $\mathcal{H}^L = \prod_{j=0}^L \mathcal{M}^j$ and $\mathcal{M}^{L+1} = \mathcal{M}_{\leq 1}(\mathcal{H}^L)$ with the product and weak* topologies respectively (See Appendices A.1.2 and B.1). Define the *space of DIDMs* of order $L$ by $\mathcal{P}(\mathcal{H}^L)$.

Given a graph-signal, its DIDM can be computed by a variant of the 1-WL algorithm. We present an extension of this algorithm in Subsection 5.1. It is important to note that the space of all DIDMs is larger than the space of DIDMs obtained by the 1-WL algorithm on graphs or graphons. In a sense, most DIDMs are formal objects that do not describe the structure of any graph (see Appendix E and Example E.7 for more details).

**DIDM-Mover's Distance.** Since DIDMs are distributions, one can define a Wasserstein distance between them. Moreover, since DIDM's are recursively measures of measures, the Wasserstein distance is also defined recursively.

**Definition 2.2** (The Unbalanced Wasserstein Distance). Let $(\mathcal{X}, d)$ be a Polish space. The *Unbalanced Wasserstein Distance* between two measures $\mu, \nu \in \mathcal{M}_{\leq 1}(\mathcal{X})$ is

$$\mathbf{OT}_d(\mu, \nu) = \inf_{\gamma \in (\mu, \nu)} \left( \int_{\mathcal{X} \times \mathcal{X}} d(x, y) d\gamma \right) + |\|\mu\| - \|\nu\||$$

where $\Gamma(\mu, \nu)$ is the set of all couplings of $\mu$ and $\nu$, and $\|\mu\| = \mu(\mathcal{X})$. See Appendix A.2.5 for more details.

The DIDM Mover's Distance is then defined as follows. First, define the *IDM distance* of order-0 on $\mathcal{H}^0 = [-1, 1]^d$ by $d^0_{\mathrm{IDM}}(x, y) := \|x - y\|_2$ for $x, y \in [-1, 1]^d$, and denote by $\mathbf{OT}_{d^0_{\mathrm{IDM}}}$ Wasserstein distance on $\mathcal{M}^1 = \mathcal{M}_{\leq 1}(\mathcal{H}^0)$. The distance $d^L_{\mathrm{IDM}}$ is defined recursively on $\mathcal{H}^L$ where the distance on $\mathcal{M}^j$ for $0 < j < L$ is $\mathbf{OT}_{d^{j-1}_{\mathrm{IDM}}}$. Explicitly, $d^L_{\mathrm{IDM}}(\mu, \nu) = \|\mu_0 - \nu_0\|$ if $L = 0$ and $d^L_{\mathrm{IDM}}(\mu, \nu) = \|\mu_0 - \nu_0\|_2 + \sum_{j=1}^L \mathbf{OT}_{d^{j-1}_{\mathrm{IDM}}}(\mu_j, \nu_j)$ if $0 < L < \infty$, where $\mu = (\mu_j)_{j=0}^L \in \mathcal{H}^L$ and $\nu = (\nu_k)_{k=0}^L \in \mathcal{H}^L$.

Finally, given two DIDMs $\Gamma_1, \Gamma_2 \in \mathcal{P}(\mathcal{H}^L)$ their DIDM-Mover's distance is defined by $\mathbf{OT}_{d^L_{\mathrm{IDM}}}(\Gamma_1, \Gamma_2)$.

In (Böker et al., 2024; Rauchwerger et al., 2025), it was shown that the space of all DIDMs is compact with respect to the DIDM-mover's distance. Moreover, it was shown that the space of DIDMs computed via 1-WL on graphons (or graphon-signals) is also compact.

## 2.4. MPNNs on General DIDMs

There is a well-defined notion of applying an MPNN directly to a DIDM, even when the DIDM does not arise from any underlying graph-signal or graphon-signal via 1-WL (Definition E.8). A key property of this construction is that MPNNs and the 1-WL algorithm satisfy a commutation property. Informally, given a graph-signal, applying the 1-WL algorithm and then applying an MPNN on the resulting DIDM, yields the same output as applying the MPNN directly to the graph-signal and then inducing a DIDM via 1-WL. See (Böker et al., 2024; Rauchwerger et al., 2025) and Appendix E.6 for details.

In addition, it was shown that MPNNs with normalized sum aggregation are Lipschitz continuous with respect to DIDM-mover's distance. However, the fact that such MPNNs are only appropriate for dense graphs limits the applicability of this result. In this paper, we also show the continuity of MPNNs on sparse graphs.

# 3. Graphop and Bofop Analysis

Graphop theory is a recent graph limit theory that extends many well known limit theories of sparse graphs. See Appendices D and O for a survey on graph limit theories. The idea is to model graphs as their adjacency matrices, and extend those to general operators over $\mathcal{L}^p(\Omega)$ spaces of functions over a probability spaces $\Omega$. Here, $\Omega$ is interpreted as the space of vertices. As opposed to graphons, which are kernel operators over $\mathcal{L}^\infty[0, 1]$ based on kernels $K : [0, 1] \times [0, 1] \to [0, 1]$ (see Appendix D.1), graphops are general operators that need not be defined with respect to a kernel. This allows graphops to encode sparse edge connectivity patterns in $\Omega \times \Omega$.

Different graphops can be defined over different vertex sets $\Omega$ and $\Omega'$. Hence, in order to facilitate a definition of a metric between graphops, all graphops over all vertex sets are represented in a common space called the *space of profiles*, as discussed in this section. To make graphops compatible with MPNNs, we consider a subclass of graphop that we call *bofops*. Since the most common data-type in graph machine learning are graphs with node features, we also pair a *signal* to each bofop, i.e., a mapping from vertices to features, and call the resulting object a *bofop-signal*.

## 3.1. Basic Definitions

The above discussion leads to the following definition from (Backhausz & Szegedy, 2022). Let $(\Omega, \mu)$ be a standard Borel probability space. For $(p, q) \in [1, \infty]^2$ a P-operator is a bounded linear operator $A : \mathcal{L}^p(\Omega) \to \mathcal{L}^q(\Omega)$, i.e., such that

$$\|A\|_{p \to q} := \sup_{0 \neq f \in \mathcal{L}^p(\Omega)} \|Af\|_q / \|f\|_p < \infty.$$

Denote by $\mathcal{B}_{p,q}(\Omega)$ the space of all P-operators with finite $(p,q)$ norm over $\Omega$, and for $r > 0$, denote by $\mathcal{B}_{p,q}^r(\Omega)$ the space of all P-operators with $(p,q)$ norms bounded by $r$. Note that $\mathcal{B}_{p,q}(\Omega) \subset \mathcal{B}_{\infty,1}(\Omega)$ for every $(p,q) \in [1,\infty]^2$.

We now restrict the notion of P-operators to include properties that make them more graph-like objects, called *graphops* (Backhausz & Szegedy, 2022). For two measurable functions $u, v \in \mathcal{L}^\infty(\Omega)$ and a P-operator $A \in \mathcal{B}_{p,q}(\Omega)$ we define the bilinear form $(v,u)_A$ by

$$(v,u)_A := \int_\Omega (Av) \cdot u \, d\mu.$$

**Definition 3.1** (Graphops). Let $A \in \mathcal{B}_{\infty,1}(\Omega)$.

- The operator $A$ is *self adjoint* if for every $v, u \in \mathcal{L}^\infty(\Omega)$, $(v,u)_A = (u,v)_A$.
- The operator $A$ is *positively preserving* if for every $v \in \mathcal{L}^\infty(\Omega)$, such that $v(x) \geq 0$ for a.e. $x \in \Omega$, we have that $Av(x) \geq 0$ for a.e. $x \in \Omega$.
- *Graphop* is a self adjoint and positively preserving P-operator.

Denote by $\mathscr{G}_{p,q}(\Omega)$ the space of all graphops with finite $(p,q)$ norm over $\Omega$, and for $r > 0$, denote by $\mathscr{G}_{p,q}^r(\Omega)$ the space of all grpahops with $(p,q)$ norms uniformly bounded by $r$. We call the space $\mathscr{G}_{1,1}(\Omega)$ the space of *bounded fiber operators (bofops)*. A measurable function $f : \Omega \to [-1,1]^d$ is called a *signal*, where $d \in \mathbb{N}_0$ is called the *feature dimension*. A pair $(A, f)$ of a bofop and a signal is called a *bofop-signal*. Denote by $\mathcal{BF}_d(\Omega)$ the space of *bofop-signals*, and by $\mathcal{BF}_d^r(\Omega)$ the bofop-signals with bofop bounded by $r$ in 1-norm.

*Remark* 3.2. Note that $\mathcal{G}_{1,1}(\Omega) = \mathcal{G}_{\infty,\infty}(\Omega)$, and the 1 and $\infty$ norms of bofops are equal (see Appendix H.1.4).

We focus in our analysis on bofops as they have a special connectivity structure which makes them highly compatible with MPNNs, as we discuss in the next subsection. There, the name *bounded fiber operator* will become clear.

Various graph like objects, both sparse and dense, can be represented as bofops, e.g., graphs and graphons. In addition, in Appendix H.1.5 we give as an example of a bofop the *equator graphop*, or *spherical graphop* presented in (Backhausz & Szegedy, 2022).

### 3.2. The Explicit Adjacency Structure of Bofops

Instead of describing the connectivity pattern underlying a bofop by a subset of $\Omega \times \Omega$, and describing node neighborhoods by subsets of $\Omega$, in graphop theory the conenctivities are characterized via measures over these spaces. Namely, there is a measure that quantifies, given any set $E \subset \Omega \times \Omega$,

what the percentage of edges within $E$ is, and there is a family of measures similarly describing neighborhoods. Such measures can in general represent sparse patterns.

The measure $\nu$ associated with the edge connectivity structure in $\Omega \times \Omega$ is defined by

$$\nu(S \times T) = (\mathbb{1}_S, \mathbb{1}_T)_A = \int_T A\mathbb{1}_S(\omega) \, d\mu(\omega), \quad (1)$$

for every two measurable sets $S, T \subset \Omega$. This measure is uniquely extended to general measurable sets in $\Omega \times \Omega$. It was shown in Theorem 6.3 in (Backhausz & Szegedy, 2022) that for every two signal $f, g$ in $\mathcal{L}_\infty(\Omega)$, we have

$$\int_\Omega (Av) \cdot u \, d\mu = \int_{\Omega^2} f(x)g(y) d\nu(x,y), \quad (2)$$

and hence $\nu$ indeed describes the connectivity underlying $A$. Neighborhood measures are defined as follows.

**Theorem 3.3.** *Let $A$ be a bofop over the Borel probability space $(\Omega, \mu)$. Then, there exists a unique measurable family of measures $(\nu_x)_{x\in\Omega}$, called* fibers*, where each $\nu_x$ is a Borel measure over $\Omega$, that satisfy the following properties.*

1. *For any signal $f$, $(Af)(x) = \int_\Omega f \, d\nu_x$.*
2. *$\|A\|_{\infty\to\infty} = \|A\|_{1\to 1} = \operatorname{ess\,sup}_{x\in\Omega} \nu_x(\Omega) < \infty$.*

*Conversely, for any family of fibers that satisfy $\operatorname{ess\,sup}_{x\in\Omega} \nu_x(\Omega) < \infty$, if the operator $A$ defined by 1 is symmetric, then it is a bofop with norm given by 2.*

Note that property 2 justifies the term *bounded fiber* operators. Note as well that each fiber $\nu_x$ represents the neighborhood of node $x \in \Omega$, and $\int_\Omega f \, d\nu_x$ is interpreted as the aggregation of $f$ about $x$ by $A$. Since there is no restriction on the fibers other than uniform boundedness, $\nu_x$ can be supported on a null-set with respect to the base measure $\mu$ of $\Omega$. In this sense, $\nu_x$ can describe a sparse neighborhood. See Section 6 of (Backhausz & Szegedy, 2022) and Appendix H.1.4 for more details.

### 3.3. Profiles as Representations of Bofop-Signals

One goal in graphop theory is to treat all graphops over all probability spaces in a unified manner. We represent different graphops, defined over different probability spaces, as points in a the same space as follows: represent each graphop by the histogram of its output values on input functions. The fact that all histograms live in the same space, regardless of $\Omega$, allows comparing any two graphops. We formalize this idea, and extend the original setting given in (Backhausz & Szegedy, 2022) to also include the signal.

**Definition 3.4** ($k$-Profile). Let $(\Omega, \mu)$ be a standard Borel probability space and $k \geq 0$ be an integer.

- For measurable functions $v_1, \ldots, v_k : \Omega \to [-1, 1]$, denote by $\mathrm{D}(v_1, \ldots, v_k)$ the joint distribution of the random variables $v_1, \ldots, v_k$, i.e., the measure in $\mathbb{R}^k$ that is the pushforward of $\mu$ under the map $x \mapsto (v_1(x), \ldots, v_k(x))$ (see Definition A.17 for pushforward).

- Let $(A, f) \in \mathcal{BF}_d^r(\Omega)$ be a bofop-signal. The *P-distribution* of $(A, f)$ with respect to $v_1, \ldots, v_k : \Omega \to [-1, 1]$ is defined as $\mathrm{D}_{(A,f)}(v_1, \ldots, v_k) := \mathrm{D}(v_1, \ldots, v_k, Av_1, \ldots, Av_k, f)$. We call $k$ the *order* of the P-distribution $\mathrm{D}_{(A,f)}(v_1, \ldots, v_k)$.

- The *k-profile* $\mathrm{S}_k(A, f)$ of $(A, f)$ is the set of all P-distributions of order $k$, i.e., $\mathrm{S}_k(A, f) = \{\mathrm{D}_{(A,f)}(v_1, \ldots, v_k) \mid v_1, \ldots, v_k : \Omega \to [-1, 1] \text{ measurable}\}$. By convention, for $k = 0$ the tuple $(v_i)_{i=1}^k$ is empty, so $\mathrm{S}_0(A, f) = \{\mathrm{D}(f)\}$.

Note that if the feature dimension is 0, the signal part is omitted, and we recover the original setting of (Backhausz & Szegedy, 2022). See Appendix H.2 for further discussion.

### 3.4. Action Convergence and the Action Metric

Backhausz and Szegedy (Backhausz & Szegedy, 2022) defined a metric between graphops based on their profile representations. Since profiles are sets of measures, a metric between them is a metric between sets. For this, we consider the Hausdorff metric (see Appendix B.3 for details).

**Definition 3.5.** Let $(\mathcal{X}, d)$ be a metric space and $A, B \subset \mathcal{X}$ be two non-empty subsets of $\mathcal{X}$. The *Hausdorff distance* $d_H$ between $A$ and $B$ is defined by

$$d_H(A, B) := \max \left\{ \sup_{a \in A} \inf_{b \in B} d(a, b) \, , \, \sup_{b \in B} \inf_{a \in A} d(a, b) \right\}.$$

In our case, $\mathcal{X} = \mathcal{P}([-1, 1]^k \times [-r, r]^k \times [-1, 1]^d)$ is the space of probability measures. Originally, in (Backhausz & Szegedy, 2022), the base metric $d$ was taken to be the Levy-Prokhorov metric (Definition B.1), which metrize the weak topology of $\mathcal{P}([-1, 1]^k \times [-r, r]^k \times [-1, 1]^d)$. Instead, we consider a different metric which metrizes the same topology, namely, the Wasserstein distance (see Appendix G for the metrization property). This leads to the following definition.

**Definition 3.6** (Action Metric). The *action metric* between two bofop-signals $(A, f), (B, g)$ is defined as

$$d_M((A, f), (B, g)) := \sum_{k=0}^{\infty} 2^{-k} d_H \left( \mathrm{S}_k(A, f), \mathrm{S}_k(B, g) \right),$$

where $d_H$ is defined with respect to the Wasserstein distance.

In Appendix H.3, we prove that the space $(\mathcal{BF}_d^r(\Omega), d_M)$ is compact. We also establish several topological properties of profiles and of spaces of graphops beyond the class of bofops.

## 4. MPNNs on Bofop-Signals

Since bofops are natural extensions of graphs, there is a canonical extension of MPNNs to bofops, as defined next.

### 4.1. Basic Definition

Let $(\varphi, \psi)$ be an MPNN model with readout, and $(A, f)$ a bofop-signal, where $f : \Omega \to [-1, 1]^d$. The application of the MPNN on $(A, f)$ is defined as follows. Initialize $\mathfrak{h}_-^{(0)} := \varphi^{(0)}(f(-))$ and compute the hidden node representations $\mathfrak{h}_-^{(l)} : \Omega \to [-1, 1]^{d_l}$ at layer $l$, with $1 \le l \le L$, and the bofop-level output $\mathfrak{H}(A, f) \in \mathbb{R}^p$, by

$$\mathfrak{h}_x^{(l)} := \varphi^{(l)}(\mathfrak{h}_x^{(l-1)}, A\mathfrak{h}_-^{(l-1)}(x)), \tag{3}$$

and

$$\mathfrak{H}(A, f) := \psi \left( \int_\Omega \mathfrak{h}_x^{(L)} \, d\mu(x) \right).$$

The expression $A\mathfrak{h}_-^{(l-1)}$ is called the *aggregated signal*. We sometimes write $\mathfrak{h}_-^{(l)}(\varphi)$ and $\mathfrak{H}^{(\varphi, \psi)}(A, f)$ to make the dependence on the MPNN model explicit in the notation.

**Aggrgation Schemes Through The Lens of Bofops.** Given a graph $G$ with an adjacency matrix $\mathbf{A} \in [0, 1]^2$, we can modify $\mathbf{A}$ in various ways to model different types of aggregations, as discussed in Subsection 2.2. Cauchy sequences of adjacency matrices $(\mathbf{A}^{(j)})_j$ (normalized or not), with respect to the action metric, always converge to limit bofops as long as they have uniformly bounded 1 (or equivalently $\infty$) induced norms. According to the normalization of the adjacency matrices, the limit bofop can be interpret as sum, normalized sum, or symmetric average aggregation. Hence, bofops extend aggregations of both dense and sparse graphs. The type of aggregation underlying the bofop is implicit, and inherent in the bofop itself.

### 4.2. Equivalence of MPNNs on Profiles and Bofops

Note that the action metric is defined in terms of profiles. Moreover, different bofop-signals can have the same profile. Hence, in order to analyze the continuity of MPNNs with respect to the action metric, we must first show that the output of any MPNN on a bofop-signal only depends on the profile corresponding to the bofop-signal. For this, we show how to directly formulate MPNNs on profiles, without referring to a specific inducing bofop-signal.

The key observation is that profiles contain the information required for aggregation and update. To implement a mes-

sage passing step, we inspect only those P-distributions in the profile that are supported on the diagonal, i.e., the set $T_d \subset \mathbb{R}^{2k+d}$ in which the $k - d + 1$ to $k$ coordinates are equal to the last $d$ coordinates (Definition I.6). In Lemma I.7 we show that such P-distributions are of the form

$$D(v_1, \ldots, v_{k-d}, f_1, \ldots, f_d, \tag{4}$$
$$Av_1, \ldots, Av_{k-d}, Af_1, \ldots, Af_d, f_1, \ldots, f_d). \tag{*}$$

This form encodes the current signal together with the aggregated signal. *Diagonal marginalization* (Definition I.6) is precisely the operation in which we first restrict the profile to P-distributions of the form (*), and then marginalize (Definition A.16) the duplicate coordinates of $f_1, \ldots, f_d$, to obtain the *aggregated profile* $\mathrm{DM}_d(\mathrm{S}_k)$.

A message passing layer on a profile takes the form $\varphi^{(l)} \tilde{*} \mathrm{DM}_d$ where $\tilde{*}$ is an operation called *pushforward on the signal* (Definition I.2). This implements the update function directly on the resulting P-distributions by simply pushing forward (Definition A.17) the last coordinate through the update function.

This is shown to be the canonical definition in the following sense: starting with a bofop-signal $(A, f)$, if we induce its profile $\mathrm{S}_k(A, f)$ and then apply a message passing layer $\varphi^{(l)} \tilde{*} \mathrm{DM}_d$, we get exactly the same result as applying the message passing layer on the bofop-signal $(A, f)$ via (3) and then inducing a profile. For the formal construction and an in-depth description of the implementation of MPNNs on profiles, see Appendix I.3.

### 4.3. Lipschitz Continuity of MPNNs With Respect to the Action Metric

The above formulation of MPNNs directly on profiles allows proving their Lipschitz continuity.

**Theorem 4.1.** *Let $(A_1, f_1) \in \mathcal{BF}_d^r(\Omega_1)$ and $(A_2, f_2) \in \mathcal{BF}_d^r(\Omega_2)$ be bofop-signals over the Borel probability spaces $(\Omega_1, \mu_1)$ and $(\Omega_2, \mu_2)$ respectively. Let $(\varphi, \psi) \in \mathcal{MP}_D(d, d_0, \ldots, d_L, p)$. Then, there exists a constant $C_{D,r}$ that depends on the number of layers $L$, $D$ and $r$, such that*

$$d_M\left((A_1, \mathfrak{h}_1^{(L)}), (A_2, \mathfrak{h}_2^{(L)})\right) \leq$$
$$C_{D,r} \cdot d_M\left((A_1, f_1), (A_2, f_2)\right).$$

*Moreover, there exists a constant $C'_{D,r}$ that depends on $L$, $D$ and $r$ such that*

$$\|\mathfrak{H}(A_1, f_1) - \mathfrak{H}(A_2, f_2)\|_2 \leq$$
$$C'_{D,r} \cdot d_M\left((A_1, f_1), (A_2, f_2)\right).$$

The proof of Theorem 4.1 is given in Appendix J, where we establish Hölder continuity in a more general graphop-signal setting.

## 5. MPNNs and DIDM-Mover's Distance

In the previous section we have shown that the space of bofop-signals is compact with respect to the action metric, and MPNNs are Lipschitz continuous. However, the action metric is too fine, and there are pairs of bofop-signals that have positive distance but any MPNN attains the same value on both. Hence, the action metric cannot serve as the basis of the Stone-Weierstrass theorem, as MPNNs do not separate points. Still, our analysis of the action metric is an essential step in the proof of compactness of another metric, the DIDM-mover's distance of bofop-signals. The latter metric is shown to match the separation power of MPNNs, while MPNNs being uniformly equicontinuous with respect to it.

### 5.1. 1-WL Algorithm on Bofop-Signals

Recall the definitions of IDMs and DIDMs from Section 2.3. We define the bofop-signal 1-WL algorithm as follows.

**Definition 5.1** (Bofop-IDMs and Bofop-DIDMs). Let $(\Omega, \mu)$ be a standard Borel probability space and $(A, f) \in \mathcal{BF}_d^r(\Omega)$ be a bofop-signal, where $r > 0$. We define the map $\gamma_{(A,f),0} : \Omega \to \mathcal{H}^0$ to be the map $\gamma_{(A,f),0}(x) := f(x)$ for every $x \in \Omega$. Inductively $\gamma_{(A,f),L+1} : \Omega \to \mathcal{H}^{L+1}$ is defined by

1. $\gamma_{(A,f),L+1}(x)(j) = \gamma_{(A,f),L}(x)(j)$ for every $j \leq L$.
2. $\gamma_{(A,f),L+1}(x)(L+1)(E) = \left(A\mathbb{1}_{\gamma_{(A,f),L}^{-1}(E)}\right)(x)$, where $E \subset \mathcal{H}^L$ is a Borel set.

Finally, for every $L \geq 0$, let $\Gamma_{(A,f),L} = \gamma_{(A,f),L_*}\mu$, i.e. the pushforward of $\mu$ via $\gamma_{(A,f),L}$. We call $\gamma_{(A,f),L}$ a *bofop-IDM* of order $L$, and $\Gamma_{(A,f),L}$ a *bofop-DIDM* of order $L$. We denote the space of bofop-DIDMs of order $L$ by $\Gamma_L(\mathcal{BF}_d(\Omega))$.

Note that although we originally defined $\mathcal{M}^{L+1} = \mathscr{M}_{\leq 1}(\mathcal{H}^L)$ in Definition 2.1, we generalize $\mathcal{M}^{L+1}$ to be the space of Borel measures with total mass at most $r$ over $\mathcal{H}^L$. This generalization does not affect our analysis. See Appendices E and K for further discussion.

Similarly to the commutation property between the 1-WL algorithm and MPNN on graph-signals and graphon-signals discussed in Section 2.4, the same property holds for bofop-signals. See Appendix K, and in particular Lemma K.3 and Lemma K.4 for full formulation.

Now, we define the *bofop-DIDM mover's distance* $\delta_{\mathrm{DIDM}}^L\left((A_1, f_1), (A_2, f_2)\right)$ between two bofop-signals as the distance between their DIDMs computed via 1-WL.

### 5.2. The Action Metric is Finer Than DIDM-Mover's

In order to prove a universal approximation theorem, one must consider a metric on which the space of input objects

is compact, and for which MPNNs are continuous and separate points. As discussed above, the action metric $d_M$ is too fine, since distinct bofop-signals may yield the same output for every MPNN model. Instead we use the DIDM-mover's distance. Hence, we must prove the compactness of the space of bofop-DIDMs with respect to DIDM-mover's distance. This is done by 1) using the compactness of the space $(\mathcal{BF}_d^r, d_M)$, and 2) showing that the action metric is finer than the DIDM-mover's metric. Since we already know that 1 is true, we are left with showing 2 next.

**Theorem 5.2.** *Let* $(A_i, f_i)_{i \in \mathbb{N}} \in \mathcal{BF}_d^r(\Omega_i)$ *be a sequence of bofop-signals over a sequence of standard Borel probability spaces* $(\Omega_i, \mu_i)$, *and* $(A, f) \in \mathcal{BF}_d^r(\Omega)$. *Then* $\lim_{n \to \infty} d_M\left((A_n, f_n), (A, f)\right) = 0$, *implies* $\lim_{n \to \infty} \delta_{DIDM}^L\left((A_n, f_n), (A, f)\right) = 0$.

The proof of Theorem 5.2 is given in Appendix M.

Corollary 5.3 now follows immediately from Theorem 5.2, and its proof is given in Appendix M.

**Corollary 5.3.** *For every* $L \in \mathbb{N}_0$ *and* $r > 0$, *the space of bofop-DIDMs* $\Gamma_L(\mathcal{BF}_d^r) \subsetneq \mathcal{P}(\mathcal{H}^L)$ *is compact.*

### 5.3. Separation Power of MPNNs on Bofop-Signals

The separation power of MPNNs on DIDMs has been established in several works, especially Theorem 14 from (Rauchwerger et al., 2025). See also Appendix C.7. Since DIDMs arising from bofop-signals are a subclass of DIDMs, the same separation result applies in this setting as well. Namely, for any two bofop-signals of positive DIDM-mover's distance, there is an MPNN that attains a different value on each.

### 5.4. Dense/Sparse Graphs are Dense in Bofop-DIDMs

We next show that bofop-DIDMs is the completion of the space of finite graph-signals, sparse or dense, under the DIDM-Mover's distance. This makes bofop-DIDMs the natural object of study when analyzing MPNNs on general graph-signals, as MPNNs are continuous with respect to DIDM-Mover's distance. Moreover, this density result shows that with respect to $\delta_{\text{DIDM}}^L$, every bofop-signal can be viewed as a limit object of finite graph-signals.

The idea is to approximate any bofop-signal by a sequence of weighted graph-signals obtained from increasingly fine equipartitions of $(\Omega, \mu)$. Namely, given a bofop-signal $(A, f) \in \mathcal{BF}_d^r(\Omega)$, we partition $\Omega$ into $n$ sets of equal measure. Each set in the equipartition is interpreted as one node of a finite graph. The node features are obtained by averaging the signal $f$ over the corresponding sets. The bofop is discretized by projecting $A$ upon the partition. As the sequence of equipartitions becomes finer, these projected bofop-signals converge to the original bofop-signal in the DIDM-Mover's distance. Therefore, we get that graph-

signals are dense in the space of bofop-signals with respect to $\delta_{\text{DIDM}}^L$. Note that since we consider bofops with uniformly bounded $\infty \to \infty$ norms (or equivalently $1 \to 1$), the corresponding graphs have the same uniform bound. Since graphs with uniform $\infty \to \infty$ norms can be either dense or sparse, we conclude that the completion of the space of sparse and dense graphs with the above uniform bound is dense in the space of bofop-signals. The full construction and proofs are given in Appendix L.

### 5.5. Hierarchy of DIDM Spaces

We have proven the compactness of the space of bofop-DIDMs $\Gamma_L(\mathcal{BF}_d^r)$. Moreover, DIDMs obtained via 1-WL on graphon-signals, which we refer to as *graphon-DIDMs*, are already known to form a compact subspace of the space of all DIDMs (Corollary 65 in (Rauchwerger et al., 2025)). Therefore, the space of graphon-DIDMs is a strict compact subspace of the bofop-DIDM space, which is a strict subset of the space of all DIDMs, which is compact itself. This established a hierarchy of DIDM spaces, useful for theoretical results in graph machine learning, where dense graphs form a proper subspace of sparse ones, while finite graph-signals are dense in the corresponding bofop-DIDM space. This hierarchy is summarized in figure 1.

## 6. Theoretical Applications

Next, we state the main theoretical applications of the compactness of the space of bofop-DIDMs, the uniform equicontinuity and separation power of MPNNs with respect to the DIDM mover's distance.

### 6.1. Universal Approximation

It was already known in (Rauchwerger et al., 2025) that the space of *all* DIDMs is compact, and MPNNs form a separating algebra of continuous functions with respect to the DIDM-mover's distance. This leads to a universal approximation: every continuous function over the domain of all DIDMs can be approximated by an MPNN. However, we show in Corollary 5.3 that the space of all DIDMs is strictly larger than the space of bofop-DIDMs. Namely, "most" DIDMs are formal objects that do not describe the structure of any bofop-signal. Since we are only interested in bofop-DIDMs, the above universal approximation theorem is not satisfactory. A priori, without establishing the compactness of bofop-DIDMs, it is not clear if, given a continuous function on the space of bofop-DIDMs, it can be extended to a continuous function over all DIDMs. Luckily, our compactness result, Corollary 5.3, allows us to formulate a universal approximation theorem directly on bofop-DIDMs: any continuous function over the space of bofop-DIDMs (with respect to DIDM-mover's distance) can be uniformly approximated by an MPNN. Moreover, since graph-signals

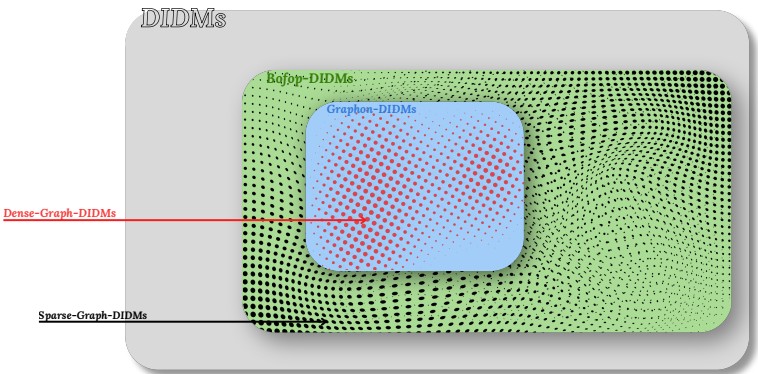

*Figure 1.* Hierarchy of DIDM spaces: the space of all DIDMs (gray); its strict subset of graphon-DIDMs, isomorphic to graphon-signals (blue); the space of dense-DIDMs, isomorphic to graph-signals, shown as a dense subset (red dots); the bigger set of bofop-DIDMs isomorphic to bofop-signals (green); the space of sparse graph-signals shown as a dense subset (black dots).

are dense in the space of bofop-signals with respect to the DIDM-Mover's distance, the same approximation result applies in particular to continuous functions from the space of graph-signals. For the formal result, see Appendix N.

### 6.2. Generalization Bound for MPNNs

In Appendix N, we prove the existence of a generalization bound for MPNNs over the space of bofop-DIDMs. The result states that for MPNN models over the class $\mathcal{MP}_D(d, d_0, \ldots, d_L, p)$, the generalization error - that is, difference between the empirical risk and the statistical risk, converge to zero as the sample size tends to infinity. This follows from the uniform equicontinuity of MPNNs and the compactness of the space $(\mathcal{BF}_d^r(\Omega), \delta_{\text{DIDM}}^L)$, which therefore admits a finite covering. The rate of convergence is shown to depend on the covering number of the space $(\mathcal{BF}_d^r(\Omega), \delta_{\text{DIDM}}^L)$. While the covering number is proven to be finite, we do not have a concrete bound on it, so our generalization bound is implicit. Full details and a formal proof are given in Appendix N.2

### 7. Conclusion

In this work, we introduced a unified approach for analyzing MPNNs on both sparse and dense graphs. Based on the work of (Backhausz & Szegedy, 2022), we obtained the compact metric space of bofop-signals – graph-like objects that is suited for sparse graphs, while unifying several graph limit theories. Our main extension of graphop theory involves: 1) incorporating node features, leading to a graphop-signal analysis, 2) focusing on a subclass of graphop-signals, which have special properties enabling a canonical extension of MPNNs to profiles, and making MPNNs Lipschitz continuous with respect to action metric,

and, 3) unifying graphop theory with the theory of DIDMs, showin that MPNNs are also uniformly equicontinuous with respect to the DIDM-mover's distance. This leads to powerful universal approximation and generalization theorems.

Regarding point 2, we note that the restriction of the theory to bofops is critical, as our continuity analysis does not work for general graphops in $\mathcal{G}_{\infty,1}(\Omega)$. In fact, we conjecture that MPNNs are not uniformly equicontinuous with respect to general graphops. Point 3 is a crucial step in the integration of grphop theory into graph machine learning, as the action metic is too fine to describe the separation power of MPNNs. In contrast, DIDMs describe the computational tree structure of MPNNs, and the DIDM-mover's distance matches the separation power of MPNNs. Along the way, we have introduced a new hierarchy of compact DIDM spaces, which describe the strict inclusion *dense objects* $\subset$ *sparse objects* $\subset$ *formal computational structures*.

Future work may explore the space of DIDMs beyond the class of bofops, and study larger classes of graphops. Another important open question is whether one can obtain explicit bounds on the covering number of the space of bofops, either in the action metric or the DIDM-mover's distance. Such formulae exist for graphon-signals with respect to cut distance (Levie, 2024), and would lead to a more explicit generalization bound for bofop-signals.

**Limitations.** Our construction of MPNNs on profiles is purely theoretical. Since profiles are defined as *sets* of measures, with no additional structure, they are not directly amenable to computational mathematics. One possible future avenue is to endow additional structure on profiles, like a probability measure, that would allow using them as a data structure for computations.

## Impact Statement

This paper presents work whose goal is to advance the field of Machine Learning. There are many potential societal consequences of our work, none which we feel must be specifically highlighted here.

## Acknowledgements

This research was supported by a grant from the United States-Israel Binational Science Foundation (BSF), Jerusalem, Israel, and the United States National Science Foundation (NSF), (NSF-BSF, grant No. 2024660), and by the Israel Science Foundation (ISF grant No. 1937/23).

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

# Appendix

## *Background*

For the convenience of the reader, we outline the contents of the appendices. Appendices A to E serve as background on topics relevant to the paper. Appendices F through N contain our contributions, extending the results from main paper, where we present full formulations and detailed proofs.

In appendix A we provide brief background on fundamental mathematical concepts from topology and measure theory. Appendix B combine these two areas and introduces the notion of weak topology, namely a topology defined on spaces of measures, together with a metrization of this space via an appropriate metric.

In appendix C we review background on machine learning, introduce the main network studied in this paper-MPNN, and discuss generalization theorem and universal approximation results. Appendix D covers background in graph limit theory, including well known limit objects of graphs such as graphons. Finally, appendix E is devoted to iterated degree measures, one of the central topics of this paper.

## A. Basic Background in Analysis

In the following sections, we provide the necessary background on mathematical topics from topology and measure theory that will be used throughout the paper.

### A.1. Basic Concepts From Topology

Point-set topology is a foundational branch of topology, providing the basic setting on which other areas of topology are built. Point-set topology allows notions such as neighborhoods and continuity to be studied independently of distance. The basic objects of study are topological spaces, which are sets endowed with a family of subsets, called open sets, satisfying certain axiomatic properties.

#### A.1.1. SOME PROPERTIES OF TOPOLOGICAL SPACES

There are various important and useful properties of topological spaces. We will focus on separability and compactness.

**Definition A.1** (Separable Space). A topological space is called separable if there exists a dense countable subset.

**Definition A.2** (Compact spaces). A topological space $(\mathcal{X}, \tau)$ is called a compact space if for every open cover of $\mathcal{X}$ there is a finite subcover.

#### A.1.2. THE PRODUCT TOPOLOGY

The product topology is a topology naturally defined on the Cartesian product of some topological spaces.

Let $\{(\mathcal{X}_i, \tau_i)\}_{i \in I}$ be a family of topological spaces, where $I$ is non-empty index set.. Their product space is denoted by $\prod_{i \in I} \mathcal{X}_i$ and consists of all tuples $(x_i)_{i \in I}$ such that $x_i \in \mathcal{X}_i$ for each $i \in I$. The product topology on $\prod_{i \in I} \mathcal{X}_i$ is the topology generated by the basis $\left\{ \prod_{i \in I} \mathcal{O}_i \ : \ \mathcal{O}_i \in \tau_i \right\}$.

For each $j \in I$, the projection map $p_j : \prod_{i \in I} \mathcal{X}_i \to \mathcal{X}_j$ is defined by $p_j((x_i)_{i \in I}) = x_j$. With respect to the product topology, all projection maps $p_j$ are continuous. Often, the product topology is defined as the coarsest topology (that is, the topology with the fewest open sets) for which each projection $p_j$ is continuous. Moreover, a sequence in the product space converges if and only if its projections onto each factor space converge.

One of the most important theorems in topology is the "Tychonoff Theorem", which states that any product of compact topological spaces (not necessarily finite or even countable) is compact in the product topology. More precisely:

**Theorem A.3** (Thychonoff Theorem). *Let $\{\mathcal{X}_j\}_{j \in I}$ be a family of compact topological spaces, where $I$ is an arbitrary set of indexes. Then their product $\prod_{j \in I} \mathcal{X}_j$ is a compact space.*

### A.1.3. METRIC SPACES

A metric space is a topological space on which a distance function is defined. This distance function is called a metric.

**Definition A.4** (Pseudometric Spaces and Metric Space). A pseudometric space is a pair $(\mathcal{X}, d)$, where $\mathcal{X}$ is a set and $d : \mathcal{X} \times \mathcal{X} \to \mathbb{R}$ is a function that satisfies:

1. $d(x_1, x_2) \geq 0$ for every $x_1, x_2 \in \mathcal{X}$.

2. $d(x_1, x_2) = d(x_2, x_1)$ for every $x_1, x_2 \in \mathcal{X}$.

3. $d(x_1, x_3) \leq d(x_1, x_2) + d(x_2, x_3)$ for every $x_1, x_2, x_3 \in \mathcal{X}$.

A metric, is a pseudometric with the additional following condition. For every $x_1, x_2 \in \mathcal{X}$, $d(x_1, x_2) = 0 \iff x_1 = x_2$.

The topology induced by a metric or a pseudometric, is the topology generated by the collection of open balls $\mathbb{B}_r(x_0) = \{x \in \mathcal{X} \mid d(x_0, x) < r\}$, for some $x_0 \in \mathcal{X}$ and $r > 0$.

### A.1.4. LIPSCHITZ AND HÖLDER CONTINUITY

Given two metric spaces $(\mathcal{X}, d_{\mathcal{X}}), (\mathcal{Y}, d_{\mathcal{Y}})$ a function $f : \mathcal{X} \to \mathcal{Y}$ is said to be Lipschitz continuous if there is $C \geq 0$ such that for every $x_1, x_2 \in \mathcal{X}$,

$$d_{\mathcal{Y}}(f(x_1), f(x_2)) \leq C \cdot d_{\mathcal{X}}(x_1, x_2). \tag{5}$$

The minimum $C$ satisfying (5) is often called the *Lipschitz constant* and often denoted by $L_f$. Let $Lip(\mathcal{X}, L)$ denote the space of all Lipschitz continuous functions $f : \mathcal{X} \to \mathcal{Y}$ with Lipschitz constant $L$.

Similarly, $f : \mathcal{X} \to \mathcal{Y}$ is Hölder continuous with exponent $\alpha > 0$ if there is $C \geq 0$ such that for every $x_1, x_2 \in \mathcal{X}$,

$$d_{\mathcal{Y}}(f(x_1), f(x_2)) \leq C \cdot \big(d_{\mathcal{X}}(x_1, x_2)\big)^{\alpha}. \tag{6}$$

The minimum $C$ satisfying (6) is often called *Hölder constant* and will be denoted as $C_f$. Let $Hol^{\alpha}(\mathcal{X}, C)$ be the space of all Hölder continuous functions $f : \mathcal{X} \to \mathcal{Y}$ with Hölder constant $C$ and exponent $\alpha$. Note that Lipschitz continuity is a particular case of Hölder continuity in which $\alpha = 1$.

### A.1.5. UNIFORM EQUICONTINUITY

The notion of Lipschitz and Hölder continuity can be extended to a more general form of continuity.

**Definition A.5** (Uniform Equicontinuity). Let $(\mathcal{X}, d_{\mathcal{X}})$ and $(\mathcal{Y}, d_{\mathcal{Y}})$ be metric spaces. A family $\mathcal{F}$ of functions between $\mathcal{X}$ and $\mathcal{Y}$ is called *uniformly equicontinuous* if for every $\epsilon > 0$ there exists $\delta > 0$ such that if $d_{\mathcal{X}}(x_1, x_2) < \delta$ and $f \in \mathcal{F}$, then $d_{\mathcal{Y}}(f(x_1), f(x_2))$.

Note that the choice of $\delta$ depends only on $\epsilon$ and not any particular $f \in \mathcal{F}$. It is easy to see that both $Lip(\mathcal{X}, C)$ and $Hol^{\alpha}(\mathcal{X}, C)$ are equicontinuous.

Uniform equicontinuity can equivalently be characterized by the existence of a common modulus of continuity (Pugh & Pugh, 2002).

**Proposition A.6.** *Consider the above setting. The family $\mathcal{F}$ is uniformly equicontinuous, if there exists a strictly increasing function $\sigma : [0, \infty) \to [0, \infty)$ that satisfies $\sigma(0) = 0$ and $\lim_{t \to 0} \sigma(t) = 0$ such that $d_{\mathcal{Y}}(f(x_1), f(x_2)) \leq \sigma(d_{\mathcal{X}}(x_1, x_2))$.*

Again, it is easy to see that this notion extends both Lipschitz and Hölder continuity by choosing $\sigma(t) = C \cdot t$ and $\sigma(t) = C \cdot t^\alpha$ respectively.

### A.1.6. SEQUENTIAL SPACES

An important property of metric spaces is that they are sequential spaces. A sequential space is a topological space whose topology is completely characterized by the convergence (or divergence) of sequences.

In metric spaces, compactness is equivalent to sequential compactness, that is, every sequence has a convergent subsequence.

**Definition A.7** (Sequential Compactness). Let $\mathcal{X}$ be a topological space. We say that $\mathcal{X}$ is sequentially compact if every sequence $\{x_n\}_{n=1}^\infty$ has a convergent subsequence, and the limit is an element of $\mathcal{X}$.

As mentioned above, in metric spaces compactness and sequential compactness are equivalent.

**Theorem A.8.** *Let $(\mathcal{X}, d)$ be a metric space. Then, $\mathcal{X}$ is compact if and only if $\mathcal{X}$ is sequentially compact.*

### A.1.7. QUOTIENT SPACE

For a pseudometric space $(\mathcal{X}, d)$, we define an equivalence relation $\sim_d$ on $\mathcal{X}$ by

$$x \sim_d y \quad \Longleftrightarrow \quad d(x, y) = 0.$$

The associated quotient map $q : \mathcal{X} \to \mathcal{X}/\sim_d$ which defined by $q(x) = [x]$, identifies points with zero distance, where $[x]$ is the equivalence class of $x$. The quotient space $\mathcal{X}/\sim_d$ is a metric space when endowed with the metric

$$\tilde{d}([x], [y]) := d(x, y),$$

and this metric is well defined.

The topology on $\mathcal{X}/\sim_d$ is defined so that a set $\mathcal{O} \subset \mathcal{X}/\sim_d$ is open if and only if $q^{-1}(\mathcal{O})$ is open in $\mathcal{X}$. Equivalently, an open set in $\mathcal{X}$ that contains a point $x$ must contain the entire equivalence class $[x]$.

The following is one of the most important properties of sequential spaces and quotient spaces, proven by S. P. Franklin (Franklin, 1965).

**Theorem A.9.** *Let $(\mathcal{X}, \tau)$ be a topological space. Then $\mathcal{X}$ is sequential space if and only if $\mathcal{X}$ is a quotient space of some metric space.*

### A.1.8. POLISH SPACES

Lastly, we define two properties of topological spaces: metrizability - the existence of metric that induce the topology, and completeness - convergence of Cauchy sequences.

**Definition A.10** (Metrizable Spaces). A topological space $(\mathcal{X}, \tau)$ is called metrizable if there exists a metric $d : \mathcal{X} \times \mathcal{X} \to \mathbb{R}$ such that the topology induced by $d$ coincides with the topology $\tau$ on $\mathcal{X}$.

**Definition A.11** (Complete Metric space). Let $(\mathcal{X}, d)$ be a metric space. A sequence $\{x_n\} \subset \mathcal{X}$ is called a Cauchy sequence if for every $\varepsilon > 0$ there exists $N \in \mathbb{N}$ such that for all $m, n \geq N$,

$$d(x_n, x_m) < \varepsilon.$$

A metric space $(\mathcal{X}, d)$ is called complete if every Cauchy sequence in $\mathcal{X}$ converges to a point in $\mathcal{X}$.

We finish this section with the last important topological spaces - Polish spaces, which combine few topological properties we defined.

**Definition A.12.** A topological space is called a Polish space if it is separable and completely metrizable, i.e., there exists a complete metric that generates its topology.

### A.2. Basic Concepts From Probability and Measure Theory

Measure theory is a branch of mathematics that formalizes and extends classical notions of area and volume to very general sets and functions. Probability theory uses measure theory by modeling events as measurable sets (i.e. sets with well-defined volume), probabilities of events as their measures, and random variables as measurable functions.

**Definition A.13** (probability space). The triple $(\Omega, \Sigma, \mu)$ where $\Omega$ is some set, $\Sigma$ is a $\sigma$-algebra on $\Omega$, and $\mu$ is a probability measure on $\Omega$ is called a probability space which often be denoted $(\Omega, \mu)$.

Integration in measure theory extends the notion of summation to continuous spaces. Integrating a measurable function can be viewed as taking a weighted sum of its values, with weights determined by the measure. For example, in probability theory, the expectation of a random variable is exactly its integral with respect to the probability measure.

### A.2.1. BOREL AND LEBESGUE $\sigma$-ALGEBRAS

Let $(\mathcal{X}, \tau)$ be a topological space. The Borel $\sigma$-algebra on $\mathcal{X}$ is the smallest $\sigma$-algebra that contains all open sets in $\tau$, i.e. it is generated by the topology. Sets in the Borel $\sigma$-algebra are called Borel sets and include, in particular, all open and closed sets, as well as sets obtained from them by countable unions, countable intersections, and complements.

The Lebesgue $\sigma$-algebra on $\mathcal{X}$ is obtained by completing the Borel $\sigma$-algebra with respect to some measure, that is, by adding all subsets of Borel sets with measure zero. Integration with respect to the Lebesgue measure is the standard notion of integration used in functional analysis.

### A.2.2. LEBESGUE FUNCTION SPACES

For a Lebesgue measurable set $E \subset \mathbb{R}^n$ let $\lambda(E)$ denote the Lebesgue measure of $E$, and let $\mathbb{1}_E$ be the indicator function of $E$. Unless stated otherwise, we denote by $(\Omega, \mu)$ a general Borel probability space.

**Definition A.14** ($L^p$-Signals). A measurable function $f : \Omega \to \mathbb{R}^d$ is said to be an element of $\mathcal{L}^p(\Omega)$, when $1 \leq p < \infty$ if

$$\|f\|_p^p := \int_\Omega |f|^p \, d\mu < \infty.$$

For $p = \infty$, we say that $f$ is an element of $\mathcal{L}^\infty(\Omega)$ if

$$\|f\|_\infty := \inf\{ C \in \mathbb{R}_+ \; : \; |f(x)| < C \text{ for a.e. } x \in \Omega\} < \infty,$$

where "a.e." stands for "almost every", that is up to a set with measure $0$.

As usual, we consider elements of $\mathcal{L}^p$ to be equivalence classes of functions which are equal almost everywhere.

### A.2.3. PRODUCT MEASURE SPACES

Given two measurable spaces with measures defined on them, one can define their product measurable space together with a measure on it, called the product measure. Although there are many possible measures on the product space, the product measure is the most natural choice.

**Definition A.15** (Product Measurable Spaces and Product Measures). Let $(\Omega_1, \Sigma_1, \mu_1)$ and $(\Omega_2, \Sigma_2, \mu_2)$ be measurable spaces.

1. We define the $\sigma$-algebra on the Cartesian product $\Omega_1 \times \Omega_2$, and denote by $\Sigma_1 \otimes \Sigma_2$ the $\sigma$-algebra generated by sets of the form $E_1 \times E_2$ where $E_1 \in \Sigma_1$ and $E_2 \in \Sigma_2$.

2. We define the product measure $\mu_1 \times \mu_2$ (often denoted by $\mu_1 \otimes \mu_2$) as the measure on $\Omega_1 \times \Omega_2$ that satisfies

$$(\mu_1 \times \mu_2)(E_1 \times E_2) = \mu_1(E_1) \cdot \mu_2(E_2),$$

where $E_1 \in \Sigma_1$ and $E_2 \in \Sigma_2$.

### A.2.4. MARGINAL MEASURES

When a measure is defined on a product space, it often represents a joint distribution of a sequence of random variables. In many situations, one is interested in the distribution of a single component, ignoring the others. Marginal measures formalize this idea by extracting the measure induced on each factor space from the joint measure on the product space.

**Definition A.16** (Marginal Measure). Let $\Omega_1, \Omega_2$ be two measurable spaces, and $\Omega = \Omega_1 \times \Omega_2$ the product measurable space with measure $\mu$. The marginal measure $\mu_{\Omega_1}$ of $\mu$ on $\Omega_1$ is the measure

$$\mu_{\Omega_1}(E_1) = \mu(E_1 \times \Omega_2)$$

for every measurable set $E_1 \subset \Omega_1$. Similarly, the marginal measure $\mu_{\Omega_2}$ of $\mu$ on $\Omega_2$ is the measure

$$\mu_{\Omega_2}(E_2) = \mu(\Omega_1 \times E_2),$$

for every measurable set $E_2 \subset \Omega_2$.

Marginal measures are uniquely determined by the underlying measure on the product space. In particular, if two measures on the product space coincide, then their induced marginals also coincide.

### A.2.5. PUSHFORWARD MEASURES

A pushforward measure in measure theory is a measure obtained by transferring a measure from one measurable space to another via a measurable function. In probability theory, pushforward measures describe the distributions of random variables as well as the joint distributions of collections of random variables.

**Definition A.17** (Pushforward Measure). Given two measurable spaces $(\Omega_1, \Sigma_1)$ and $(\Omega_2, \Sigma_2)$, a measurable function $f : \Omega_1 \to \Omega_2$ and a measure $\nu : \Sigma_1 \to [0, \infty]$, the pushforward of $\nu$ via $f$ is defined as the measure $f_*(\nu) : \Sigma_2 \to [0, \infty]$ given by

$$f_*(\nu)(E) = \nu\big(f^{-1}(E)\big), \quad \forall E \in \Sigma_2.$$

The following is a simple, yet useful result, showing that the pushforward of a measure under a composition of measurable functions equals the pushforward obtained by applying the two functions in sequence.

**Proposition A.18** (Composition Of Pushforward). *For two measurable functions* $f : (\Omega_1, \Sigma_1) \to (\Omega_2, \Sigma_2)$ *and* $g : (\Omega_2, \Sigma_2) \to (\Omega_3, \Sigma_3)$ *and a measure* $\nu : \Sigma_1 \to [0, \infty]$, *we have that the pushforward of* $\nu$ *via* $g \circ f$ *is equal to the pushforward of* $f_*\nu$ *via* $g$. *Namely*

$$(g \circ f)_*\nu = g_*(f_*\nu).$$

The main property of pushforward measures is the following "change of variables" formula, demonstrating how integration against the measure is performed (see Theorem 3.6.1 in (Bogachev, 2007)).

**Theorem A.19** (Change Of Variables). *Let* $(\Omega_1, \Sigma_1, \nu)$ *and* $(\Omega_2, \Sigma_2)$ *be measurable spaces. Let* $f : \Omega_1 \to \Omega_2$ *be a measurable function, and denote by* $f_*\nu$ *the pushforward measure on* $\Omega_2$. *Let* $g : \Omega_2 \to \mathbb{R}$ *be a measurable function. Then* $g$ *is integrable with respect to* $f_*\nu$ *if and only if* $g \circ f$ *is integrable with respect to* $\nu$. *In this case,*

$$\int_{\Omega_2} g(y)\, d(f_*\nu)(y) = \int_{\Omega_1} g(f(x))\, d\nu(x).$$

*Remark* A.20 (Equality of Marginal Measures and Pushforward Measures). The notion of marginal measures (Definition A.16) can be expressed naturally in terns of pushforward measures under the coordinate projection maps. In particular, if $p_{\Omega_1} : \Omega_1 \times \Omega_2 \to \Omega_1$ denotes the projection onto $\Omega_1$, then the marginal measure $\mu_{\Omega_1}$ is the pushforward of $\mu$ under $p_{\Omega_1}$, namely $\mu_{\Omega_1} = (p_{\Omega_1})_*\mu$. Similarly, if $p_{\Omega_2} : \Omega_1 \times \Omega_2 \to \Omega_2$ is the projection onto $\Omega_2$, then $\mu_{\Omega_2} = (p_{\Omega_2})_*\mu$.

Motivated by the equivalence between marginal measures and pushforwards via coordinate projections, one can consider the inverse problem of constructing a joint measure whose marginals are given. This leads to the notion of a coupling between probability measures.

**Definition A.21** (Couplings). Let $(\Omega_1, \Sigma_1)$ and $(\Omega_2, \Sigma_2)$ be measurable spaces, and let $\nu_1$ and $\nu_2$ be probability measures on $\Omega_1$ and $\Omega_2$, respectively. A *coupling* of $\nu_1$ and $\nu_2$ is a probability measure $\gamma$ on the product measurable space $(\Omega_1 \times \Omega_2, \Sigma_1 \otimes \Sigma_2)$ such that its marginal measures satisfy

$$\gamma_{\Omega_1} = \nu_1 \quad \text{and} \quad \gamma_{\Omega_2} = \nu_2.$$

The set of all couplings of $\nu_1$ and $\nu_2$ is denoted by $\Gamma(\nu_1, \nu_2)$.

A.2.6. DISINTEGRATION

The disintegration theorem is a result in measure theory and probability theory that gives a rigorous meaning to the idea of restricting a measure to subsets that may have measure zero. In this sense, disintegration can be viewed as the inverse operation to the construction of a product measure.

**Theorem A.22** (The Disintegration Theorem). *Let $(\Omega, \Sigma, \mu)$ be a probability space, let $(\mathcal{X}, \tau)$ be a Borel space, and let $f : \Omega \to \mathcal{X}$ be a measurable function. Denote by $\nu = f_* \mu$ the pushforward measure on $\mathcal{X}$.*

*Then, there exists a family of probability measures $\{\mu_x\}_{x \in \mathcal{X}}$ on $(\Omega, \Sigma)$, called* fiber measures, *such that:*

1. *For $\nu$-almost every $x \in \mathcal{X}$,*
$$\mu_x\big(\Omega \setminus f^{-1}(\{x\})\big) = 0.$$

2. *For every $E \in \Sigma$, the map*
$$x \mapsto \mu_x(E)$$

 *is measurable.*

3. *For every integrable function $\varphi : \Omega \to \mathbb{R}$,*

$$\int_{\Omega} \varphi(\omega) \, d\mu(\omega) = \int_{\mathcal{X}} \left( \int_{\Omega} \varphi(\omega) \, d\mu_x(\omega) \right) d\nu(x). \tag{7}$$

*The family $\{\mu_x\}_{x \in \mathcal{X}}$ is unique up to $\nu$-almost everywhere equality and is called the disintegration of $\mu$ with respect to $f$.*

Intuitively (and informally), $\mu_x$ can be interpreted as a measure supported on the level set $\{w \in \Omega \mid f(w) = x\}$. Under this interpretation, the integral on the left-hand side of (7) is computed by a repeated integral on the right-hand side, where $\mathcal{X}$ parametrizes the set of level sets, and the inner integral integrates along each level set. The space $\mathcal{X}$ is often called the *carrier space*. To illustrate this idea consider the following example. Let $\Omega = [0, 1]^2$ with $\mu$ being the Lebesgue measure $\lambda^2$ on $\mathbb{R}^2$. Consider a one-dimensional subset of $\Omega$ such as $L_x := \{x\} \times [0, 1]$, for some $x \in [0, 1]$. while $L_x$ has $\lambda^2$-measure zero, we would like that the measure $\lambda^2$ "restricted" to the one dimensional line $L_x$ will correspond to the Lebesgue measure over $[0, 1]$. In this way, the measure of a measurable subset $E \subset [0, 1]^2$ can be obtained by integration of the one dimensional slices $E \cap L_x$, over all slices. If $\mu_x$ denotes the one dimensional Lebesgue measure on $L_x$ then

$$\mu(E) = \int_{[0,1]} \mu_x(E \cap L_x) \, dx.$$

# B. Special Topologies and Metrics for Sets and Measures

Understanding convergence of probability measures is a central problem in probability theory. Suitable topologies and metrics provide a way to describe such convergence.

## B.1. Weak and Weak* Topology of Spaces of Measures

Let $(\mathcal{X}, d)$ be a metric space equipped with its Borel $\sigma$-algebra. The weak topology on the space of probability (or finite) measures on $\mathcal{X}$ is the coarsest topology for which the maps $\mu \mapsto \int f \, d\mu$ are continuous for all bounded continuous functions $f : \mathcal{X} \to \mathbb{R}$. Equivalently, a sequence of measures converges in the weak topology if and only if the integrals of any fixed bounded continuous function on the sequence of measures converge (Folland, 1999).

There are additional equivalent definitions of weak convergence of sequences of measures. The equivalence of these conditions is often known as the Portmanteau Theorem. A nice property of the weak topology is that if $\mathcal{X}$ is separable, compact or polish, then so as the space of probability measures over $\mathcal{X}$, induced with the weak topology.

In addition to the weak topology, there is a topology often used on spaces of measures, known as the weak* topology. Roughly speaking, the weak* topology is defined similarly to the weak topology, but with test functions that are continuous and vanish at infinity. When $(\mathcal{X}, d)$ is compact metric space, the weak and weak* topologies coincide on $\mathcal{P}(\mathcal{X})$.

## B.2. Levy-Prokhorov Metric

Let $(\mathcal{X}, d)$ be a metric space equipped with the Borel $\sigma$-algebra, and let $\mathcal{P}(\mathcal{X})$ be the space of all Borel probability measures over $\mathcal{X}$. The *Levy-Prokhorov* metric $d_{\mathrm{LP}}$ is a metric over $\mathcal{P}(\mathcal{X})$ that metrize the weak topology.

**Definition B.1** (Levy-Prokhorov metric)**.** The Levy-Prokhorov metric $d_{\mathrm{LP}}$ on $\mathcal{P}(\mathcal{X})$ is defined by

$$d_{\mathrm{LP}}(\mu, \nu) = \inf \left\{ \epsilon > 0 \,\middle|\, \mu(E) \leq \nu(E^\epsilon) + \epsilon \text{ and } \nu(E) \leq \mu(E^\epsilon) + \epsilon \text{ for every measurable } E \subset \mathcal{X} \right\},$$

where $E^\epsilon$ is the $\epsilon$-neighborhood of $E$ is defined by

$$E^\epsilon := \left\{ x \in \mathcal{X} \,\middle|\, \exists e \in E, d(x,e) < \epsilon \right\}.$$

## B.3. Hausdorff Distance

The Hausdorff distance, or Hausdorff metric, measures how far two subsets of a metric space are from each other. The Hausdorff distance was first introduced by Felix Hausdorff in his book (Hausdorff, 1914). Informally, the Hausdorff distance measures the largest distance from a point in one set to the closest point in the other, taken symmetrically between the two sets.

**Definition B.2** (Hausdorff Distance)**.** Let $(\mathcal{X}, d)$ be a metric space and $A, B \subset \mathcal{X}$ be two non-empty subsets of $\mathcal{X}$. The *Hausdorff distance* $d_H$ between $A$ and $B$ is defined by

$$d_H^d(A, B) := \max \left\{ \sup_{a \in A} \inf_{b \in B} d(a,b) \,,\, \sup_{b \in B} \inf_{a \in A} d(a,b) \right\}.$$

Where the context is clear, we will often omit the base metric from the notation, and write $d_H$ instead of $d_H^d$. Note that $d_H(A, B) = 0$ if and only if $\mathrm{cl}(A) = \mathrm{cl}(B)$, where $\mathrm{cl}$ is the closure in $d$. It follows that $d_H$ is a pseudometric for all subsets in $\mathcal{X}$, and it is a metric for closed sets.

Let $\mathscr{C}(\mathcal{X})$ denote the space of all closed subsets of $\mathcal{X}$. An important property of the Hausdorff metric is that the space $(\mathscr{C}(\mathcal{X}), d_H)$ "inherits" some topological properties of $\mathcal{X}$ (See Theorems 7.3.7, 7.3.8 in (Burago et al., 2001)).

**Theorem B.3.** *Let $(\mathcal{X}, d)$ be a metric space. Then:*

1. *If $\mathcal{X}$ is complete, then $(\mathscr{C}(\mathcal{X}), d_H)$ is also complete.*

2. *If $\mathcal{X}$ is compact, then $(\mathscr{C}(\mathcal{X}), d_H)$ is also compact.*

# C. Background in Machine Learning and GNNs

In this section, we review the basic concepts from machine learning and graph neural networks (GNNs) that are required for the analysis developed in this paper.

## C.1. Attributed graphs

A graph is a pair $G = (V, E)$, where $V$ is the node set, and the edge set is given by $E \subset V \times V$, where $(v, u) \in E$ if and only if $v$ and $u$ are connected by an edge. We denote by $|V|$ the number of nodes in the graph, and by $|E|$ as the number of edges. For $v \in V$, let $\mathcal{N}(v) = \{u \in V | (v, u) \in E\}$ be the *neighborhood* of $v$, and $\deg(v) := |\mathcal{N}(v)|$ as the node's degree.

An attributed graph is a graph $G = (V, E)$, where each $v \in V$ is equipped with a feature vector $\mathrm{f}_v \in \mathbb{R}^d$, and $d$ is called the feature dimension. The pair $(G, \mathbf{f})$ is often referred to as *graph-signal*, where the mapping $\mathbf{f} : V \to \mathbb{R}^d$, assigns a feature value $\mathbf{f}(v) := \mathrm{f}_v$ to every node.

## C.2. Message Passing Graph Neural Networks

Message passing neural networks (MPNNs) (Gilmer et al., 2017), are a class of neural networks designed for graph structured data. MPNNs takes graphs, or attributed graphs as an input (in case of graphs without features, the signal is taken to be the constant function 1), and iteratively update node features through message passing, in which features from the neighbors are

aggregated. Common aggregated schemes include summation, averaging and coordinate-wise maximum. Our focus will be on normalized sum aggregation, which has been shown to achieve performance comparable to sum aggregation (Levie, 2024). In the context of performing graph regression or classification, the output of the MPNN is given by global pooling, which is an aggregation of the set of all nodes features.

We define an MPNN model as follows,

**Definition C.1** (MPNN model). Let $L \in \mathbb{N}_0$ and let $p, d_0, \ldots, d_L, d \in \mathbb{N}_0$. An *L-layer MPNN model* is a collection $\varphi = (\varphi^{(l)})_{l=0}^L$ of Lipschitz continuous functions, where

$$\varphi^{(0)} : \mathbb{R}^d \to \mathbb{R}^{d_0}, \qquad \varphi^{(l)} : \mathbb{R}^{2d_{l-1}} \to \mathbb{R}^{d_l}, \quad 1 \leq l \leq L,$$

where the functions $\varphi^{(l)}$ are called *update functions*. Given a Lipschitz continuous function $\psi : \mathbb{R}^{d_L} \to \mathbb{R}^p$, the tuple $(\varphi, \psi)$ is called an *MPNN model with readout*, where $\psi$ is referred to as the readout function. We call $L$ the depth of the MPNN, $d$ the input feature dimension, $d_0, \ldots, d_L$ the hidden feature dimensions, and $p$ the output feature dimension.

An MPNN which processes graph-signals is defined as follows.

**Definition C.2** (MPNN on Graph-signals). Let $(\varphi, \psi)$ be an L-layer MPNN model with readout, and $(G, \mathbf{A}, \mathbf{f})$ be a weighted graph-signal with adjacency matrix $\mathbf{A}$, where $\mathbf{f} : V(G) \mapsto \mathbb{R}^d$. The application of the MPNN on $(G, \mathbf{f})$ is defined as follows: initialize $\mathfrak{g}_-^{(0)} := \varphi^{(0)}(\mathbf{f}(-))$ and compute the hidden node representations $\mathfrak{g}_-^{(l)} : V(G) \to \mathbb{R}^{d_l}$ at layer $l$, with $1 \leq l \leq L$ and the graph-level output $\mathfrak{G} \in \mathbb{R}^p$ by

$$\mathfrak{g}_v^{(l)} := \varphi^{(l)}\left(\mathfrak{g}_v^{(l-1)}, \mathbf{A}\mathfrak{g}_-^{(l-1)}\right),$$

$$\mathfrak{G} := \psi\left(\frac{1}{|V|} \sum_{v \in V(G)} \mathfrak{g}_v^{(L)}\right).$$

The expression $\mathbf{A}\mathfrak{g}_-^{(l-1)}$ is called the *aggregated feature* of the node $v \in V(G)$.

Given an unweighted graph, one can normalize its adjacency matrix in various ways. For example, one can $\mathbf{A}_{i,j} = e_{i,j}/|V|$, where $e_{i,j} = 1$ if there is an edge between nodes $i$ and $j$, and 0 otherwise. In this case, the aggregation is called *normalized sum aggregation*, and takes the form $\frac{1}{|V|} \sum_{u \in \mathcal{N}(v)} \mathfrak{g}_u^{(l-1)}$.

The architecture defined above corresponds to the Graph Isomorphism Network (GIN) (Xu et al., 2019). While more general formulations of MPNNs allow explicit message functions (Levie, 2024), we show in Appendix F that any such MPNN can be implemented with our model using twice the number of layers.

### C.3. General Setting of Statistical Learning

In the statistical learning context, we assume that the dataset consists of independent and identically distributed (i.i.d.) random samples from a probability space that contains all possible data $\mathcal{X}$. Each input $x \in \mathcal{X}$ is associated with a ground truth value $y_x \in \mathcal{Y}$, where $\mathcal{Y}$ is a measurable output space, such as finite set of label in classification or a subset of $\mathbb{R}$ in regression. A learning model is represented by a measurable function $\Theta : \mathcal{X} \to \mathcal{Y}$ that is called a network. To quantify the accuracy of the model, one introduce a loss function $\mathcal{L} : \mathcal{Y}^2 \to \mathbb{R}_+$ that measures the similarity between the output of the networks $\Theta(x)$ and the ground truth value $y_x$. The accuracy of the network $\Theta$ across all inputs is called the *statistical risk*, and is defined as $R_{\text{stat}} = \mathbb{E}_{x \sim \mathcal{X}}\left[\mathcal{L}\big(\Theta(x), y_x\big)\right]$. The primary learning goal is to identify a network $\Theta$ that minimizes the statistical risk. In practice, however, the underlying data distribution is unknown, making the statistical risk intractable to compute directly. Instead, we rely on the availability of a dataset $X = \{x_i\}_{i=1}^N \subset \mathcal{X}$, consisting of $N \in \mathbb{N}$ random and independent samples, each associated with corresponding values $\{y_{x_i}\}_{i=1}^N \subset \mathcal{Y}$. We estimate the statistical risk via a "Monte Carlo approximation" (Caflisch, 1998) called the *empirical risk* $R_{\text{emp}}(\Theta) = \frac{1}{N} \sum_{j=1}^N \mathcal{L}\big(\Theta(x_j), y_{x_j}\big)$. A key subtlety is that the learned model itself depends on the sampled dataset through the training procedure, so the empirical risk is not a classical Monte Carlo estimate of the statistical risk for a fixed function. The practical selection of the network $\Theta$ involves optimizing the empirical risk. the goal in generalization analysis is to prove that low empirical risk implies low statistical risk. One proof technique is to bound the generalization error $E = \sup_{\Theta} \left|R_{\text{emp}}(\Theta) - R_{\text{stat}}(\Theta)\right|$. Situations in which a model achieves small empirical risk but large statistical risk are commonly described as overfitting. This is the philosophy behind approaches such as VC-dimension, Rademacher dimension etc. (Shalev-Shwartz & Ben-David, 2014).

## C.4. Generalization Analysis

The goal in generalization analysis is to derive conditions that guarantee that the performance of a trained network on the training set is close to the performance of the network on the test set. This analysis is important because the ability of the network to perform well on unseen data is a requirement for machine learning to be applicable to real world problems.

## C.5. Generalization Analysis of GNNs

Past works on MPNN generalization bounds, such as (Maskey et al., 2022; Liao et al., 2021; Scarselli et al., 2018; Garg et al., 2020), often rely on specific assumptions about data distributions and practical architectures. Recent contributions include a survey of generalization theory for GNNs that unifies complexity and stability-based approaches (Vasileiou et al., 2025b). Unlike prior graphon-based studies that impose data generative model assumptions (Maskey et al., 2022; 2025), Levie (Levie, 2024) derives uniform generalization bounds for any regular MPNNs over broad classes of various graph-signal distributions without an assumption on the data as generative model. Rauchwerger and Levie (Rauchwerger & Levie, 2025) further extend these results and obtained improved asymptotic rates in the resulting generalization bounds.

## C.6. Universal Approximation Theorems and Expressivity

The goal in expressivity analysis is to study what types of functions can be represented or approximated by neural networks, or, more generally, to study the ability of neural networks to distinguish between data points. Universal approximation theorems state that neural networks, under certain conditions, can approximate any continuous function on a compact domain with sufficient accuracy, given enough parameters such as the number of layers. These theorems provide a theoretical foundation for the expressivity of neural networks, suggesting that, in principle, neural networks have the capacity to represent a wide variety of functions.

The first universal approximation theorems asserted that over a compact space, every continuous function can be approximated by some multi-layer-perceptron (Cybenko, 1989; Funahashi, 1989; Leshno et al., 1993). The proof relies on the Stone-Weierstrass theorem (Rudin, 1976)[Theorem 7.32], which states that an algebra of continuous functions over some compact domain that separates points can approximate any continuous function.

**Definition C.3.** Let $K$ be a compact space. Consider the space of continuous functions from $K$ to $\mathbb{R}$, $C(K)$. Let $A \subset C(K)$,

1. $A$ is unital subalgebra if $1 \in A$, and if $f, g \in A$, $\alpha, \beta \in \mathbb{R}$ implies that $\alpha f + \beta g \in A$, and $fg \in A$.

2. $A$ separates points of $K$ if for $s, t \in K$ with $s \neq t$, there exists $f \in A$ such that $f(s) \neq f(t)$.

**Theorem C.4** (Stone-Weierstrass Theorem). *Let $K$ be a metric compact space, and $A \subset C(K)$ be a unital subalgebra which separates points of $K$. Then $A$ is dense in $C(K)$.*

## C.7. GNN Expressivity via Graph Isomorphism Tests

The study of the ability of GNNs to separate points is of interest on its own. Two graphs are separated by a GNN architecture if there is GNN that assigns different output values to these graphs. The study of the separation power of neural networks is often called *expressivity analysis* (Geerts & Reutter, 2022; Böker et al., 2024; Azizian & Lelarge, 2021; Xu et al., 2019; Morris et al., 2019; Rauchwerger et al., 2025). A GNN with greater expressive power can identify more nuanced distinctions between graphs, thus improving its capability in tasks like graph classification or regression.

## C.8. GNN Universal Approximation Theorems

Universal approximation theorems can also be derived for GNNs. For that, one needs to show that GNNs separate points, endow the input space of graphs with a suitable compact topology, and consider an algebra of functions that can be approximated by the network.

In (Xu et al., 2019) it was shown that GNNs possess expressivity comparable to the 1-WL isomorphism test, which distinguishes non-isomorphic graphs through iterative node label refinement. If the test fails, It can not determine whether the graphs are isomorphic or not. Based on this characterization, the authors established a universal approximation result for GNNs over graph isomorphism classes, showing that sufficiently expressive GNNs can approximate any function that is invariant under graph isomorphism and distinguishable by 1-WL.

In (Brüel Gabrielsson, 2020) a universal approximation theorem was proved for GNNs based on injective graph representations. The results show that injectivity of the aggregation and readout mechanisms is sufficient to guarantee universal approximation on bounded graphs, and enable approximation on unbounded graphs when restricted to compact subsets of the input space.

The work of (Geerts & Reutter, 2022) introduces a tensor-based algebraic framework used for analyzing GNNs, capturing operations such as summations and index manipulations. This framework is used to study the separation power of GNN architectures, leading to universality results for a broad class of models, including higher-order message passing neural networks.

## D. Background on Graph Limit Theory and GNNs

To analyze GNNs in a setting that is independent of graph size and node labeling, we consider limit objects of sequences of graphs. In graph limit theory, large graphs can be represented as elements of a compact metric space, allowing tools from topology and analysis to be applied.

### D.1. Graphon Analysis

Often, when analyzing the space of graphs, in order to include graph of all sizes, it is beneficial to extend the space to also include graphs with "continuous node sets" called graphons. A graphon is a symmetric measurable function $W : [0, 1]^2 \to [0, 1]$. Graphons arise both as a natural limit of sequences of dense graphs, and as the fundamental defining elements of exchangeable random graph models (Borgs et al., 2008; Lovász, 2012). A graphon can be seen as a probabilistic model from which finite graphs are sampled, by drawing i.i.d nodes $\{x_n\}$ from $[0, 1]$, and connecting each two nodes $x_i, x_j$ with probability $W(x_i, x_j)$. This concept that a graphon can be interpreted as a probabilistic model, and this interpretation is utilized to study the transferability of graph filters on different graphs sampled from the same graphon (Maskey et al., 2023).

Graphons can also be seen as graphs with continuous node set, and the value $W(x, y)$ represents the probability of the edge $(x, y)$ to be in the graph, or the edge weight between $x$ and $y$. Every finite graph $G = (V, E)$ with $n$ nodes and adjacency matrix $\{a_{i,j}\}_{i,j \in [n]}$ can be seen as a graphon. Let $\mathcal{P}_n = \{I_1, \ldots, I_n\}$ be a equipartition of $[0, 1]$ into $n$ disjoint intervals of equal length, then $G$ induces the picewise constant graphon $W_G$, defined by $W_G(x, y) = a_{\lceil xn \rceil, \lceil yn \rceil}$[2].

Let $\mathcal{W}_0$ denote the space of all graphons.

A kernel is a measurable function $K : [0, 1]^2 \to \mathbb{R}$.

**Definition D.1** (Cut -Norm). Let $K$ be a kernel.

$$\|K\|_\square = \sup_{S,T} \left| \int_S \int_T K(x, y) \, dx \, dy \right|,$$

where the supremum is taken over all measurable subsets of $[0, 1]$.

To define a measure of similarity between two graphons, we introduce the cut metric and the cut distance. The cut distance is a pseudometric on the space of graphons $\mathcal{W}_0$, considering all possible permutations of the graphons. When considering finite graphs for example, the cut distance reflects the fact that different labeling of the graph's nodes should not affect the measurement of similarity between the graphs.

**Definition D.2** (Cut-Distance). The cut metric between two graphons $W_1, W_2 \in \mathcal{W}_0$ is defined by

$$d_\square(W_1, W_2) = \|W_1 - W_2\|_\square.$$

We define the cut distance between two graphons $W_1, W_2$ by

$$\delta_\square(W_1, W_2) = \inf_\phi d_\square(W_1^\phi, W_2),$$

where the infimum is taken over all measure preserving bijections $\phi : [0, 1] \to [0, 1]$, and $W_1^\phi(x, y) := W_1(\phi(x), \phi(y))$ for a.e $x, y \in [0, 1]$.

---

[2]By convention $\lceil 0 \rceil = 1$

The cut distance is a pseudometric on the space of graphons $\mathcal{W}_0$. Two graphons $W_1, W_2 \in \mathcal{W}_0$ are considered equivalent, and denoted by $W_1 \sim W_2$, if $\delta_\square(W_1, W_2) = 0$. By abuse of terminology, going forward, a graphon is referred to as an equivalence class of graphons up to $\sim$. When considering equivalence classes of graphons, the cut distance becomes a metric.

When endowed with the cut distance, the resulting metric space $(\mathcal{W}_0, \delta_\square)$ of graphons is compact, and the space of graphs associated with their induced graphons is a dense subset of this space (Lovász & Szegedy, 2007).

### D.2. Graphon Analysis of GNNs

The study of GNNs via graphon analysis offers insights into the networks' expressivity and generalization across complex graph structures. Related graphon-based approaches have also been used to study transferability properties of GNNs. (Ruiz et al., 2020; 2023). The analysis of GNNs extends their understanding by introducing metrics on spaces of objects that generalize graphs, such as the graphon space and cut distance as described in the previous section.

(Levie, 2024) extends traditional graphon theory to graphon-signals, pairing graphs with signals on their nodes to form graphon-signals. A graph signal is a pair $(G, \mathbf{f})$, where $G$ is a graph with node set $[n]$ and signal $\mathbf{f} \in \mathbb{R}^{n \times d}$ that assigns the value $f_i \in \mathbb{R}^d$ to every node $i \in [n]$, that represents the node's features. See Appendix C.1. Graph-signals can be extended to graphon-signal, where the nodes set is represented by $[0, 1]$, and the signal is a multi-dimensional bounded measurable function on $[0, 1]$.

**Definition D.3** (Graphon-signal). The pair $(W, f)$ where $W \in \mathcal{W}_0$ is a graphon and $f : [0, 1] \to [-r, r]^d$ for some $r > 0$ is called a *graphon-signal*. Denote by $\mathcal{WL}_r^d$ the space of graphon-signals.

Similarly to graphs, graph-signals induce graphon-signals in the following way.

**Definition D.4** (Induced Graphon-signal). Let $(G, \mathbf{f})$ be a graph-signal with node set $[n]$ and adjacency matrix $\mathbf{A} = \{a_{i,j}\}_{i,j \in [n]}$. Let $\{I_k\}_{k=1}^n$ with $I_k = [(k-1)/n, k/n)$ be the equipartition of $[0, 1]$ into $n$ intervals. The graphon-signal $(W, f)_{(G,\mathbf{f})} = (W_G, f_\mathbf{f})$ induced by $(G, \mathbf{f})$ is defined by

$$W_G(x, y) = \sum_{i,j \in [n]} a_{ij} \mathbb{1}_{I_i}(x) \mathbb{1}_{I_j}(y), \quad \text{and} \quad f_\mathbf{f}(z) = \sum_{i=1}^n \mathbf{f}_i \mathbb{1}_{I_i}(z).$$

We use an extension of MPNN to process graphon-signals in such a way that application of message passing layer on an induced graphon-signal is equivalent to first applying an MPNN on graph-signal and then inducing it (Levie, 2024). We first recall the definition of an MPNN on graph-signals (Definition C.2), and consider the specific choice of normalized sum aggregation as the aggregation function.

**Definition D.5** (MPNNs on Graphon-signals.). Let $(\varphi, \psi)$ be an L-layer MPNN with readout, and $(W, f)$ be a graphon-signal where $f : [0, 1] \mapsto [-1, 1]^d$. The application of the MPNN on $(W, f)$ is defined as follows: initialize $\mathfrak{w}_-^{(0)} := \varphi^{(0)}(f(-))$, and compute the hidden node representation $\mathfrak{w}_-^{(t)} : [0, 1] \to \mathbb{R}^{d_t}$ at layer $t$, with $1 \leq t \leq L$ and the graphon-level output $\mathfrak{W} \in \mathbb{R}^d$ by

$$\mathfrak{w}_x^{(t)} := \varphi^{(t)} \left( \mathfrak{w}_x^{(t-1)}, \int_{[0,1]} W(x, y) \mathfrak{w}_y^{(t-1)} d\lambda(y) \right) \quad and \quad \mathfrak{W} := \psi \left( \int_{[0,1]} \mathfrak{w}_x^{(L)} d\lambda(x) \right)$$

Also denote $\mathfrak{w}(\varphi)_v^{(t)}$ or $\mathfrak{w}(\varphi, W, f)_v^{(t)}$, and $\mathfrak{W}(\varphi, \psi)$ or $\mathfrak{W}(\varphi, \psi, W, f)$ in order to stress the dependence of the $\mathfrak{w}$ and $\mathfrak{W}$ on $\varphi, \psi$ and $(W, f)$.

Indeed, Lemma E.1 in (Levie, 2024) justifies this definition by showing that applying an MPNN on an induced graphon-signal is equivalent to the application of an MPNN on graph signal and then inducing it. Namely,

$$\mathfrak{W}(\varphi, \psi, W_G, f_\mathbf{f}) = \mathfrak{G}(\varphi, \psi, G, \mathbf{f}). \tag{8}$$

Here, we note that MPNN with normalized sum aggregation are only appropriate for dense graphs. Indeed, if the graph is sparse, then $|V| \gg |\mathcal{N}(v)|$, and the aggregation will always give a close-to-zero outcome. Hence, such an MPNN will

always give roughly the same output for all sparse graphs. Moreover, sum aggregation is only asymptotically appropriate for sparse graphs, otherwise, the aggregation diverges to infinity as the size of the graph increases.

By defining a new metric, the *graphon-signal cut distance*, which extends the cut distance, the space of graph signals is shown to be dense in the space of graphon signal, which itself is shown to be a compact metric space. The analysis reveals that MPNNs are Lipschitz continuous functions over the graphon-signal metric space, leading to a generalization bound for MPNNs, indicating that a well performing MPNN on training graph-signals is likely to perform similarly on test graph-signals.

Rauchwerger and Levie (Rauchwerger & Levie, 2025) extend these existing results to multidimensional signals and obtain improved generalization bound. In addition, they remove the symmetry assumption on graphons, thereby enabling the analysis for both directed and undirected graphs.

The analysis of the expressive power of MPNNs is mostly studied using the 1-WL test, which addresses the graph isomorphism problem. However, the graph isomorphism problem is binary, focusing on the question whether two graphs are isomorphic or not, not giving insight to the degree of similarity between the graphs. (Böker et al., 2024) addresses this issue, extending both MPNNs and the 1-WL test to graphons, and shows that the continuous variant of the 1-WL test provides a topological characterization of the expressive power of MPNNs on graphons. The paper establishes the connection between the similarity of graphs and their output via MPNNs. In addition, a universal approximation theorem is proved, stating that functions on graphons that are invariant with respect to the 1-WL test, can be approximated by MPNNs.

A recent extension was presented in (Rauchwerger et al., 2025), which generalizes the analysis of graphs with no node features presented in (Böker et al., 2024), to graph-signals, as suggested in section 5 of (Böker et al., 2024). Rauchwerger et al. introduce a new metric on graph-signals, called the DIDMs-mover's distance, and establish a universal approximation theorem for MPNN defined on this space. For a more extensive background, we refer the reader to Appendix E .

Recent work has also explored higher-order graphon neural networks and their relation to approximation theory and cut-distance continuity (Herbst & Jegelka, 2025).

### D.3. Sparse Graph Limit Theories

While graphons have been proven useful in the analysis of graph limit theory, they suffer from a crucial limitation when sparse graph limits are considered. Due to the relation between the cut-norm and the $\mathcal{L}^1$ norm, graph sequences with a subquadratic number of edges converge to the zero graphon in the cut-distance. Several approaches have been proposed to overcome this limitation. One such approach is based on stretched graphons, together with an appropriately rescaled metric known as the stretched cut-distance (Borgs et al., 2018a; Jian et al., 2023; Ji et al., 2024). By rescaling the underlying domain, stretched graphons essentially "stretch" the graph, yielding nontrivial limit objects for sparse graph sequences. Another similar approach is to relax the boundedness assumption on graphons by considering $\mathcal{L}^p$ graphons, which generalize standard graphons from bounded measurable function to measurable function with finite $\mathcal{L}^p$ norm.

Another approach on bounded-degree graph sequences and their corresponding limit objects, known as graphings (Lovász, 2012).

A recent approach introduced in (Backhausz & Szegedy, 2022) unifies and generalizes both dense and sparse graph limits within a single framework by representing graph limits as linear operators, referred to as graphops. Backhausz and Szegedy introduce a metric under which the space of graphops is compact. Recent work has also used graphop analysis to study approximation and transferability properties of GNNs on sparse graphs (Le & Jegelka, 2023). Graphops theory forms the main foundation of this paper. For an extensive background and further extensions, we refer the reader to appendix H.1.

## E. Analysis of GNNs via Iterated Degree Measures

Iterated Degree Measures, IDMs, are used as the building blocks of a measure-theoretic analogue of the classic 1-Weisfeiler-Leman (1-WL) algorithm. The 1-Weisfeiler-Leman algorithm was originally introduced as a fingerprinting technique for chemical molecules: for the history and general background regarding 1-WL, see (Grohe, 2020) and references therein. In (Grebík & Rocha, 2022) the authors define IDMs to generalize the 1-WL and its properties to graphons. In their generalization IDMs represent the colors in this extension of 1-WL. For each graphon, the 1-WL algorithm gives a distribution over the set of IDMs. The space of all such distributions, namely *computation DIDM* (where DIDM is short for Distribution of Iterated Degree Measures), is strictly smaller than the ambient space of *DIDMs* which is defined more generally.

The background setting for the formulation of IDMs in the context of graphop-signal, which is the object we consider in this work, is the work (Rauchwerger et al., 2025) where the authors expand the definitions of IDMs and computation IDMs from unattributed graphons (Grebík & Rocha, 2022; Böker et al., 2024) to attributed graphons. This idea is also in line with an idea outlined in Section 5 of (Böker et al., 2024). In (Rauchwerger et al., 2025), the authors develop a version of the 1-WL test for graphon-signals, extending the existing formulation for graphons.

### E.1. Iterative Degree Measures

We follow the construction described in (Rauchwerger et al., 2025) with slight modifications that are more appropriate for our setting. These changes do not affect the analysis. The IDMs' base space is taken to be the space of node attributes $[-1, 1]^d \subset \mathbb{R}^d$. We define the *space of iterated degree measures (IDMs)* of order-L, $\mathcal{H}^L$, inductively by first defining $\mathcal{H}^0 = \mathcal{M}^0 := [-1, 1]^d$. Then, for every $L > 0$, we let $\mathcal{H}^L = \prod_{i \leq L} \mathcal{M}^i$ and $\mathcal{M}^{L+1} = \mathcal{M}_{\leq r}(\mathcal{H}^L)$, where $\mathcal{M}_{\leq r}(\cdot)$ denotes the space of all nonnegative Borel measures with total mass at most $r$, and the topologies of $\mathcal{H}^L$ and $\mathcal{M}^{L+1}$ are the product and weak* topologies respectively; see Appendices A.1.2, A.2.3, and B.1 for details regarding these topologies. It can be shown that these spaces are standard Borel spaces, and that the weak* topology is metrizable, and, in fact, compactly metrizable. We elaborate on this later in this section. In this construction, each color at any layer is given by the histogram of colors from the previous layer, concatenated with the colors from all previous layers. Let $\mathcal{P}(\mathcal{H}^L)$ be *the space of distributions of iterated degree measures (DIDMs)* of order-L. We consider probability distributions of IDMs, as the final step of 1-WL involves considering the histogram of colors in the nodes of the graph. The 1-WL test distinguishes non-isomorphic graphs by comparing their color distributions, where different distributions imply non-isomorphic graphs.

### E.2. 1-WL Test for Graphon-Signals

Denote by $p_{L,j} : \mathcal{H}^L \mapsto \mathcal{H}^j$ and $p_L : \mathcal{H}^L \to \mathcal{M}^L$ the projections where $j \leq L < \infty$. Namely, for $u \in \mathcal{H}^L$, we have that $p_{L,j}(u)$ is a vector consisting of the first $j + 1$ coordinates of $u$, and $p_L(u)$ is the last coordinate of $u$. We denote by $\alpha(x)(j)$ the j-th entry of the vector $\alpha(x)$. The paper (Rauchwerger et al., 2025) defines a graphon-signal 1-WL analog as follows (see Def. D.3 of a graphon-signal).

**Definition E.1.** Let $[0, 1]$ be the interval with the standard Borel $\sigma$-algebra $\mathcal{B}$ and let $(W, f)$ be a graphon-signal. Define $\gamma_{(W,f),0} : [0, 1] \to \mathcal{H}^0$ to be the map $\gamma_{(W,f),0}(x) := f(x)$ for every $x \in [0, 1]$. Inductively, define $\gamma_{(W,f),L+1} : [0, 1] \to \mathcal{H}^{L+1}$ by

1. $\gamma_{(W,f),L+1}(x)(j) = \gamma_{(W,f),L}(x)(j)$, for every $j \leq L$ and

2. $\gamma_{(W,f),L+1}(x)(L+1)(E) = \int_{\gamma^{-1}_{(W,f),L}(E)} W(x,-)d\lambda$, whenever $E \subset \mathcal{H}^L$ is a Borel set.

For every $L \in \mathbb{N}_0$, let $\Gamma_{(W,f),L}$ be the pushforward of $\lambda$ via $\gamma_{(W,f),L}$. Here, $\gamma_{(W,f),L}$ is called a *computation iterated degree measure (computation IDM)* of order-L and $\Gamma_{(W,f),L}$ a *distribution of computation iterated degree measure (computation DIDM)* of order-L.

Note that by the boundedness of graphons, each measure that is defined by 2, is actually in $\mathcal{M}_{\leq 1}$.

The pushforward of $\lambda$ via $\gamma_{(W,f),L}$ can be viewed as the color histogram across the whole graphon (see Appendix A.2.5 for more details on the definition of pushforward). The extension of 1-WL isomorphism test to graphons is defined as the comparison of the two graphon-DIDMs of two given graphon-signals. If their graphon-DIDMs are not identical, the two graphon-signals are deemed non-isomorphic, otherwise it is undetermined.

In this work, we extend Definition E.1 to a wider class of graph limit objects, called bofops (see Section 5.1 and Appendix K). Hence, to distinguish between the above definition and our more general one, we call computation DIDM as defined in Definition E.1 *graphon-DIDMs*. Moreover, we refer to graphon-DIDMs that were obtained from induced graphon-signals as *dense graph-DIDMs*. Recall Definition D.4 for induced graphon-signal. Informally, note that for sparse graphs, the number of vertices in a typical neighborhood is much smaller than the number of vertices of the graph, and hence, in the context of IDMs, the measure of a neighborhood is "small". Therefore, the claim that graphons are appropriate primarily for dense graphs is true for DIDMs as well, just as it does in aggregation-based considerations (see Section 2.2). This observation justifies the terminology dense graph-DIDM.

We adapt topology on the space of graphon-signals, metrized by a distance between their associated DIDMs (Rauchwerger

et al., 2025). This topology is shown to be coarser than the cut distance (Theorem 63 in (Rauchwerger et al., 2025)), which is well known to be suited mainly for dense graphs, since sparse graphs are considered similar to the zero-graphon.

Let us summarize this for later referencing.

**Definition E.2.** A computation-DIDM computed from a graphon-signal is called a *graphon-DIDM*. A computation-DIDM computed from a graphon-signal induced by a graph-signal is called a *dense graph-DIDM*. In a similar manner, we call *graph-IDM*, an IDM that is computed for a graphon induced by a graph-signal, and call *graphon-IDM* an IDM that is computed for a general graphon.

### E.3. DIDM Mover's Distance

In the construction made in (Rauchwerger et al., 2025) the space of DIDMs is given a metric called the *DIDM Mover's Distance*. We follow this construction here. This metric explicitly metrizes the topology of $\mathcal{H}^L$ which is defined above. Before we define this metric, we define *The Unbalanced Earth Mover's Distance* (Séjourné et al., 2023) which will be used in the definition of DIDM Mover's Distance. The definition uses the notion of couplings for unbalanced measures (the notion was defined for probability measure in A.21): a *coupling* $\gamma$ between two measures $\mu, \nu$ on measure spaces $\mathcal{X}, \mathcal{Y}$ respectively is a joint measure on $\mathcal{X} \times \mathcal{Y}$ such that $(p_{\mathcal{X}})_* \gamma = \mu, (p_{\mathcal{Y}})_* \gamma \leq \nu$, given that $\|\mu\| \leq \|\nu\|$, where $\|\mu\| = \mu(\mathcal{X})$ and $\|\nu\| = \nu(\mathcal{Y})$. Here $p_{\mathcal{X}}, p_{\mathcal{Y}}$ are projections from $\mathcal{X} \times \mathcal{Y}$ to $\mathcal{X}$ and $\mathcal{Y}$ respectively (see A.2.5 for the definition of pushforward measures).

**Definition E.3** (The Unbalanced Earth Mover's Distance). Let $(\mathcal{X}, d)$ be a Polish space. The *Unbalanced Earth Mover's Distance* between two measures $\mu, \nu \in \mathcal{M}_{\leq r}(\mathcal{X})$ is

$$\mathbf{OT}_d(\mu, \nu) = \inf_{\gamma \in (\mu, \nu)} \left( \int_{\mathcal{X} \times \mathcal{X}} d(x, y) d\gamma(x, y) \right) + |\|\mu\| - \|\nu\||$$

where $\Gamma(\mu, \nu)$ is the set of all couplings of $\mu$ and $\nu$, and $\|\mu\|, \|\nu\|$ denote the "mass" of $\mu$ and $\nu$ respectively. Namely, $\|\mu\| = \mu(\mathcal{X})$ and $\|\nu\| = \nu(\mathcal{X})$.

The DIDM Mover's Distance is then defined as follows. First, define the *IDM distance* of order-0 on $\mathcal{M}^0 = \mathcal{H}^0 = [-1, 1]^d$ for $d \in \mathbb{N}_0$ by $d^0_{\text{IDM}}(x, y) := \|x - y\|_2$ for $x, y \in [-1, 1]^d$, and denote by $\mathbf{OT}_{d^0_{\text{IDM}}}$ the optimal transport distance on $\mathcal{M}^1 = \mathcal{M}_{\leq r}(\mathcal{H}^0)$. The distance $d_{\text{IDM}}^L$ is defined recursively on $\mathcal{H}^L = \prod_{j \leq L} \mathcal{M}^j$ where the distance on $\mathcal{M}^j = \mathcal{M}_{\leq r}(\mathcal{H}^{j-1})$ for $0 < j < L - 1$ is $\mathbf{OT}_{d^{j-1}_{\text{IDM}}}$. Explicitly,

$$d_{\text{IDM}}^L(\mu, \nu) := \begin{cases} \|\mu_0 - \nu_0\|_2 + \sum_{j=1}^L \mathbf{OT}_{d^{j-1}_{\text{IDM}}}(\mu_j, \nu_j) & , \quad \text{if } 0 < L < \infty \\ \|\mu_0 - \nu_0\|_2 & , \quad \text{if } L = 0 \end{cases}$$

for $\mu = (\mu_j)_{j=0}^L \in \mathcal{H}^L$ and $\nu = (\nu_k)_{k=0}^L \in \mathcal{H}^L$. Note that $\mu_0, \nu_0 \in [-1, 1]^d$ are points in a Euclidean space and not measures. Finally, for two DIDMs $\mu^{(L)}, \nu^{(L)} \in \mathcal{P}(\mathcal{H}^L)$ we consider the distance $\mathbf{OT}_{d_{\text{IDM}}^L}$.

We can now define the DIDM Mover's Distance. This is a distance between graphon-signals, when represented as DIDMs via 1-WL. It is the optimal transport cost between the DIDMs of the two graphon-signals (two measures on the space $\mathcal{H}^L$) with the ground metric $d_{\text{IDM}}^L$.

**Definition E.4** (DIDM Mover's Distance). Given two graphon-signals $(W_1, f_1), (W_2, f_2)$ and $L \geq 1$, the *DIDM Mover's Distance* between $(W_1, f_1)$ and $(W_2, f_2)$ is defined as

$$\delta_{\text{DIDM}}^L((W_1, f_1), (W_2, f_2)) := \mathbf{OT}_{d_{\text{IDM}}^L}(\Gamma_{(W_1, f_1)}, \Gamma_{(W_2, f_2)}).$$

Recall that $\Gamma_{(W, f)}$ was defined as the pushforward of the Lebesgue measure $\lambda$ on $[0, 1]$ via the map $\gamma_{(W, f), L}$. Therefore, intuitively, $\delta_{\text{IDM}}^L(\cdot, \cdot)$ is the minimum cost to transport node-wise IDMs from one graphon to another.

See (Rauchwerger et al., 2025), Theorem. 4, for a proof of the validity of the definitions of the two metrics.

### E.4. Compactness of the space of DIDMs and Computation DIDMs

Both spaces $\mathcal{H}^L$ (the space of IDMs), and $\mathcal{P}(\mathcal{H}^L)$ (the space of DIDMs) are compact. The topologies of the spaces are the product topology and the weak* topology respectively. These topologies are also the topologies induced by the metrics $d_{\text{IDM}}^L$ and $\delta_{\text{DIDM}}^L$ respectively (see (Rauchwerger et al., 2025), Theorem 4).

**Theorem E.5.** *The spaces $\mathcal{H}^L$ and $\mathscr{P}(\mathcal{H}^L)$ are compact spaces for any $L \in \mathbb{N}_0$.*

It is shown in Theorem 64 in (Rauchwerger et al., 2025) that the space of graphon-signal $\mathcal{WL}_r^d$ equipped with the pseudometric $\delta_{\mathrm{DIDM}}^L$ is compact. Similarly to our discussion in Appendix D.1 with the cut-distance, by passing to a quotient space $\mathcal{WL}_r^d / \delta_{\mathrm{DIDM}}^L$ with respect to the relation that identifies graphon-signals with zero distance, the space $(\mathcal{WL}_r^d / \delta_{\mathrm{DIDM}}^L, \delta_{\mathrm{DIDM}}^L)$ becomes a metric space. Recall Appendix A.1.7 for quotient spaces.

**Theorem E.6.** *The pseudometric space $(\mathcal{WL}_r^d, \delta_{DIDM}^L)$ and the metric space $(\mathcal{WL}_r^d / \delta_{DIDM}^L, \delta_{DIDM}^L)$ are compact.*

### E.5. Hierarchy of DIDM Spaces: Graphon-DIDMs Are a Strict Subset of General DIDMs

Combining the compactness properties of the spaces of DIDMs and graphon-DIDMs yields the following dichotomy: either every DIDM arises from graphon-signal through the 1-WL algorithm, or that the space of graphon-DIDMs is a strict subset of the space of all DIDMs. The following example shows that the latter holds; namely, we construct a DIDM that cannot arise from any graphon-signal via 1-WL. We use this terminology: we say that a measure $\mu$ is supported on $A$ if $\mu(B) = \mu(A \cap B)$ for every measurable set $B$.

*Example* E.7. Consider the set of features $[-1, 1]$ (recall that the set of features, or node attributes, is defined to be $[-r, r]^d$, and set $r = 1$ and $d = 1$ in this example). Set $L = 1$. We define an order-1 DIDM and show that it cannot be obtained by applying the 1-WL to any graphon-signal. Take a probability measure $\alpha \in \mathscr{P}(\mathcal{H}^1)$ to be the pushforward of the Lebesgue measure on $[0, 1]$ via the mapping $[0, 1] \ni x \mapsto \left(1, \lambda|_{[0, 1 - \frac{x}{3} - \frac{1}{3}]}\right) \in \mathcal{H}^1$ , where $\lambda|_{[0, a]}(A) = \lambda(A \cap [0, a])$ for a Borel set $A \subset [-1, 1]$ (Note that the restriction of a measure is a measure, see (Tao, 2011)). Observe that this map is a measurable map with respect to the Borel $\sigma$-algebra on $\mathcal{H}^1$, therefore we have that $\alpha$ is a DIDM (see Appendix A.2.5 for more details on the definition of pushforward). Now let $(W, f)$ be a graphon, and suppose that $\Gamma_{(W,f),1} = \alpha$. If $W(x, y) = 0$ for a.e. $(x, y) \in [0, 1]$, then $\gamma_{(W,f),1}(x)(1)$ is the trivial measure (assigning measure zero to every set) for a.e. $x \in [0, 1]$, so the marginal of $\Gamma_{(W,f),1}$ on the second axis is a delta distribution on the trivial measure. Since the marginal of $\alpha$ on the second axis is not a delta on the trivial measure, this is a contradiction. Hence, $W(x, y) \neq 0$ on a set of positive measure. By the base of the 1-WL algorithm (Definition E.1), $\gamma_{(W,f),1}(x)(0) = \gamma_{(W,f),0}(x) = f(x)$ for a.e. $x \in [0, 1]$. Since the marginal distribution of $\alpha$ on the first axis is supported in $\{1\}$, which must be equal to the pushforward of $\lambda$ under $x \mapsto \gamma_{(W,f),1}(0) = f(x)$, we conclude that $f(x) = 1$ for a.e. $x \in [0, 1]$. Then for every $x \in [0, 1]$, such that $f(x) = 1$ by the first step of the 1-WL algorithm on $(W, f)$, by step 2 in Definition E.1, for every set $B \subset [-1, 1] = \mathcal{H}^0$ such that $B \cap \{1\} = \emptyset$, we have that $\gamma_{(W,f),1}(x)(1)(B) = 0$. This is because since $\gamma_{(W,f),0}^{-1}(B) = \emptyset$ when $B$ does not contain 1:

$$\gamma_{(W,f),1}(x)(1)(B) = \int_{\gamma^{-1}_{(W,f),0}(B)} W(x, -) d\lambda = 0.$$

Therefore $\gamma_{(W,f),1}(x)(1)$ is a measure supported on $\{1\}$. Hence, $\Gamma_{(W,f),1}$, the pushforward of $\lambda$ under $\gamma_{(W,f),1}$, has a marginal on the second axis (an element of $\mathscr{P}(\mathcal{M}^1)$) which is supported on the measures of $\mathcal{M}_1 = \mathscr{M}_{\leq 1}([-1, 1])$, which are supported on $\{1\}$. This is a contradiction to the definition of $\alpha$, which has a marginal on the second axis that is not supported on the measures of $\mathscr{M}_{\leq 1}([-1, 1])$ which are supported on $\{1\}$. Hence, $\alpha$ is a DIDM that cannot be a graphon-DIDM.

We therefore stress that the space of DIDMs is strictly larger than that of graphon-DIDMs, in the sense that the closure of graphon-DIDMs is not all of DIDMs. It is also shown that the space of computation DIDMs of graph-signals is dense in the space of graphon-DIDMs, see Appendix I.3 in (Rauchwerger et al., 2025).

### E.6. MPNNs on graph-signals, graphon-signal, and DIDMs

Recall that in Appendix D.2 we presented an extension of MPNN which processes graph-signals to process graphon-signals in such a way that for any graph-signal, applying the MPNN to the graph yields exactly the same output as applying the MPNN to the corresponding induced graphon-signal.

Böker et. al. (Böker et al., 2024) integrate MPNNs into the framework of IDMs, and extend the definition of MPNN to process IDMs in such a way the application of MPNN to a graph (or to its induced graphon) yields exactly the same output as applying the MPNN on the IDM obtained from running the 1-WL algorithm on the graph (Appendix C1 in (Böker et al., 2024)). Building on that (Rauchwerger et al., 2025) extend the result from MPNNs that process graphs to graph-signals.

Recall that $p_{L,j} : \mathcal{H}^L \mapsto \mathcal{H}^j$ and $p_L : \mathcal{H}^L \to \mathcal{M}^L$ are the projections where $j \leq L < \infty$. Namely, for $u \in \mathcal{H}^L$, we have that $p_{L,j}(u)$ is a vector consisting of the first $j + 1$ coordinates of $u$, and $p_L(u)$ is the last coordinate of $u$.

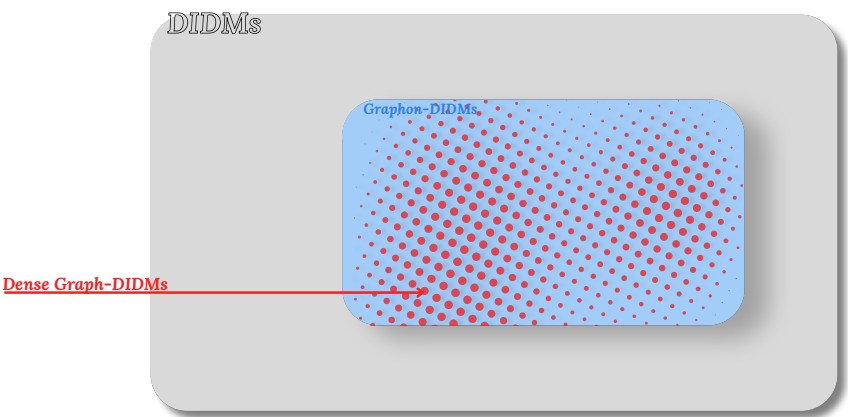

*Figure 2.* Hierarchy of DIDM spaces: the space of all DIDMs (black); its strict subset of graphon-DIDMs, isomorphic to graphon-signals (blue); and the space of dense-DIDMs, isomorphic to graph-signals, shown as a dense subset (red dots).

**Definition E.8** (MPNNs on IDMs and DIDMs). Let $(\varphi, \psi)$ be an MPNN model with readout. The application of the MPNN on IDMs and DIDMs is defined as follows: initialize $\mathfrak{f}_{-}^{(0)} := \varphi^{(0)}(-)$, and compute the hidden IDM node representations $\mathfrak{f}_{-}^{(t)} : \mathcal{H}^t \to \mathbb{R}^{d_t}$ on any order-$t$ IDM $\tau \in \mathcal{H}^t$, and the DIDM-level output $\mathfrak{F} \in \mathbb{R}^d$ on an L-order DIDM $\nu \in \mathscr{P}(\mathcal{H}^L)$, by

$$\mathfrak{f}_{\tau}^{(t)} := \varphi^{(t)} \left( \mathfrak{f}_{p_{t,t-1}(\tau)}^{(t-1)}, \int_{\mathcal{H}^{t-1}} \mathfrak{f}_{-}^{(t-1)} \, dp_t(\tau) \right) \quad \text{and} \quad \mathfrak{F} := \psi \left( \int_{\mathcal{H}^L} \mathfrak{f}_{-}^{(L)} d\nu \right).$$

We also denote $\mathfrak{f}(\varphi)_{\tau}^{(t)}$ or $\mathfrak{f}(\varphi, \tau)^{(t)}$, and $\mathfrak{F}(\varphi, \psi)$ or $\mathfrak{F}(\varphi, \psi, \nu)$.

As discussed above, Lemma 11 in (Rauchwerger et al., 2025) shows the commutation between the application of MPNN to graphon-signals (Definition D.5) and to IDMs (Definition E.8 ).

**Lemma E.9.** *Let $(W, f)$ be a graphon-signal and $(\varphi, \psi)$ and L-layer MPNN model with readout. Then, given the corresponding grpahon-IDMs $\{\gamma_{(W,f),t}\}_{t=0}^L$ and graphon-DIDM $\Gamma_{(W,f),L}$, we have that $\mathfrak{w}(\varphi, W, f)_x^{(t)} = \mathfrak{f}(\varphi)_{\gamma_{(W,f),t}(x)}^{(t)}$ for any $t \in [L]$ and $x \in [0, 1]$. Similarly, $\mathfrak{W}(\varphi, \psi, W, f) = \mathfrak{F}(\varphi, \psi, \Gamma_{(W,f),L})$.*

We summarize this section with the following diagram that shows the commutation relation between MPNN on graph-signals, induced graphon-signals and the corresponding IDMs and DIDMs. This can be vied as combination of Lemma E.9 and Equation (8).

$$
\begin{array}{ccc}
(G, f) & \xrightarrow{\varphi} & \mathfrak{g}^{(L)} \\
\downarrow & & \| \; \text{Eq. (8)} \\
(W_G, f_f) & \xrightarrow{\varphi} & \mathfrak{w}^{(L)} \\
\text{1-WL} \downarrow & & \| \; \text{Lem. E.9} \\
\Gamma_{(W_G, f_f), L} & \xrightarrow{\varphi} & \mathfrak{f}^{(L)}
\end{array}
\Bigg\} \xrightarrow{\text{Readout } \psi} \mathbb{R}^p
$$

### E.7. Continuity of MPNNs With Respect To DIDM Mover's Distance

The MPNN model on IDMs and IDMs defined in E.8 is such, that given an MPNN model, it induces a function from the space of DIDMs to vectors. This function is shown to be Lipschitz continuous. This formulation allows to state and prove the following theorem which is crucial for the generalization analysis in (Rauchwerger et al., 2025).

**Theorem E.10.** *Let $\varphi$ be an L-layer MPNN model. Then, there exists a constant $C_{\varphi}$, that depends only on L, the number of*

*layers and the Lipschitz constants of the model's update functions, such that*

$$\|\mathfrak{f}(\varphi, \alpha)^{(L)} - \mathfrak{f}(\varphi, \beta)^{(L)}\|_2 \le C_\varphi \cdot d_{\mathrm{IDM}}^L(\alpha, \beta)$$

*for all $\alpha, \beta \in \mathcal{H}^L$. If $\varphi$ has a readout function $\psi$, then for all $\mu, \nu \in \mathscr{P}(\mathcal{H}^L)$ there exists a constant $C_{(\varphi,\psi)}$ that depends only on $C_\varphi$ and the Lipschitz constant of the model's readout function, such that*

$$\|\mathfrak{F}(\varphi, \psi, \mu) - \mathfrak{F}(\varphi, \psi, \nu)\|_2 \le C_{(\varphi,\psi)} \cdot \boldsymbol{OT}_{d_{IDM}^L}(\mu, \nu).$$

As a corollary MPNNs are continuous over graphon-signals with respect to DIDM Mover's Distance.

## E.8. Expressivity of MPNNs via DIDM Mover's Topology

In (Rauchwerger et al., 2025) it was shown that the expressivity of MPNNs can be characterized via the DIDM Mover's Distance. Namely, convergence of a sequence of DIDMs is equivalent to convergence of their output with respect to any MPNN. This is the content of the following theorem,

**Theorem E.11** ((Rauchwerger et al., 2025), Corollary 12). *Let $L \in \mathbb{N}_0$ and $d > 0$ be fixed. Let $\nu \in \mathscr{P}(\mathcal{H}^L)$ and $(\nu_i)_i$ be a sequence with $\nu_i \in \mathscr{P}(\mathcal{H}^L)$. Then, $\nu_i \to \nu$ if and only if $\mathfrak{F}(\varphi, \psi, \nu_i) \to \mathfrak{F}(\varphi, \psi, \nu)$ for all L-layer MPNN models $\varphi$ with a readout function $\psi : \mathbb{R}^{d_L} \to \mathbb{R}^d$.*

We can interpret this important theorem in the following way: two DIDMs are close to each other if and only if the outputs of any MPNN on them are close to each other. Hence, MPNNs have the same separation power as the DIDM Mover's Distance.

We saw in Lemma E.9 that MPNNs on IDMs and DIDMs extend canonically MPNNs on graphon-signals. We therefore have that convergence of graphon-signals is equivalent to convergence of all MPNN outputs.

## E.9. Universal Approximation Theorems based on DIDM Mover's Topology

The notion of universal approximation was introduced in C.6. In this section, we present the results concerning universal approximation from (Rauchwerger et al., 2025). Following (Rauchwerger et al., 2025), we define the set $\mathcal{N}_L^{d_L} := \{\mathfrak{f}(\varphi)_-^{(L)} : \mathcal{H}^L \mapsto \mathbb{R}^{d_L} | \varphi$ is an L-layer MPNN model$\} \subseteq C(\mathcal{H}^L, \mathbb{R}^{d_L})$, where $C(\mathcal{H}^L, \mathbb{R}^{d_L})$ is the set of all continuous functions $\mathcal{H}^L \mapsto \mathbb{R}^{d_L}$ w.r.t $d_{\mathrm{IDM}}^L$ and the Euclidean metrics. Similarly we define the set $\mathcal{N}\mathcal{N}_L^d := \{\mathfrak{F}(\varphi, \psi, -) : \mathscr{P}(\mathcal{H}^L) \mapsto \mathbb{B}_r^d(\varphi, \psi) | (\varphi, \psi)$ is an L-layer MPNN model with readout$\} \subseteq C(\mathscr{P}(\mathcal{H}^L), \mathbb{R}^d)$, where one considers the metric $\boldsymbol{OT}_{d_{\mathrm{IDM}}^L}$ on the domain.

The following theorem states that any continuous function from DIDMs to scalars can be approximated by an MPNN on DIDMs.

**Theorem E.12.** *Let $L \in \mathbb{N}_0$. Then, the set $\mathcal{N}_L^1$ is uniformly dense in $C(\mathcal{H}^L), \mathbb{R})$ and the set $\mathcal{N}\mathcal{N}_L^1$ is uniformly dense in $C(\mathscr{P}(\mathcal{H}^L), \mathbb{R})$.*

Define the set $\mathcal{N}\mathcal{N}_L^d(\mathcal{W}\mathcal{L}_r^d) := \{\mathfrak{F}(\varphi, \psi, -, -) : \mathcal{W}\mathcal{L}_r^d \to \mathbb{R}^d | (\varphi, \psi)$ is an L-layer MPNN model with readout$\}$. As argued before, the space of graphon-DIDMs, $\Gamma_{(W,f),L}(\mathcal{W}\mathcal{L}_r^d)$, is not dense in $\mathscr{P}(\mathcal{H}^L)$ since it is a strict compact subset of it. In (Rauchwerger et al., 2025), the authors prove directly a universal approximation theorem for functions over the space $\mathcal{W}\mathcal{L}_r^d$, continuous with respect to DIDM-mover's distance. Denote by $C(\mathcal{W}\mathcal{L}_r^d, \mathbb{R})$ the space of functions from the graphon-signals to the real numbers, continuous with respect to DIDM-mover's distance.

**Theorem E.13.** *Let $L \in \mathbb{N}_0$. Then, the set $\mathcal{N}\mathcal{N}_L^1(\mathcal{W}\mathcal{L}_r^d)$ is uniformly dense in $C(\mathcal{W}\mathcal{L}_r^d, \mathbb{R})$.*

## *Results*

The following appendices F through N contain our contributions, extending the results from main paper, where we present full formulations and detailed proofs.

- In appendix F, we prove that the standard MPNN architecture, which applies explicit message function to the graph-signals, can be implemented within our simplified model using twice the number of layers and without explicit message functions.

- Appendix G is devoted to the necessary background and supportive lemmas from optimal transport, which are used to prove topological properties of the spaces of graphop-signals and profiled, as well as to prove the continuity of MPNNs with respect to the induced topology.

- An extension to graphop-theory, originally introduced in (Backhausz & Szegedy, 2022), to the setting of graphop-signals is presented in Appendix H.1. This appendix introduces the relevant definitions and establishes the compactness properties of the corresponding spaces, which constitute the main objects of study in this paper.

- In appendix I we apply MPNN models on both P-operator-signals and their profiles, and prove what we call the *commutation property*, which states that applying an MPNN layer to a P-operator-signal and then taking its profile yields the same result as applying the corresponding MPNN layer directly to the profile.

- Hölder continuity of MPNN on P-operator-signals and profiles is proven in Appendix J using results developed earlier in Appendices G and H.1.

- An extension to the analysis of GNNs via iterated degree measures that was discussed in Appendix E to the framework of bofop-signals is presented in Appendices K and M. These sections establish various topological properties of the corresponding IDM spaces, together with a comprehensive discussion of the hierarchy of DIDM spaces.

- Finally, our main results of the paper-namely, a universal approximation theorem and generalization bounds of MPNNs for sparse graphs, are proved in Appendix N.

## F. Equivalence of MPNN Architectures

In this section, we discuss the expressive equivalence between the general *Alternative MPNN* model-often referred to in the literature as message passing neural networks (Gilmer et al., 2017)) which passes the node's labels through message functions (Levie, 2024), and a simplified architecture that only aggregates node features without message functions, which we refer to through the paper as MPNN (often known as "Graph Isomorphism Network" (Xu et al., 2019)). We show that an Alternative MPNN layer with a non-linear message function can be implemented using two layers of MPNN where we consider a general aggregation function (see C.2).

**Definition F.1** (Alternative MPNN Model). Let $L \in \mathbb{N}_0$ and
$d, d_0, \ldots, d_L, p_0, \ldots, p_{L-1}, p \in \mathbb{N}_0$. We call the tuple $(\varphi, \phi)$ such that $\varphi$ is any collection $\varphi = (\varphi^{(t)})_{t=0}^{L}$ of Lipschitz continuous functions $\varphi^{(0)} : \mathbb{R}^d \to \mathbb{R}^{d_0}$ and $\varphi^{(t)} : \mathbb{R}^{d_{t-1} \times p_{t-1}} \to \mathbb{R}^{d_t}$, for $1 \leq t \leq L$, and $\phi$ is any collection $\phi = (\phi^{(t)})_{t=1}^{L}$ of (Lipschitz continuous) message functions $\phi^{(t)} : \mathbb{R}^{2d_{t-1}} \to \mathbb{R}^{p_{t-1}}$, for $1 \leq t \leq L$, an $L$-layer alternative MPNN model, and call $\varphi^{(t)}$ update functions. For Lipschitz continuous $\psi : \mathbb{R}^{d_L} \to \mathbb{R}^p$, we call the tuple $(\varphi, \phi, \psi)$ an alternative MPNN model with readout, where $\psi$ is called a readout function. We call $L$ the depth of the MPNN, $d$ the input feature dimension, $d_0, \ldots, d_L$ the hidden node feature dimensions, $p_0, \ldots, p_{L-1}$ the hidden message dimensions, and $p$ the output feature dimension.

### F.1. Architectural Definitions

We distinguish between two variants of the message passing architecture.

**Alternative MPNN**: This architecture includes explicit message functions. The aggregation depends on a learnable message function $\phi$. For simplicity, we consider the case where the message depends only on the source node features, i.e., $\phi(f_u^{(l)})$. The update rule is given by Definition F.1:

$$f_v^{(l+1)} = \varphi\left(f_v^{(l)}, \mathbf{A}(\phi \circ f_-^{(l)})(v)\right). \tag{9}$$

**MPNN**: This is the simplified architecture without explicit message functions (Definition C.1) which we consider through the paper. Aggregation is performed directly on the hidden features, and learnability is restricted to the update function $\varphi$. An MPNN layer update is given by

$$f_v^{(l+1)} = \varphi\left(f_v^{(l)}, \mathbf{A}f_-^{(l)}(v)\right). \tag{10}$$

### F.2. Proof of Equivalence

We show that a single layer of the Alternative MPNN (9) can be implemented exactly by two layers of the simplified MPNN (10).

Layer 1: Define the first MPNN layer to use an update function $\eta_1$ that ignores the aggregated neighbor information and augments the node features with the message transformation:

$$f_v^{(l+\frac{1}{2})} = \eta_1\left(f_v^{(l)}, \mathbf{A}f_-^{(l)}(v)\right) := \left(f_v^{(l)}, \phi(f_v^{(l)})\right).$$

Thus, the intermediate feature $f_v^{(l+\frac{1}{2})}$ concatenates the original feature with its message representation.

Layer 2: The second MPNN layer aggregates the augmented features coordinate wise:

$$\mathbf{A}f_-^{(l+\frac{1}{2})}(v) = \left(\mathbf{A}f_-^{(l)}(v),\ \mathbf{A}(\phi \circ f_-^{(l)})(v)\right).$$

Define the update function $\eta_2$ by

$$\eta_2\left(f_v^{(l+\frac{1}{2})}, \mathbf{A}f_-^{(l+\frac{1}{2})}(v)\right) := \varphi\left(p_1\left(f_v^{(l+\frac{1}{2})}\right),\ p_2\left(\mathbf{A}f_-^{(l+\frac{1}{2})}(v)\right)\right),$$

where $p_1$ and $p_2$ denote projection onto the first and second components, respectively. Substituting the definitions yields

$$f_v^{(l+1)} = \varphi\left(f_v^{(l)}, \mathbf{A}(\phi \circ f_-^{(l)})(v)\right),$$

which recovers (9) exactly.

## G. Background and Supportive Lemmas from Optimal Transport and Wasserstein Distance

Given two probability distributions $\mu$ and $\nu$, the optimal transport problem (or Monge Kantorovich problem) formalizes the question "what is the minimal cost of transporting the mass described by $\mu$ to the mass described by $\nu$?" This is often illustrated by the minimal cost of moving a pile of sand to fill a hole, where the goal is to find the most efficient way to rearrange the sand. In many settings however, one may compare measures of unequal total mass, leading to the unbalanced optimal transport, which extends the classic theory (Séjourné et al., 2023).

### G.1. The Wasserstein Distance

The Wasserstein distance arises as the minimal value of the optimal transport problem when both measures have equal total mass.

**Definition G.1.** Let $(\mathcal{X}, d)$ be a complete and separable metric space, and $\mathcal{P}(\mathcal{X})$ be the set of all Borel probability measures over $\mathcal{X}$. For $p \in [1, \infty)$ and any two Borel probability measures $\mu, \nu \in \mathcal{P}(\mathcal{X})$, we define the *Wasserstein distance* of order $p$ between $\mu$ and $\nu$ by

$$W_p(\mu, \nu) = \left(\inf_{\gamma \in \Gamma(\mu,\nu)} \int_{\mathcal{X}^2} d^p(x,y)\, d\gamma(x,y)\right)^{\frac{1}{p}},$$

where $\Gamma(\mu, \nu)$ is the set of all couplings of $\mu$ and $\nu$, that is, the probability measures over $\mathcal{X}^2$ with marginals with respect to the first and second axes in $\mathcal{X}^2$ being equal to $\mu$ and $\nu$ respectively (see definition A.21 for couplings).

Note that in Definition E.3 we defined the unbalanced Earth Mover's distance, which allows comparison between measures with unequal total mass. Although this distance is closely related to, and often coincides with the Wasserstein distance, we distinguish between the two notions by using separate names, each reflecting the context in which the corresponding metric is more appropriate.

*Remark* G.2. A useful property of Wasserstein distance is that the infimum in definition G.1 is always attained, so we can write minimum instead of infimum (Theorem 4.1 in (Villani et al., 2009)) .

Clearly, by Hölder inequality we have that if $p \leq q$, then $W_p \leq W_q$. We will focus on the case where $p = 1$, $\mathcal{X}$ is some Euclidean space $\mathbb{R}^n$, and $d$ is the $\ell_1$ metric, which we often write as $\|x - y\|_1$, where $\|z\|_1 = \sum_{j=1}^n |z_j|$ for $z = (z_j)_{j=1}^n$. By abuse of notation, we write the Wasserstein distance as $W$ instead of $W_1$ in this case.

*Remark* G.3. Although the Wasserstein distance is not a strict metric, as it may take the value $+\infty$, most metric properties continue to hold when infinite distances are permitted (see (Burago et al., 2001)). However, under some boundedness assumptions, we will prove in Lemma G.10 that the Wasserstein distance between two measures in our setting is not only finite, but uniformly bounded.

### G.2. Kantorovich and Rubinstein duality theorem

The next theorem is a useful result for the Wasserstein distance, specific for $p = 1$, called the "*Kantorovich and Rubinstein duality theorem.*"

**Theorem G.4.** *For $\mathcal{X}, \mu$ and $\nu$ as in Definition G.1, we have*

$$W(\mu, \nu) = \sup_{f, L_f \leq 1} \left\{ \int_{\mathcal{X}} f \, d\mu - \int_{\mathcal{X}} f \, d\nu \right\},$$

*where the supremum is taken over all Lipschitz continuous functions $f : \mathcal{X} \to \mathbb{R}$ with Lipschitz constant $L_f \leq 1$.*

### G.3. Wasserstein Spaces

Traditionally, to avoid the situation described above in remark G.3 , the *Wasserstein space* of order $p \geq 1$ is defined as the set of probability measures with finite moment of order $p$ (Villani et al., 2009; Ambrosio et al., 2005). We propose a similar notion, more suitable to our setting, which is equivalent to the standard definition. For a probability measure $\nu$ on $\mathbb{R}^n$, and $p \geq 1$, let

$$\tau_p(\nu) = \max_{1 \leq i \leq n} \int_{\mathbb{R}^n} |x_i|^p \, d\nu(x) \in [0, \infty]. \tag{11}$$

**Definition G.5.** Let $\mathcal{P}(\mathbb{R}^n)$ be the space of Borel probability measures over $\mathbb{R}^n$. For $p \geq 1$ and $n \in \mathbb{N}$, the Wasserstein space of order $p$ is defined to be

$$\mathcal{P}_p(\mathbb{R}^n) := \left\{ \nu \in \mathcal{P}(\mathbb{R}^n) \,\Big|\, \tau_p(\nu) < \infty \right\}.$$

We further define the space

$$\mathcal{P}_{p,c}(\mathbb{R}^n) := \left\{ \nu \in \mathcal{P}(\mathbb{R}^n) \,\Big|\, \tau_p(\nu) \leq c \right\}.$$

We now prove the compactness of the space $\mathcal{P}_{p,c}(\mathbb{R}^n)$. While this is a standard result that can be found in most textbooks regarding optimal transport (see for example Proposition 7.1.5 in (Ambrosio et al., 2005) or Theorem 6.9 in (Villani et al., 2009)), since we slightly modified the setting to our particular case, we prove it for completeness.

We begin by proving that the space is closed.

**Lemma G.6.** *Let $p \geq 1$ and $c \geq 0$. Then the set $\mathcal{P}_{p,c}(\mathbb{R}^n)$ is closed in the metric space $(\mathcal{P}_p(\mathbb{R}^n), W)$.*

*Proof.* Let $(\nu_k) \subset \mathcal{P}_{p,c}(\mathbb{R}^n)$ be a sequence such that $\nu_k \to \nu$ in $W$. For every $i \in \{1, \ldots, n\}$ and $M > 0$, define

$$\phi_M(t) := \min\{|t|^p, M\} \text{ for } t \in \mathbb{R}, \qquad f_{i,M}(x) := \phi_M(x_i) \text{ for } x = (x_i)_{i=1}^n \in \mathbb{R}^n.$$

Let $R := M^{1/p}$. The function $t \mapsto |t|^p$ is $pR^{p-1}$-Lipschitz on $[-R, R]$, and $\phi_M$ is constant outside this interval, hence $L_{\phi_M} \leq pR^{p-1} = pM^{\frac{p-1}{p}}$, where $L_{\phi_M}$ is the Lipschitz constant of $\phi_M$. Since the coordinate projection $x \mapsto x_i$ is 1–Lipschitz with respect to $\| \cdot \|_1$, it follows that $L_{f_{i,M}} \leq pM^{\frac{p-1}{p}}$.

By the Kantorovich–Rubinstein duality theorem (Theorem G.4), we have

$$\left| \int f_{i,M} \, d\nu_k - \int f_{i,M} \, d\nu \right| \le L_{f_{i,M}} W(\nu_k, \nu) \le p M^{\frac{p-1}{p}} W(\nu_k, \nu),$$

which tends to 0 as $k \to \infty$. Therefore,

$$\int f_{i,M} \, d\nu = \lim_{k \to \infty} \int f_{i,M} \, d\nu_k.$$

Since $0 \le f_{i,M}(x) \le |x_i|^p$ for every $x \in \mathbb{R}^n$, we obtain

$$\int f_{i,M} \, d\nu_k \le \int |x_i|^p \, d\nu_k \le \tau_p(\nu_k) \le c,$$

and hence $\int f_{i,M} \, d\nu \le c$ for every $M > 0$. Finally, as $f_{i,M}(x) \to |x_i|^p$ pointwise when $M \to \infty$, the monotone convergence theorem yields

$$\int |x_i|^p \, d\nu = \lim_{M \to \infty} \int f_{i,M} \, d\nu \le c.$$

Since this holds for each $i \in \{1, \ldots, n\}$, we conclude that $\tau_p(\nu) \le c$, i.e. $\nu \in \mathcal{P}_{p,c}(\mathbb{R}^n)$. □

Using the closeness of the space, we move to prove compactness.

**Lemma G.7.** *For any $p > 1$ and $c > 0$, the space $(\mathcal{P}_{p,c}(\mathbb{R}^n), W)$ is compact.*

The proof relies on the characterization that a set of probability measures is relatively compact in Wasserstein distance if and only if it is tight and has uniformly integrable first moments ((Ambrosio et al., 2005), Proposition 7.1.5).

**Proposition G.8.** *A set $\mathcal{P} \subset \mathcal{P}_p(\mathbb{R}^n)$ of measures is relatively compact (with respect to the topology induced by the Wasserstein metric) if and only if the following two conditions are satisfied:*

1. *$\mathcal{P}$ is tight, that is, for every $\epsilon > 0$ there exists a compact set $K_\epsilon \subset \mathbb{R}^n$ such that for every $\nu \in \mathcal{P}$, $\nu(K_\epsilon) \ge 1 - \epsilon$.*

2. *$\mathcal{P}$ is uniformly integrable, that is,*

$$\lim_{R \to \infty} \sup_{\nu \in \mathcal{P}} \int_{\{\|x\|_1 > R\}} \|x\|_1 \, d\nu(x) = 0.$$

*Proof.* We begin by proving tightness. We show that uniform bound on $\tau_p$ implies uniform bound on $\tau_1$. For any $x \in \mathbb{R}^n$, we denote its coordinates by $(x_j)_{j=1}^n$. We have that $|x_j| \le |x_j|^p + 1$ for every $1 \le j \le n$. Hence, for every $\nu \in \mathcal{P}_{p,c}(\mathbb{R}^n)$,

$$\int_{\mathbb{R}^n} |x_j| \, d\nu(x) \le \int_{\mathbb{R}^n} (|x_j|^p + 1) \, d\nu(x),$$

taking the maximum over $1 \le j \le n$ gives

$$\tau_1(\nu) \le \tau_p(\nu) + 1 \le c + 1.$$

By Markov's inequality, for every $M > 0$,

$$\nu(\{\, x \in \mathbb{R}^n \, : \, |x_j| > M \,\}) \le \frac{1}{M} \int_{\mathbb{R}^n} |x_j| \, d\nu \le \frac{c+1}{M}.$$

Note that the complement of the cube $[-M, M]^n$ is the set of all $x \in \mathbb{R}^n$ such that there exists at least one coordinate $x_j$ such that $|x_j| > M$. Hence,

$$\nu(\mathbb{R}^n \setminus [-M, M]^n) \le \sum_{j=1}^n \nu(\{\, x \in \mathbb{R}^n \, : \, |x_j| > M \,\}) \le \frac{n(c+1)}{M}.$$

Given $\epsilon > 0$, we choose $M$ large enough ssuch that $\frac{n(c+1)}{M} < \epsilon$, thus proving tightness.

We move to prove uniform integrability. Denote by $B_{>R} := \{\, x \in \mathbb{R}^n : \|x\|_1 > R \,\}$. We will prove that

$$\lim_{R \to \infty} \int_{B_{>R}} \|x\|_1 \, d\nu(x) = 0,$$

for every $\nu \in \mathcal{P}_{p,c}(\mathbb{R}^n)$.

For every $x \in B_{>R}$ we have that $\|x\|_1 = \|x\|_1^p \cdot \frac{1}{\|x\|_1^{p-1}} < \|x\|_1^p \cdot \frac{1}{R^{p-1}}$. Then

$$\int_{B_{>R}} \|x\|_1 \, d\nu(x) < \frac{1}{R^{p-1}} \int_{B_{>R}} \|x\|_1^p \, d\nu(x)$$
$$\leq \frac{n^{p-1}}{R^{p-1}} \int_{B_{>R}} \sum_{j=1}^{n} |x_j|^p \, d\nu(x) \leq \frac{n^p c}{R^{p-1}},$$

where the second to last inequality holds because of Hölder inequality, and the last inequality holds because $\int |x_j|^p \, d\nu \leq c$ for every $\nu \in \mathcal{P}_{p,c}(\mathbb{R}^n)$. As $R$ tends to $\infty$, we get that the expression $\frac{n^p c}{R^{p-1}}$ tends to 0, thus proving uniform integrability.

Relative compactness combined with the closeness of $\mathcal{P}_{p,c}(\mathbb{R}^n)$ (lemma G.6) gives that the space $(\mathcal{P}_{p,c}(\mathbb{R}^n), W)$ is compact. $\qquad\square$

*Remark* G.9. Convergence in Wasserstein distance implies weak convergence, and hence convergence in Levy-Prokhorov metric (see Definition B.1 and Theorem 5.9 in (Santambrogio, 2015)). When adding the uniform integrability condition, the converse direction holds. Specifically, weak convergence of probability measures together with uniform integrability implies convergence in $W$, thus showing that both the Levy-Prokhorov metric and Wasserstein distance metrize the weak topology on $\mathcal{P}_{p,c}(\mathbb{R}^n)$.

Next, we prove the following lemma, that states that the distance between two probability measures in $\mathcal{P}_{p,c}(\mathbb{R}^n)$ is uniformly bounded, and the bound only depends on $p, c$ and the dimension $n$.

**Lemma G.10.** *Let* $p \geq 1$, $c > 0$ *and* $\mu, \nu \in \mathcal{P}_{p,c}(\mathbb{R}^n)$. *Then*

$$W(\mu, \nu) \leq 2n c^{1/p}.$$

*Proof.* Let $\gamma \in \Gamma(\mu, \nu)$ be a coupling of $\mu$ and $\nu$. For $x, y \in \mathbb{R}^n$, write $x = (x_i)_{i=1}^n, y = (y_i)_{i=1}^n$. Then for every $1 \leq i \leq n$, we have

$$|x_i - y_i| \leq |x_i| + |y_i|.$$

Integrating both sides with respect to $\gamma$

$$\int_{\mathbb{R}^n \times \mathbb{R}^n} |x_i - y_i| \, d\gamma(x, y) \leq \int_{\mathbb{R}^n \times \mathbb{R}^n} |x_i| \, d\gamma(x, y) + \int_{\mathbb{R}^n \times \mathbb{R}^n} |y_i| \, d\gamma(x, y).$$

Since $\gamma$ has marginals $\mu$ and $\nu$

$$\int_{\mathbb{R}^n \times \mathbb{R}^n} |x_i - y_i| \, d\gamma(x, y) \leq \int_{\mathbb{R}^n} |x_i| \, d\mu(x) + \int_{\mathbb{R}^n} |y_i| \, d\nu(y).$$

It is well known that for every measurable function $g$, if $q \leq p$ then $\|g\|_q \leq \|g\|_p$ (recall that we are integrating with respect to probability measures). Choosing $q = 1$ gives,

$$\int_{\mathbb{R}^n} |x_i| \, d\mu(x) \leq \left( \int_{\mathbb{R}^n} |x_i|^p \, d\mu(x) \right)^{\frac{1}{p}} \leq c^{\frac{1}{p}}, \qquad \int_{\mathbb{R}^n} |y_i| \, d\nu(y) \leq \left( \int_{\mathbb{R}^n} |y_i|^p \, d\nu(y) \right)^{\frac{1}{p}} \leq c^{\frac{1}{p}}.$$

Putting all together gives

$$
\int_{\mathbb{R}^n \times \mathbb{R}^n} \|x - y\|_1 \, d\gamma(x, y) = \int_{\mathbb{R}^n \times \mathbb{R}^n} \sum_{i=1}^n |x_i - y_i| \, d\gamma(x, y) =
$$

$$
\sum_{i=1}^n \int_{\mathbb{R}^n \times \mathbb{R}^n} |x_i - y_i| \, d\gamma(x, y) \leq \sum_{i=1}^n \Big( \int_{\mathbb{R}^n \times \mathbb{R}^n} |x_i - y_i|^p \Big)^{\frac{1}{p}} d\gamma(x, y) \leq
$$

$$
\sum_{i=1}^n 2c^{1/p} = 2nc^{1/p}.
$$

Taking the infimum of all couplings of $\mu$ and $\nu$ we get

$$
W(\mu, \nu) \leq 2nc^{1/p}.
$$

$\square$

# H. Graphop-Signal Analysis

A recent approach to graph limit theory that extends the well known approaches for dense and sparse graph limit was introduced in (Backhausz & Szegedy, 2022). This approach generalizes several past graph limit theories for dense graphs, as well as for sparse graphs, such as stretched graphons, $\mathcal{L}^p$ graphons and graphings (Jian et al., 2023; Ji et al., 2024). In (Backhausz & Szegedy, 2022), a new perspective was presented by focusing on how the adjacency matrix (and its extension for graphs of "infinite order") acts on signals as an operator.

## H.1. Graphop-Signals

In this section, we introduce the main theme of the paper "*Graphop-Signal Analysis*". We extend the setting that was presented in (Backhausz & Szegedy, 2022), by generalizing the notion of P-operators and graphops to incorporate signals, thereby providing the formulation for our following analysis

### H.1.1. P-OPERATOR-SIGNALS

Various linear operators can be linked with graphs, including the kernel operator associated with graphons and $\mathcal{L}^p$ graphons, Laplace operators, and Markov kernel operators - related to random walks on graphs (Borgs et al., 2019; 2018b). Here, the operator acts on some $\mathcal{L}^p(\Omega)$ space, where $\Omega$ is a Borel probability space parameterizing the set of vertices. Hence, the following definition, proposed in (Backhausz & Szegedy, 2022), is a natural extension that generalizes all of the above concepts and more.

**Definition H.1** (P-Operator (Backhausz & Szegedy, 2022))**.** Let $(\Omega, \mathcal{F}, \mu)$ be a probability space.

1. A P-operator is a bounded linear operator $A : \mathcal{L}^\infty(\Omega) \to \mathcal{L}^1(\Omega)$, i.e., such that

$$
\|A\|_{\infty \to 1} := \sup_{f \in \mathcal{L}^\infty(\Omega), f \neq 0} \frac{\|Af\|_1}{\|f\|_\infty}
$$

   is finite. Denote by $\mathcal{B}(\Omega)$ the set of all $P$-operators on $\Omega$.

2. For $(p, q) \in [1, \infty]^2$, a P-operator $A$ is said to have a finite $(p, q)$ norm if the norm

$$
\|A\|_{p \to q} := \sup_{f \in \mathcal{L}^\infty(\Omega), f \neq 0} \frac{\|Af\|_q}{\|f\|_p}
$$

   is finite. Denote by $\mathcal{B}_{p,q}(\Omega)$ the space of all P-operators with finite $(p, q)$ norm.

*Remark* H.2. For a measurable function $f$ on the probability space $\Omega$, it is well known that if $p' \geq p$ then $\|f\|_{p'} \geq \|f\|_p$. If $p, p', q, q' \in [1, \infty]$ satisfy $p' \geq p$ and $q' \leq q$, then

$$
\|A\|_{p' \to q'} \leq \|A\|_{p \to q}.
$$

Therefore, $\mathcal{B}_{p,q}(\Omega) \subset \mathcal{B}_{p',q'}(\Omega)$. In particular, we have that $\mathcal{B}(\Omega) = \mathcal{B}_{\infty,1}(\Omega)$ contains every $\mathcal{B}_{p,q}(\Omega)$ for every $(p,q) \in [1,\infty]^2$, even if we omit the requirement that $A$ be a P-operator in 2. of Definition H.1, and simply take $A$ as a general linear operator.

Since the most standard data type in graph machine learning are graphs with node features, we extend the definition of P-operators to accommodate this, and introduce *P-operator-signals*. These will be one of the main objects of our analysis.

**Definition H.3** (P-Operator-Signal). Let $d \geq 0$. A *P-operator-signal* is a pair $(A, f)$, where $A : \mathcal{L}^\infty(\Omega) \to \mathcal{L}^1(\Omega)$ is a P-operator (Definition H.1), and $f : \Omega \to [-1, 1]^d$ is measurable. The integer $d$ is referred to as the *feature dimension* of the signal. The space of all P-operator-signals in which the P-operator has finite $(p, q)$ norm, for some $(p, q) \in [1, \infty]^2$, and $d$ is the feature dimension, is denoted by $\mathcal{B}_{p,q,d}(\Omega)$. The space of all P-operators-signals with $\|A\|_{p \to q} \leq r$ is denoted by $\mathcal{B}_{p,q,d}^r(\Omega)$.

We will often denote by $\mathcal{B}_d(\Omega)$ and $\mathcal{B}_d^r(\Omega)$ instead of $\mathcal{B}_{\infty,1,d}(\Omega)$ and $\mathcal{B}_{\infty,1,d}^r(\Omega)$ respectively. In the degenerate case $d = 0$, the signal component is absent and a P-operator-signal reduces to a P-operator, recovering the original setting of (Backhausz & Szegedy, 2022).

Given $n \in \mathbb{N}$, and $n$ signals $(f_i : \Omega \to [-1, 1]^{d_i})_{i \in [n]}$, let $(A, (f_1 \ldots, f_n))$ be the P-operator-signal whose signal $f = (f_1 \ldots, f_n)$ is the concatenation of the signals $f_i$, $i \in [n]$. Note that every signal $f : \Omega \to [-1, 1]^d$ can be written as concatenation $(f_1, \ldots, f_d)$ where each $f_i$ is 1-channel signal. That is, $f_i : \Omega \to [-1, 1]$ for every $i \in [d]$.

### H.1.2. EXAMPLES OF P-OPERATORS

Here, we present basic examples of P-operators.

**Graphs.** Let $G = (V, E)$ be a graph with $|V| = n$. We take $(\Omega, \mu)$ to be the probability space $([n], \mu_{[n]})$, where $[n] = \{1, \ldots, n\}$ and $\mu_{[n]}$ is the uniform measure on $[n]$ (i.e. the counting measure). Each vector in $\mathbb{R}^n$ can be seen as a function $v : [n] \to \mathbb{R}$, and matrices are linear operators over $\mathbb{R}^n$. There are various options to define the P-operator that corresponds to the graph $G$. One of which is to take the adjacency matrix $A_G$, and then the application of $A_G$ on $v$ is simply the matrix multiplication $A_G \cdot v$. Another option is to normalize $A_G$ by the size of the graph, and then $(A_G)v = \frac{1}{|V|} A_G \cdot v$. Another interesting representation of graphs as matrices is as a graph Laplacian. The combinatorial Laplacian of $G$, denoted by $L(G)$ is defined by

$$(L(G))(v)(i) = d(i)v(i) - \sum_{(j,i) \in E} v(j),$$

where $d(i)$ denotes the degree of the vertex $i \in [n]$. For more details see section 5 in (Backhausz & Szegedy, 2022).

**Graphons.** Let $(\Omega, \mu)$ be the space $[0, 1]$ with the Lebesgue measure. We previously discussed how graphs can be represented by graphons (see Section D.1). Graphons can also be seen as kernel operators that act on measurable functions over $[0, 1]$ (see for example section 8 in (Backhausz & Szegedy, 2022), or (Lovász, 2012)). For a graphon $W : [0, 1]^2 \to [0, 1]$ and a measurable function $f \in \mathcal{L}^2([0, 1])$, we define the P-operator that corresponds to the graphon $T_W$ as

$$(T_W f)(x) = \int_0^1 W(x, y) f(y) \, dy.$$

**Additional Examples.** Sparse graph limits can also be represented by P-operators, such as stretched graphons, $\mathcal{L}^p$ graphons, and graphings. The reader can find brief introduction to these concepts in Appendix O. Sparse graph limits can also be described using neighborhoods that are sets of measure 0, as we explain in Section H.1.4.

### H.1.3. GRAPHOP-SIGNALS

We now restrict the notion of P-operators to include properties that make them more graph-like objects, called *graphops*. The construction of graphops was proposed in (Backhausz & Szegedy, 2022), and we extend it trivially to *graphop-signals*. To define graphops, we will use the following notation. For two measurable functions $v, u \in L^\infty(\Omega)$, and a P-operator $A : \mathcal{L}^\infty(\Omega) \to \mathcal{L}^1(\Omega)$ we define the bilinear form $(v, u)_A$ by

$$(v, u)_A = \int_\Omega (Av) \cdot u \, d\mu.$$

Note that $|(v, u)_A| \le \|v\|_\infty \|u\|_\infty \|A\|_{\infty \to 1} < \infty$.

**Definition H.4** (Graphop-signal). Let $A \in \mathcal{B}(\Omega)$ be a P-operator.

- The operator $A$ is called *self adjoint* if for every $u, v \in \mathcal{L}^\infty(\Omega)$, $(v, u)_A = (u, v)_A$.

- The operator $A$ is *positively preserving* if for every $v \in \mathcal{L}^\infty(\Omega)$ such that $v(x) \ge 0$ for a.e. $x \in \Omega$, we have that $Av(x) \ge 0$ for a.e. $x \in \Omega$.

- The operator $A$ is called a *graphop* if it is positively preserving and self adjoint. Denote by $\mathscr{G}(\Omega)$ the space of all graphops over $\Omega$.

- The pair $(A, f)$ where $A \in \mathscr{G}(\Omega)$ is a graphop, and $f : \Omega \to [-1, 1]^d$ is a $d$-channel signal is called *graphop-signal*.

Denote by $\mathscr{G}_{p,q,d}(\Omega)$ the space of all graphop-signals in which the graphop has finite $(p, q)$ norm, for some $(p, q) \in [1, \infty]^2$, and $d$ is the feature dimension. We further denote by $\mathscr{G}_{p,q,d}^r(\Omega)$ the spaces the spaces of all graphop-signals with uniformly bounded $(p, q)$ norms respectively, when $r > 0$ is the uniform bound. We often denote $\mathscr{G}_d(\Omega)$ and $\mathscr{G}_d^r(\Omega)$ instead of $\mathscr{G}_{\infty,1,d}(\Omega)$ and $\mathscr{G}_{\infty,1,d}^r(\Omega)$ respectively. In the degenerate case $d = 0$, the signal component is absent and a graphop-signal reduces to a graphop, recovering the original setting of (Backhausz & Szegedy, 2022).

### H.1.4. THE EXPLICIT ADJACENCY STRUCTURE OF GRAPHOPS

We now show that, given a graphop, one can define a meaningful notion of a corresponding edge set in $\Omega^2$ and vertex neighborhoods. More accurately, these sets are characterized by measures on $\Omega^2$ and $\Omega$ respectively. Hence, graphops are natural extensions of graphs to infinite vertex sets. Moreover, these edge and neighborhood measures can in general be supported on null sets with respect to the base measure $\mu$ of $\Omega$. Hence, graphops are interpreted as sparse objects in the null-set support case.

The following theorem and corollary, presented in section 6 of (Backhausz & Szegedy, 2022), gives a unique representation of graphops. It is shown that each graphop can be represented uniquely by a finite measure over $\Omega^2$.

**Theorem H.5** (Theorem 6.3 in (Backhausz & Szegedy, 2022)). *Let $A \in \mathscr{G}(\Omega)$ be a graphop over a Borel probability space $(\Omega, \mathcal{F}, \mu)$. Then, there is a unique finite measure $\nu$ on $(\Omega \times \Omega, \mathcal{F} \otimes \mathcal{F})$ with the following properties.*

1. *$\nu$ is symmetric, i.e. $\nu(S \times T) = \nu(T \times S)$ holds for every $S, T \in \mathcal{F}$.*

2. *The two marginal distribution of $\nu$ on $\Omega$ are absolutely continuous with respect to $\mu$.*

3. *$(f, g)_A = \int_{\Omega^2} f(x) g(y) \, d\nu(x, y)$ holds for every $f, g \in L^\infty(\Omega)$.*

*Conversely, if $\nu$ is a finite measure on $(\Omega \times \Omega, \mathcal{F} \otimes \mathcal{F})$ satisfying the first two properties, then there is a unique graphop $A$ that satisfies the third property.*

The measure $\nu$ from Theorem H.5, that represents the graphop $A \in \mathscr{G}(\Omega)$, is defined by

$$\nu(S \times T) = (\mathbb{1}_S, \mathbb{1}_T)_A = \int_T A\mathbb{1}_S(\omega) \, d\mu(\omega), \tag{12}$$

for every two measurable sets $S, T \subset \Omega$. This measure is uniquely extended to general measurable sets in $\Omega \times \Omega$. The application of the graphop $A$ to a signal $f$ at a fixed point $x \in \Omega$ corresponds to integrating the signal over a fiber measure $\nu_x$. These fiber measures are obtained via the disintegration of $\nu$ and describe the "neighborhoods" within $A$ for every $x \in \Omega$ (recall Theorem A.22).

*Remark* H.6 (Fiber Measures). We can recover the graphop $A$ from the measure $\nu$ using disintegration. By the disintegration theorem (Theorem A.22), we obtain a family of measures $\{\nu_y\}_{y \in \Omega}$ on the measurable space $\Omega$ such that

$$(Af)(y) = \int_\Omega f \, d\nu_y.$$

Here, the mapping from $\Omega^2$ to the carrier space in the definition of disintegration is the projection upon the second axis $\Omega$ of $\Omega^2$.

In the following, we extend theorem H.5, and give a characterization of graphops with finite $\|\cdot\|_{\infty\to\infty}$ norm.

**Theorem H.7** ($\mathcal{L}^\infty \to \mathcal{L}^\infty$ boundedness of graphops)**.** *Let $A$ be a graphop over a Borel probability space $(\Omega, \mu)$, and let $\nu$ be its unique representing measure on $\Omega \times \Omega$ as in Theorem H.5. Let $\{\nu_x\}_{x\in\Omega}$ be the family of fiber measures obtained by disintegrating $\nu$ with respect to $\mu$ using the projection upon the second axis $\Omega$. Then $A$ is a bounded operator $A : \mathcal{L}^\infty(\Omega) \to \mathcal{L}^\infty(\Omega)$ if and only if*

$$\operatorname*{ess\,sup}_{x\in\Omega} \nu_x(\Omega) < \infty.$$

*Moreover, in this case,*

$$\|A\|_{\infty\to\infty} = \operatorname*{ess\,sup}_{x\in\Omega} \nu_x(\Omega).$$

*Proof.* By Theorem H.5 and the disintegration construction, the action of the graphop $A$ is

$$(Af)(x) = \int_\Omega f(y)\, d\nu_x(y) \quad \text{for } \mu\text{-a.e. } x \in \Omega,$$

for every $f \in \mathcal{L}^\infty(\Omega)$.

Assume first that

$$M := \operatorname*{ess\,sup}_{x\in\Omega} \nu_x(\Omega) < \infty.$$

Then for any $f \in \mathcal{L}^\infty(\Omega)$ and for $\mu$-a.e. $x$,

$$|(Af)(x)| \leq \int_\Omega |f(y)|\, d\nu_x(y) \leq \|f\|_\infty\, \nu_x(\Omega) \leq M\|f\|_\infty.$$

Taking the essential supremum over $x$ shows that $\|Af\|_\infty \leq M\|f\|_\infty$. Moreover, $M$ is the supremum of $\|Af\|_\infty / \|f\|_\infty$, realized for $f = \mathbb{1}_\Omega$. Hence, $\|A\|_{\infty\to\infty} = M$.

Conversely, suppose that $A : \mathcal{L}^\infty(\Omega) \to \mathcal{L}^\infty(\Omega)$ is bounded with operator norm $C$. Applying $A$ to the constant function $\mathbb{1}_\Omega$, we obtain

$$(A\mathbb{1}_\Omega)(x) = \int_\Omega 1\, d\nu_x = \nu_x(\Omega) \quad \text{for } \mu\text{-a.e. } x.$$

Therefore,

$$\operatorname*{ess\,sup}_{x\in\Omega} \nu_x(\Omega) = \|A\mathbb{1}_\Omega\|_\infty \leq C\|\mathbb{1}_\Omega\|_\infty = C,$$

showing that the fiber measures have essentially uniformly bounded total mass.

$\square$

Note that by the self-adjointness assumption of graphops and the duality of $\mathcal{L}^1(\Omega)$ and $\mathcal{L}^\infty(\Omega)$ we have that $\mathscr{G}_{\infty,\infty} = \mathscr{G}_{1,1}$ (this is also immediate from Lemma 2.1 of (Hrušková, 2022)). We then introduce our main object of interest, *Bounded Fibers Operator* (bofop).

**Definition H.8.** Let $(\Omega, \mu)$ be a Borel probability space, and let $A$ be a graphop over $\Omega$. We say that $A$ is *Bounded fibers operator* (bofop), if $A$ has finite $\infty \to \infty$ or equivalently $1 \to 1$ norms. Namely, $A \in \mathscr{G}_{\infty,\infty}(\Omega)$ or $\mathscr{G}_{1,1}(\Omega)$. Furthermore, we define the space of all bofops over $\Omega$ as $\mathcal{BF}(\Omega)$.

Similarly to Definitions H.3 and H.4, we call the pair $(A, f)$ bofop-signal if $A$ is a bofop, and $f$ is $d$-channel signal. The space of graphop-signals is denoted by $\mathcal{BF}_d(\Omega)$, and the space of bofop-signals with uniformly bounded $\infty \to \infty$ norm by $\mathcal{BF}_d^r(\Omega)$.

### H.1.5. EXAMPLE: SPHERICAL GRAPHOP

As an example of a graphop and its fibers, we recall the *spherical graphop*, presented in (Backhausz & Szegedy, 2022). We will show that this graphop is in $\mathscr{G}_{\infty,\infty}$, thus illustrating that various limit objects of sparse graphs can be represented as $\mathcal{L}^\infty \to \mathcal{L}^\infty$ graphops.

Consider $S^2 = \{(x, y, z) : x^2 + y^2 + z^2 = 1\}$ with the uniform probability measure $\mu$. We define a set of "edges" $E \subset S^2 \times S^2$ by: $(a, b) \in E$ if and only if $a \perp b$. Then $(S^2, E)$ is a Borel graph, i.e., $E$ os a Borel set (see our discussion on graphing in Appendix O.3). Let $\nu_a$ be the uniform probability measure on $\mathcal{N}(a) = \{b \in \Omega : b \perp a\}$. We define the spherical graphop $A : \mathcal{L}^\infty(\Omega, \mu) \to \mathcal{L}^\infty(\Omega, \mu)$, by

$$(Af)(a) := \int_{b \perp a} f(b) \, d\nu_a.$$

**Proposition H.9.** *The spherical graphop $A$ has operator norm $\|A\|_{\infty \to \infty} = 1$. In particular $A \in \mathcal{BF}_d^r(\Omega)$.*

*Proof.* Fix $f \in \mathcal{L}^\infty(S^2)$. For each $a \in S^2$ we have, by definition,

$$(Af)(a) = \int_{\mathcal{N}(a)} f(b) \, d\nu_a(b), \qquad \mathcal{N}(a) := \{b \in S^2 : \ b \perp a\}.$$

Since $\nu_a$ is the *normalized uniform measure* on the circle $\mathcal{N}(a)$, it is a probability measure and so $\nu_a(\mathcal{N}(a)) = 1$. Hence, for every $a \in S^2$,

$$|(Af)(a)| \leq \int_{\mathcal{N}(a)} |f(b)| \, d\nu_a(b) \leq \|f\|_\infty \int_{\mathcal{N}(a)} 1 \, d\nu_a(b) = \|f\|_\infty.$$

Taking the essential supremum over $a$ gives $\|Af\|_\infty \leq \|f\|_\infty$. Therefore $A$ defines a bounded linear operator $\mathcal{L}^\infty(S^2) \to \mathcal{L}^\infty(S^2)$, and

$$\|A\|_{L^\infty \to L^\infty} \leq 1.$$

Moreover, for the constant function $\mathbb{1}_\Omega$ we have

$$(A\mathbb{1}_\Omega)(a) = \int_{\mathcal{N}(a)} 1 \, d\nu_a = 1 \quad \text{for all } a \in S^2,$$

so $\|A\mathbb{1}_\Omega\|_\infty = \|\mathbb{1}_\Omega\|_\infty = 1$, which implies $\|A\|_{\infty \to \infty} \geq 1$. So,

$$\|A\|_{\infty \to \infty} = 1,$$

and in particular $A \in \mathscr{G}_{\infty, \infty}(\Omega)$. $\qquad\square$

We can interpret the spherical graphop as representation of a sparse structure as follows. Although each neighborhood $\mathcal{N}(a)$ has $\mu$-measure zero, the associated probability measure $\nu_a$ is nontrivial. As a result, the operator $A$ produces nontrivial outputs by aggregating signals on a *sparse* neighborhood, i.e., a neighborhood of measure zero.

### H.2. Profiles as Representations of Graphop-Signals

One goal in graphop theory is to treat all graphops over all probability spaces in a unified manner. The idea is to represent different graphops, that act on different probability spaces $\Omega$ and $\Omega'$, as points in a the same space. This unified representation allows defining a metric on the space of all P-operators over all domains $\Omega \in \mathbf{\Omega}$ that come from some predefined arbitrary space of probability spaces $\mathbf{\Omega}$. To eliminate the dependency on the specific domain $\Omega$, P-operators are represented via the histogram of their output values on input functions.

#### H.2.1. PROFILES

In the following definition we formalizes the above idea. This is a natural extension of (Backhausz & Szegedy, 2022) from P-operators to P-operator-signals.

**Definition H.10** ($k$-Profile)**.** Let $\Omega$ be a probability space and $k \geq 0$ be an integer.

- Let $v_1, \ldots, v_k : \Omega \to [-1, 1]$ be measurable. We denote by $\mathrm{D}(v_1, \ldots, v_k)$ the joint distribution of the random variables $v_1, \ldots, v_k$, i.e., the Borel measure in $\mathbb{R}^k$ that is the pushforward of $\mu$ under the map $x \mapsto (v_1(x), \ldots, v_k(x))$ (see Definition A.17 for pushforward).

- Let $(A, f) \in \mathcal{B}_d(\Omega)$ be a P-operator-signal. The *P-distribution* of $(A, f)$ with respect to $v_1, \ldots, v_k : \Omega \to [-1, 1]$ is defined to be $\mathrm{D}_{(A,f)}(v_1, \ldots, v_k) := \mathrm{D}(v_1, \ldots, v_k, Av_1, \ldots, Av_k, f)$. We call $k$ the order of the P-distribution $\mathrm{D}_{(A,f)}(v_1, \ldots, v_k)$.

- The $k$-profile $\mathrm{S}_k(A, f)$ of $(A, f)$ is the set of all P-distributions of order $k$, i.e., $\mathrm{S}_k(A, f) = \{\mathrm{D}_{(A,f)}(v_1, \ldots, v_k) \mid v_1, \ldots, v_k : \Omega \to [-1, 1] \text{ measurable}\}$. By convention, for $k = 0$ the tuple $(v_i)_{i=1}^k$ is empty, so $\mathrm{S}_0(A, f) = \{\mathrm{D}(f)\}$.

- The *profile* $\mathrm{S}(A, f)$ of $(A, f)$ is defined to be the sequence of $k$-profiles of all orders $k \geq 0$. Explicitly

$$\mathrm{S}(A, f) := (\mathrm{S}_k(A, f))_{k=0}^{\infty}.$$

Using the definition of $k$-profiles, we can now define the space of all profiles representing P-operator-signals. This allows us to treat different operators solely through their sets of distributions, independently of their original domain $\Omega$.

**Definition H.11** (Space of Profiles). Let $r > 0$, $(p, q) \in [1, \infty]^2$, and $d \geq 0$. We define the space of $k$-profiles, denoted by $\boldsymbol{S_k}$, as the collection of all $k$-profiles generated by some P-operator-signals in $\mathcal{B}_{p,q,d}^r(\Omega)$ across all probability spaces $\Omega$. That is

$$\boldsymbol{S_{k,d,p,q}^r} := \left\{ \mathrm{S}_k(A, f) \mid (\Omega, \mu) \text{ is a probability space}, \ (A, f) \in \mathcal{B}_{p,q,d}^r(\Omega) \right\}. \tag{13}$$

Furthermore, we define the space of profiles $\boldsymbol{S}$ as the collection of all profiles generated by some P-operator-signals in $\mathcal{B}_{p,q,d}^r(\Omega)$ across all probability spaces $\Omega$. That is

$$\boldsymbol{S_{d,p,q}^r} := \left\{ \mathrm{S}(A, f) \mid (\Omega, \mu) \text{ is a probability space}, \ (A, f) \in \mathcal{B}_{p,q,d}^r(\Omega) \right\}. \tag{14}$$

Since "the set of all probability spaces" is not a well defined object in set theory, one considers an arbitrary set $\boldsymbol{\Omega}$ of probability spaces, and Definition H.11 is restricted to P-operator-signals over all $\Omega \in \boldsymbol{\Omega}$. The analysis does not depend on the choice of $\boldsymbol{\Omega}$.

In graphop theory, different P-operators over different probability spaces are represented via their profiles. The profile is a set of P-distributions, each representing a histograms of output values attained by the P-operator on a set of input signals. Note that P-distributions of the same order $k$ are measures over the Euclidean space of dimension $k$, irrespective of the specific probability space $\Omega$ underlying the P-operator. Hence, representing them as profiles puts all P-distributions on equal footing. Next we use this unified representation to define a metric between P-operator-signals.

### H.2.2. ACTION CONVERGENCE OF P-OPERATOR-SIGNALS

Extending the work of Backhausz and Szegedy (Backhausz & Szegedy, 2022), we define a convergence of sequences of P-operator-signals via their associated profiles. The resulting topology is induced by the Hausdorff distance between profiles (see section B.3), and is then metrized by an appropriate metric.

By definition, the Hausdorff distance is defined for subsets of some metric space. Since we define a metric between profiles, which are sets of Borel probability measures, we use the Wasserstein distance as the base metric (see Appendix G). Originally, Backhausz and Szegedy used the Levy-Prokhorov metric $d_{\mathrm{LP}}$ between probability measures (see Definition B.1), as it metrizes the weak topology (see Appendix B.2). However, we found that methods from optimal transport are quite useful for graphop analysis, so we changed the base metric to be the Wasserstein metric. As we shown in Remark G.9, the Wasserstein distance also metrizes the weak topology under our setting.

For the convenience of the reader, we restate Definition B.2 in our specific framework.

**Definition H.12** (Hausdorff Distance between $k$-profiles). Given two $k$-profiles $S_1, S_2 \subset \boldsymbol{S_{k,d,p,q}^r}$, their *Hausdorff distance* is defined as

$$d_H(S_1, S_2) := \max \left\{ \sup_{\mu_1 \in S_1} \inf_{\mu_2 \in S_2} W(\mu_1, \mu_2) , \ \sup_{\mu_2 \in S_2} \inf_{\mu_1 \in S_1} W(\mu_1, \mu_2) \right\}.$$

As discussed above, we define convergence of P-operator-signals via convergence of their profiles in the Hausdorff distance.

**Definition H.13** (Action Convergence Of P-operator-signals). We say that a sequence of P-operator-signals $\{(A_i, f_i) \in \mathcal{B}_d(\Omega_i)\}_{i=1}^{\infty}$ is action convergent if for every $k \geq 0$ the sequence $\{\mathrm{S}_k(A_i, f_i)\}_{i=1}^{\infty}$ is a Cauchy sequence in $d_H$.

This leads us to define the *action convergence metric*, which is defined using the Hausdorff distance between profiles.

**Definition H.14** (Metrization Of Action Convergence). The *action metric* between two $P$-operator-signals $(A, f), (B, g)$ is defined to be

$$d_M((A, f), (B, g)) := \sum_{k=0}^{\infty} 2^{-k} d_H \left( \mathrm{S}_k(A, f), \mathrm{S}_k(B, g) \right).$$

We note that for lack of a name of the above metric in the original paper (Backhausz & Szegedy, 2022), we name it here the *action metric*. The action metric induces a sequential topology, and convergence of sequences in this topology is exactly action convergence.

Under the above definitions, in Section 8 of (Backhausz & Szegedy, 2022) it was proven that when restricted to graphons, action convergence of P-operators is equivalent to graphon convergence in cut distance. Furthermore, it was proven that on the graphon space, the two metrics $d_M$ and the cut metric are equivalent (see Theorem 8.2 in (Backhausz & Szegedy, 2022)).

In (Backhausz & Szegedy, 2022), Theorem 2.9, the existence of a limit object for every action convergent sequence was proven. Explicitly, for an action convergent sequence of P-operators with uniformly bounded $(p, q)$ norms, where $(p, q) \in [1, \infty]^2$, there exists a P-operator that is the limit of the sequence in the action metric. As the extension of the proof to P-operator-signals follows identical arguments as in section 4 of (Backhausz & Szegedy, 2022), we state the theorem without repeating the proof.

**Theorem H.15** (Existence of Action Limit Object). *Let $d \geq 0$, $r > 0$, and $(p, q) \in [1, \infty]^2$. Let $\{(A_n, f_n)\}_{n=1}^{\infty}$ be an action convergent sequence of P-operator-signals in $\mathcal{B}_{p,q,d}^r(\Omega_n)$. Then, there exists a probability space $\Omega$, and a P-operator-signal $(A, f) \in \mathcal{B}_{p,q,d}^r(\Omega)$ such that*

$$\lim_{n \to \infty} d_M((A_n, f_n), (A, f)) = 0.$$

*Moreover, the limit operator satisfies the norm bound $\|A\|_{p \to q} \leq r$.*

H.2.3. COMPACTNESS OF THE SPACE OF PROFILES

We now prove that under the uniform bound of the $(p, q)$ norms of P-operators, we have that profiles are subsets of the Wasserstein spaces. See Definition G.5. This observation is a key property in the proof of compactness of the space of profiles.

**Lemma H.16.** *Let $(p, q) \in [1, \infty] \times [1, \infty)$, and let $(A, f) \in \mathcal{B}_{p,q,d}^r(\Omega)$ be P-operator-signal. Then, for every $k \geq 0$, $\mathrm{S}_k(A, f) \subset \mathcal{P}_{q,c}(\mathbb{R}^{2k+d})$, where $c = \max\{1, r^q\}$.*

*Proof.* For $k = 0$ the result is trivial. Let $\nu := \mathrm{D}_{(A,f)}(v_1, \ldots, v_k) \in \mathrm{S}_k(v_1, \ldots, v_k)$. Since for every $1 \leq j \leq k, 1 \leq i \leq d$, $\|v_j\|_{\infty}, \|f_i\|_{\infty} \leq 1$, and $\|Av_j\|_q \leq r$, we have that

$$
\begin{aligned}
\tau_q(\nu) &= \max_{n \in [2k+d]} \left\{ \int_{\mathbb{R}^{2k+d}} |x_n|^q \, d\nu(x) \right\} \\
&= \max_{i \in [k], j \in [d]} \left\{ \int_{\mathbb{R}^{2k+d}} |v_i(\omega)|^q \, d\mu(\omega), \int_{\Omega} |Av_i(\omega)|^q \, d\mu(\omega), \int_{\Omega} |f_j(\omega)|^q \, d\mu(\omega) \right\} \\
&\leq \max\{1, r^q\} := c.
\end{aligned}
$$

We conclude that $\mathrm{D}_{(A,f)}(v_1, \ldots, v_k) \in \mathcal{P}_{q,c}(\mathbb{R}^{2k+d})$, therefore $\mathrm{S}_k(A, f) \subset \mathcal{P}_{q,c}(\mathbb{R}^{2k+d})$, thus completing the proof. $\square$

**Corollary H.17.** *Let $(A, f) \in \mathcal{B}_{\infty,\infty,d}^r(\Omega)$ be a P-operator-signal. Then, for every $q \geq 1$, $\mathrm{S}_k(A, f) \subset \mathcal{P}_{q,c}(\mathbb{R}^{2k+d})$, where $c = \max\{1, r^q\}$.*

Let $\overline{\mathrm{S}_k(A, f)}$ be the closure of $\mathrm{S}_k(A, f)$ in $(\mathcal{P}_{q,c}(\mathbb{R}^{2k+d}), W)$. By theorem G.7, the space $(\mathcal{P}_{q,c}(\mathbb{R}^{2k+d}), W)$ is compact for every $q > 1$. Consequently, as a closed subset of a compact space, $\overline{\mathrm{S}_k(A, f)}$ is itself compact. Recall from section B.3 the distance between a set and its closure is zero. We therefore identify the profile with its closure. By abuse of notation, we write $\mathrm{S}_k(A, f)$ to denote the compact set $\overline{\mathrm{S}_k(A, f)}$.

We conclude this section with an important result. We prove that the space of all profiles, equipped with the Hausdorff distance, is compact. We then use the Tychonoff theorem to conclude that the space of P-operator-signals is a compact space as well A.3.

**Theorem H.18.** *For $k, d \geq 0$ $(p, q) \in [1, \infty] \times (1, \infty]$ and $r > 0$, let $(\boldsymbol{S}^r_{k,d,p,q}, d_H)$ be the space of all $k$-profiles of P-operator-signals with uniformly bounded $(p, q)$ norm, where $r$ denotes the uniform bound. Then, the space $(\boldsymbol{S}^r_{k,d,p,q}, d_H)$ is compact.*

*Proof.* By theorem B.3, the space of all closed subsets of a compact metric space, metrized by the Hausdorff distance, is compact. Since $k$-profiles are compact sets, to prove that $\boldsymbol{S}^r_{k,d,p,q}$ is compact, it suffices to show that it is a closed set. Let $\{S_n\}_{n=1}^{\infty}$ be a sequence of profiles in $\boldsymbol{S}^r_{k,d,p,q}$ that converges to some limit set $S$ with respect to $d_H$. This convergence implies that the underlying sequence of P-operator-signals is a Cauchy sequence. By Theorem H.15, there exists a limit P-operator-signal $(A, f) \in \mathcal{B}^r_{p,q,d}$ such that its profile is exactly $S$. Thus, $S \in \boldsymbol{S}^r_{k,d,p,q}$, proving that the space is closed. As a closed subset of a compact space, $(\boldsymbol{S}^r_{k,d,p,q}, d_H)$ is compact. $\qquad\square$

By Tychonoff's theorem, we can directly derive compactness of the space of P-operator-signals with the metric $d_M$.

**Corollary H.19.** *For $k, d \geq 0$ $(p, q) \in [1, \infty] \times (1, \infty]$ and $r > 0$, let $(\mathcal{B}^r_{p,q,d}, d_M)$ be the space of all P-operator-signals with uniformly bounded $(p, q)$ norm, where $r$ denotes the uniform bound. Then the space $(\mathcal{B}^r_{p,q,d}, d_M)$ is compact metric space and $\boldsymbol{S}^r_{d,p,q}$ is compact in the product topology.*

*Proof.* We start the proof by proving that $\boldsymbol{S}^r_{d,p,q}$ is compact in the product topology. Let $\mathcal{K}$ be the Cartesian product of the spaces of $k$-profiles of any order $k \geq 0$,

$$\mathcal{K} := \prod_{k=0}^{\infty} \boldsymbol{S}^r_{k,d,p,q}.$$

By Theorem H.18, for every $k \geq 0$ the space $(\boldsymbol{S}^r_{k,d,p,q}, d_H)$ is compact, so by Tychonoff's theorem, we obtain that $\mathcal{K}$ is compact with respect to the product topology.

Recall that the profile of P-operator-signal is the sequence of $k$-profile of any order $k$. Namely, if $(A, f) \in \mathcal{B}^r_{p,q,d}$, then $\mathrm{S}(A, f) = (\mathrm{S}_k(A, f))_{k=0}^{\infty}$. Since for every $k$, $\mathrm{S}_k(A, f) \in \boldsymbol{S}^r_{k,d,p,q}$ we have that $\boldsymbol{S}^r_{d,p,q} \subset \mathcal{K}$, (recall Equation (14)). Therefore, to prove that $\boldsymbol{S}^r_{d,p,q}$ is compact, it sufficient to prove that it is closed. Let $S^{(n)}$ be a sequence of profiles in $\boldsymbol{S}^r_{d,p,q}$ converging to a set $S$. By definition, for every $n$, there exists a P-operator-signal $(A_n, f_n)$ such that $S^{(n)} = \mathrm{S}(A_n, f_n)$. Since convergence of profiles implies convergence of $k$-profiles for every $k \geq 0$, and this equivalent to action convergence by definition, (Definition H.13) we have that the sequence $(A_n, f_n)$ is action convergent. By the existence of limit object (Theorem H.15), there exists a P-operator-signal $(A, f)$ such that $\lim_{n \to \infty} d_M((A_n, f_n), (A, f)) = 0$, and therefore $S = \mathrm{S}(A, f)$. We obtain that $S \in \boldsymbol{S}^r_{d,p,q}$ and $\boldsymbol{S}^r_{d,p,q}$ is compact.

Since the action metric is a weighted countable sum of $k$-profiles, we can identify the space $(\mathcal{B}^r_{p,q,d}, d_M)$ with the space $\boldsymbol{S}^r_{d,p,q}$ using the map that sends a P-operator-signal to its profile, we obtain that $(\mathcal{B}^r_{p,q,d}, d_M)$ is compact.

$\qquad\square$

### H.3. Compactness of the Space of Graphop-Signals

In this section we prove compactness of the space of graphop-signals. Recall that a graphop is a self-adjoint and positively preserving P-operator. The idea of the proof is the following. Since the space of P-operator-signals is compact as we proved in the previous section, we will prove that the space of graphop-signals is closed. We show that if a sequence of graphop-signals converge to some P-operator-signal, then the limit object is itself a graphop-signal. The proof is divided into two parts: first, we show that an action limit of self adjoint P-operators is a self adjoint P-operator. Second, we show that an action limit of positively preserving P-operators is positively preserving. Note that similar results were proven in (Backhausz & Szegedy, 2022; Hrušková, 2022). However, since our setting is a bit different, and the original proofs did not include the signal part, we propose a different proof technique that involves methods from optimal transport, but also include the signal.

**Theorem H.20.** *Let $p \in [1, \infty]$, $q \in (1, \infty]$, $d \in \mathbb{N}$ and $r > 0$. Let $\{(A_n, f_n) \in \mathscr{G}^r_{p,q}(\Omega_n)\}_{n \in \mathbb{N}}$ be a sequence of graphop-signals that action converges to a P-operator-signal $(A, f) \in \mathcal{B}^r_{p,q,d}(\Omega)$. Then $(A, f) \in \mathscr{G}^r_{p,q,d}(\Omega)$. That is, an action limit of graphops is itself a graphop.*

*Proof.* We divide the proof into two parts. First, we prove that an action limit of a sequence of self adjoint P-operators, is itself self adjoint.

Let $u_1, u_2 : \Omega \to [-1, 1]$, and denote by $\nu := D_{(A,f)}(u_1, u_2)$. We will prove that

$$\int_\Omega u_1(\omega) \cdot A u_2(\omega) \, d\mu = \int_\Omega u_2(\omega) \cdot A u_1(\omega) \, d\mu,$$

or equivalently

$$\int_{\mathbb{R}^{4+d}} x_1 \cdot x_4 \, d\nu(\vec{x}) = \int_{\mathbb{R}^{4+d}} x_2 \cdot x_3 \, d\nu(\vec{x}),$$

where $\vec{x} = (x_j)_{j \in [4+d]}$.

By the action convergence of $(A_n, f_n)$ to $(A, f)$, there exists a sequence of measures $\nu_n := D_{(A_n, f_n)}(w^{(n)}, u^{(n)})$, where $w^{(n)}, u^{(n)} : \Omega_n \to [-1, 1]$, such that $W(\nu_n, \nu) \to 0$. Since for every $n \in \mathbb{N}$, $A_n$ is self adjoint, we have that

$$\int_{\Omega_n} w^{(n)} \cdot A_n u^{(n)} \, d\mu_n = \int_{\Omega_n} u^{(n)} \cdot A_n w^{(n)} \, d\mu_n,$$

or equivalently,

$$\int_{\mathbb{R}^{4+d}} x_1 \cdot x_4 \, d\nu_n(\vec{x}) = \int_{\mathbb{R}^{4+d}} x_2 \cdot x_3 \, d\nu_n(\vec{x}). \tag{15}$$

Next, we prove that

$$\lim_{n \to \infty} \int_{\mathbb{R}^{4+d}} x_2 \cdot x_3 \, d\nu_n = \int_{\mathbb{R}^{4+d}} x_2 \cdot x_3 \, d\nu.$$

If $q = \infty$, the support of $\nu_n$ is compact, and therefore the limit holds by weak convergence. If $q < \infty$, we can not directly use weak convergence, as the function $\vec{x} \mapsto x_2 \cdot x_3$ is not bounded, since on the support of $\nu_n$, the coordinate $x_3$ corresponds to $A_n w^{(n)}$ which is not necessarily bounded.

Let $(\gamma_n)_{n=1}^\infty$ be a sequence of the optimal couplings of $\nu_n$ and $\nu$ realizing the Wasserstein distance. By the convergence of $\nu_n$ to $\nu$ we have that

$$\int_{\mathbb{R}^{4+d} \times \mathbb{R}^{4+d}} \|\vec{x} - \vec{y}\|_1 \, d\gamma_n \to 0.$$

Consider the difference

$$\left| \int x_2 \cdot x_3 \, d\nu_n - \int y_2 \cdot y_3 \, d\nu \right| = \left| \int (x_2 \cdot x_3 - y_2 \cdot y_3) \, d\gamma_n \right| \leq \int \left| x_2 \cdot x_3 - y_2 \cdot y_3 \right| d\gamma_n.$$

By the triangle inequality we have that

$$\left| \int x_2 \cdot x_3 \, d\nu_n - \int y_2 \cdot y_3 \, d\nu \right| \leq \int |x_2| \cdot |x_3 - y_3| \, d\gamma_n + \int |y_3| \cdot |x_2 - y_2| \, d\gamma_n.$$

Consider the first term. Since the support of every coupling is a subset of the Cartesian product of the two supports of the measures, we have that $|x_2| \leq 1$, and hence

$$\int |x_2| \cdot |x_3 - y_3| \, d\gamma_n \leq \int |x_3 - y_3| \, d\gamma_n,$$

which clearly tends to $0$ as $n$ tends to infinity. For the second term, we use Hölder inequality with exponents $q$ and $q' = \frac{q}{q-1}$ and get

$$\int |x_y| \cdot |x_2 - y_2| \, d\gamma_n \leq \left( \int |y_3|^q \, d\gamma_n \right)^{1/q} \cdot \left( \int |x_2 - y_2|^{q'} \, d\gamma_n \right)^{1/q'}.$$

Again, since the support of $\gamma_n$ is a subset of the Cartesian product of the supports of $\nu_n$ and $\nu$, we have that $|x_2 - y_2|$ is bounded, and therefore converges to $0$. The convergence holds because

$$\int |y_3|^q \, d\gamma_n = \int |A_n v_n|^q \, d\nu_n \leq r^q.$$

We proved that

$$\lim_{n \to \infty} \int_{\mathbb{R}^{4+d}} x_2 \cdot x_3 \, d\nu_n = \int_{\mathbb{R}^{4+d}} x_2 \cdot x_3 \, d\nu,$$

and similarly we can prove that

$$\lim_{n \to \infty} \int_{\mathbb{R}^{4+d}} x_2 \cdot x_3 \, d\nu_n = \int_{\mathbb{R}^{4+d}} x_2 \cdot x_3 \, d\nu,$$

using that $A_n$ are self adjoint finish this part of the proof.

We move to prove Positively preserving of the limit. Let $w : \Omega \to [0,1]$. We will prove that $\nu := \mathrm{D}_{(A,f)}(w)$ is supported on $[0,1]^2$. By the action convergence of $(A_n, f_n)$ to $(A, f)$, there exists a sequence of measures $\nu_n := \mathrm{D}_{(A_n,f_n)}(w_n)$, where $w_n : \Omega_n \to [-1,1]$, such that $W(\nu_n, \nu) \to 0$. Let $\nu'_n := \mathrm{D}_{(A_n,f_n)}(|w_n|)$. By lemma J.3, we have that

$$W(\nu'_n, \nu) \le W(\nu_n, \nu) \to 0,$$

since $\nu'_n = |\cdot|_* \nu_n$ where $|\cdot|$ is the function that puts absolute value on the first and second coordinates.

Since convergence of measures in $\mathcal{P}_{q,r}$ in Wasserstein distance is equivalent to the weak convergence, we get that if the supports of $\nu_n$ are subsets of some closed set, which in our case is $[0,1]^2$, then the support of the limit $\nu$ is also a subset of $[0,1]$, which implies that $A$ is positively preserving. $\qquad\square$

Combination of Theorem H.20 and Corollary H.19, gives us compactness of the space graphop-signals with respect to the action metric.

**Corollary H.21.** *For $k, d \ge 0$ $(p,q) \in [1, \infty] \times (1, \infty]$ and $r > 0$, let $(\mathscr{G}^r_{p,q,d}, d_M)$ be the space of all graphop-signals with uniformly bounded $(p,q)$ norm, where $r$ denotes the uniform bound. Then the space $(\mathscr{G}^r_{p,q,d}, d_M)$ is compact. In particular, we have that $(\mathcal{BF}^r_d, d_M)$ is compact.*

# I. MPNNs on P-Operator-Signals and Profiles

We now formally define the MPNN architecture for P-operator-signals, generalizing standard message passing on graphs and message passing on graphons (Levie, 2024).

## I.1. MPNNs on P-Operator-Signals

Consider an $L$-layer MPNN model with readout $(\varphi, \psi)$ (recall definition C.1). We now define how an MPNN model processes a P-operator-signal.

**Definition I.1** (MPNN On P-operator-signal). Let $(\varphi, \psi)$ be a MPNN model with readout, and $(A, f)$ a P-operator-signal, where $f : \Omega \to [-1,1]^d$. The application of the MPNN on $(A, f)$ is defined as follows: initialize $\mathfrak{h}_-^{(0)} := \varphi^{(0}(f(-))$ and compute the hidden node representations $\mathfrak{h}_-^{(l)} : \Omega \to [-1,1]^{d_l}$ at layer $l$, with $1 \le l \le L$ and the P-operator level output $\mathfrak{H}_{(A,f)} \in \mathbb{R}^p$ by

$$\mathfrak{h}_x^{(l)} := \varphi^{(l)}(\mathfrak{h}_x^{(l-1)}, A\mathfrak{h}_-^{(l-1)}(x)),$$

and

$$\mathfrak{H}(A, f) := \psi\Big( \int_\Omega \mathfrak{h}_x^{(L)} \, d\mu(x) \Big).$$

From the definition, we see that aggregation in MPNN on P-operator-signals is performed by applying the P-operator to the signal, coordinate-wise along the feature dimension.

We often denote by $\mathfrak{h}^{(l)}_{\varphi,(A,f)} : \Omega \to [-1,1]^{d_l}$ the signal obtained after $l$ message passing layers on the P-operator-signal $(A, f)$.

## I.2. MPNNs Only See Profiles of P-Operator-Signals

In this section, we introduce two equivalent approaches for constructing MPNNs on profiles: one using a representative $(A, f)$ of the profile, and the other directly on the profile without referencing a specific representative. Since different

P-operator-signals can produce the same profile, the application of MPNNs on profiles should not depend on the choice of representative. Consequently, the information encoded in the P-operator-signal becomes redundant, and we only need to consider the profile itself.

Consider the equivalence relation: $(A_1, f_1) \sim (A_2, f_2)$ if there is $k \in \mathbb{N}$ such that $\mathrm{S}_k(A_1, f_1) = \mathrm{S}_k(A_2, f_2)$. Since our goal is to apply MPNN directly on profiles - without reference to a specific representative of the equivalence class, we sometimes abuse notation and write $\mathrm{S}_{k,d} :=$ instead of $\mathrm{S}_k(A, f)$ where $d \in \mathbb{N}$ is an indication of the feature dimension, and $(A, f)$ is a representative of the equivalence class $[A, f] := \{(A', f') \mid (A', f') \sim (A, f)\}$. We further define the class of all P-operator-signals that induce a given profile $\mathrm{S}_{k,d}$ by $[\mathrm{S}_{k,d}] := \{(A, f) \mid S_k(A, f) = \mathrm{S}_{k,d}\}$. We call $(A, f) \in [\mathrm{S}_{k,d}]$ a representative inducer of $\mathrm{S}_{k,d}$, or representative in short.

### I.2.1. OPERATIONS ON PROFILES

Next, we introduce two important operations on $k$-profiles that will be used later to apply MPNN on $k$-profiles. Our operations apply to profiles with no specific reference to a representative.

Our first operation is called *pushforward on the signal*. We will show that pushforward on the signal applied on a $k$-profile yields another $k$-profile, in which the signal is given by the composition of the original signal with the map along which the pushforward is taken. This operation will later be used to apply the update function to a profile, producing the signal at the next layer.

**Definition I.2** (Pushforward On The Signal). Let $\mathrm{S}_{k,d}$ be a $k$-profile, and let $\phi : [-1, 1]^d \to [-1, 1]^p$ for some $d, p \in \mathbb{N}$. For a P-distribution $\nu \in \mathrm{S}_{k,d}$ we define the *pushforward on the signal* as the pushforward of $\nu$ via the map

$$(x_1 \ldots, x_{2k+d}) \mapsto (x_1 \ldots, x_{2k}, \phi(x_{2k+1}, \ldots, x_{2k+d})),$$

and denote it by $\phi \tilde{*} \nu$. Denote by $\phi \tilde{*} \mathrm{S}_{k,d} := \{\phi \tilde{*} \nu \mid \nu \in \mathrm{S}_{k,d}\}$ the set of all Borel probability measures of the form $\phi \tilde{*} \nu$ for $\nu \in \mathrm{S}_{k,d}$.

Note that the notation $\tilde{*}$ is used for both P-distribution and $k$-profiles, and the meaning will be clear from context. A pushforward on the signal refers to the standard pushforward operation applied solely to the last coordinate of the P-distributions encoding the signal, which motivates the terminology.

*Remark* I.3 (Pushforward on the Signal gives Profile). Let $(A, f)$ be a representative of the profile $\mathrm{S}_{k,d}$, i.e., $(A, f) \in [\mathrm{S}_{k,d}]$. By our definition, we have

$$\phi \tilde{*} \mathrm{D}_{(A,f)}(v_1, \ldots, v_k) = \mathrm{D}_{(A,\phi(f))}(v_1 \ldots, v_k),$$

and

$$\phi \tilde{*} \mathrm{S}_k(A, f) = \mathrm{S}_k(A, \phi(f)),$$

hence, the set of measures $\phi \tilde{*} \mathrm{S}_k(A, f)$ is a profile.

In the following lemma, we prove that regardless of which representative of the equivalence class is chosen, applying a function on the signal and inducing the new profile is independent of the original representative.

**Lemma I.4.** *Let* $\mathrm{S}_k(A, f) = \mathrm{S}_k(A', f')$ *be k-profiles of P-operator-signals* $(A, f)$ *and* $(A', f')$ *over probability spaces* $(\Omega, \mu)$ *and* $(\Omega', \mu')$ *respectively. Let* $\phi : [-1, 1]^d \to [-1, 1]^p$ . *Then,*

$$\mathrm{S}_k(A, \phi(f)) = \mathrm{S}_k(A', \phi(f')),$$

*or equivalently*

$$\phi \tilde{*} \mathrm{S}_k(A, f) = \phi \tilde{*} \mathrm{S}_k(A', f').$$

*Proof.* For $x \in \Omega$, let $v(x) = \big(v_1(x), \ldots, v_k(x)\big)$, and for $x' \in \Omega'$ let $v'(x') = \big(v'_1(x'), \ldots, v'_k(x')\big)$. By the equality of the profiles, for every P-distribution $\mathrm{D}_{(A,f)}(v) \in \mathrm{S}_k(A, f)$ there is a tuple $v' = (v'_1 \ldots, v'_k)$ and a corresponding P-distribution $\mathrm{D}_{(A',f')}(v') \in \mathrm{S}_k(A', f')$, such that $\mathrm{D}_{(A,f)}(v) = \mathrm{D}_{(A',f')}(v')$ .

Then for every measurable set $E \subset \mathbb{R}^{2k+p}$,

$$\mathrm{D}_{(A,\phi(f))}(v)(E) = \mu(\{\, x \in \Omega \mid \big(v(x), Av(x), \phi(f(x))\big) \in E \,\}).$$

Denote
$$E_{\phi^{-1}} = \{(y_1 \in [-1,1]^k, y_2 \in \mathbb{R}^k, y_3 \in [-1,1]^d) \mid (y_1, y_2, \phi(y_3)) \in E\}.$$

So $\{x \in \Omega \mid (v(x), Av(x), \phi(f(x))) \in E\}$ is the set of all $x \in \Omega$ such that there exist $(y_1 \in [-1,1]^k, y_2 \in [-1,1]^k, y_3 \in [-1,1]^d)$ satisfying $(v(x), Av(x), f(x)) = (y_1, y_2, y_3)$ with $(y_1, y_2, \phi(y_3)) \in E$. Thus

$$\mathrm{D}_{(A,\phi(f))}(v)(E) = \mu(\{x \in \Omega \mid (v(x), Av(x), \phi(f(x))) \in E\}) = \mathrm{D}_{(A,f)}(v)(E_{\phi^{-1}}),$$

and by the above equality of the P-distributions

$$\begin{aligned}
\mathrm{D}_{(A,f)}(v)(E_{\phi^{-1}}) &= D_{(A',f')}(v')(E_{\phi^{-1}}) \\
&= \mu(\{x \in \Omega \mid (v'(x), Av'(x), \phi(f'(x))) \in E\}) = \mathrm{D}_{(A',\phi(f'))}(v')(E).
\end{aligned}$$

$\square$

Our next operation is called *diagonal marginalization*. We show that diagonal marginalization applied to a profile yields another profile that captures the aggregation of the corresponding P-operator-signal. This operation consists of first restricting the profile to P-distributions supported on an appropriately defined diagonal set, and then marginalizing these diagonal measures. Diagonal marginalization will later be used to implement the aggregation step in MPNN on profiles.

**Definition I.5** (Restricted profile). Let $\mathrm{S}_{k,d}$ be a $k$-profile, and let $E \subset \mathbb{R}^{2k+d}$ be a measurable set. We define the *restricted profile* and write $\mathrm{S}_{k,d}^E$ as the set of all P-distributions in $\mathrm{S}_{k,d}$ that are supported on $E$. Explicitly

$$\mathrm{S}_{k,d}^E = \{\nu \in \mathrm{S}_{k,d} \mid \nu(F) = \nu(F \cap E) \text{ for every measurable } F \subset \mathbb{R}^{2k+d}\}.$$

**Definition I.6** (Aggregated Profile). Let $\mathrm{S}_{k,d}$ be a $k$-profile, with $k \geq d$. Define:

1. **Diagonal Subset.** The *diagonal subset* $T_d \subseteq \mathbb{R}^{2k+d}$ is defined to be

$$\begin{aligned}
T_d := \Big\{ (y_1 &\in [-1,1]^{k-d}, y_2 \in [-1,1]^d, \\
&y_3 \in \mathbb{R}^{k-d}, y_4 \in \mathbb{R}^d, y_5 \in [-1,1]^d) \,\Big|\, y_2 = y_5 \Big\}.
\end{aligned}$$

2. **Diagonally Restricted Profile.**

    Following definition I.5, we define the *diagonally restricted profile* to be $\mathrm{S}_{k,d}^{T_d}$. Each $\nu \in \mathrm{S}_{k,d}^{T_d}$ is called *diagonal P-distribution*.

3. **Diagonal Marginalization.** Let $\nu \in \mathrm{S}_{k,d}^{T_d}$ be a diagonal P-distribution. The *diagonal marginalization* of $\nu$, denoted by $\mathrm{DM}_d(\nu)$, is the Borel measure on $\mathbb{R}^{2k}$ obtained by marginalizing out the coordinates $y_2$. Formally,

$$\mathrm{DM}_d(\nu) := p_* \nu,$$

    where $p : \mathbb{R}^{2k+d} \to \mathbb{R}^{2k}$ is the projection

$$p : (y_1, y_2, y_3, y_4, y_5) \mapsto (y_1, y_3, y_4, y_5).$$

4. **Aggregated Profile.** The *aggregated profile* $\mathrm{DM}_d(\mathrm{S}_{k,d})$ is the collection of all diagonal marginalizations of diagonal P-distributions in $\mathrm{S}_{k,d}^{T_d}$. Explicitly,

$$\mathrm{DM}_d(\mathrm{S}_{k,d}) := \Big\{ \mathrm{DM}_d(\nu) \,\Big|\, \nu \in \mathrm{S}_{k,d}^{T_d} \Big\}.$$

In Lemma I.7, we show that $\mathrm{DM}_d(\mathrm{S}_{k,d})$ is indeed a $(k-d)$-profile of a P-operator-signal whenever $\mathrm{S}_{k,d}$ is, thus justifying the name. Moreover, we prove that applying aggregation to a P-operator-signal, which is a representative of some profile, and then inducing the corresponding profile from the aggregated P-operator-signal, is independent of the representative. Moreover, the lemma shows that diagonal marginalization is equivalent to aggregation on P-operator-signal.

**Lemma I.7.** *Let* $S_k(A, f) = S_k(A', f')$ *be a k-profile induced by the two P-operator-signals* $(A, f), (A', f')$, *where* $f$ *and* $f'$ *are two* $d$ *channel signals. Consider the aggregation operation* $\mathrm{Agg}$ *on the P-operator-signal* $(A, f)$ *defined by*

$$\mathrm{Agg}\big((A, f)\big) = \big(A, (Af, f)\big).$$

*Then*

$$S_{k-d}(\mathrm{Agg}(A, f)) = S_{k-d}(\mathrm{Agg}(A', f')),$$

*and*

$$S_{k-d}(\mathrm{Agg}(A, f)) = \mathrm{DM}_d(S_k(A, f)).$$

*Proof.* For every $D_{(A,f)}(v_1, \ldots, v_k) \in S_k(A, f)$ denote $v_f = (v_1, \ldots, v_{k-d}, f_1, \ldots, f_d)$. and similarly for $D_{(A',f')}(v'_1, \ldots, v'_k) \in S_k(A', f')$ denote $v'_{f'} = (v'_1, \ldots, v'_{k-d}, f'_1, \ldots, f'_d)$.

For every measurable set $E \subset \mathbb{R}^{2k+d}$, we have

$$\{x \in \Omega \mid \big(v_f(x), Av_f(x), f(x)\big) \in E\} = \{x \in \Omega \mid \big(v_f(x), Av_f(x), f(x)\big) \in E \cap T_d\},$$

where $T_d$ is the diagonal set defined in definition I.6 . Recall that $S_k^{T_d}(A, f)$ is the set of all P-distributions in $S_k(A, f)$ that are supported on the diagonal, and $S_k^{T_d}(A', f')$ is defined similarly. Since $S_k(A, f) = S_k(A', f')$, we have that $S_k^{T_d}(A, f) = S_k^{T_d}(A', f')$. Next, we prove that every P-distribution in $S_k^{T_d}(A, f)$ is of the form $D_{(A,f)}(u_1, \ldots, u_{k-d}, f_1, \ldots, f_d)$. Indeed, let $D_{(A,f)}(u_1, \ldots, u_k) \in S_k^{T_d}(A, f)$. For the complement of the diagonal set, $T_d^c$, we have

$$\begin{aligned} 0 &= D_{(A,f)}(u_1, \ldots, u_k)(T_d^c) \\ &= \mu\big(\{\, x \in \Omega \mid \big(u_1(x), \ldots, u_k(x), Au_1(x), \ldots, Au_k(x), f_1(x), \ldots, f_d(x)\big) \in T_d^c \,\}\big) \\ &= \mu\big(\{\, x \in \Omega \mid \exists j \in [d], \ u_{k-d+j}(x) \neq f_j(x) \,\}\big). \end{aligned}$$

Therefore, for every $j \in [d]$, and a.e. $x \in \Omega$, $u_{k-d+j}(x) = f_j(x)$. Hence, we can write

$$S_k^{T_d}(A, f) = \{\, D_{(A,f)}(u_1, \ldots, u_{k-d}, f_1, \ldots, f_d) \mid (u_1, \ldots, u_{k-d}) : \Omega \to [-1, 1]^{k-d} \,\},$$

where $u_1, \ldots, u_{k-d}$ are measurable. Similarly,

$$S_k^{T_d}(A', f') = \{\, D_{(A',f')}(u'_1, \ldots, u'_{k-d}, f'_1, \ldots, f'_d) \mid (u'_1, \ldots, u'_{k-d}) : \Omega' \to [-1, 1]^{k-d} \,\}.$$

Note that since $S_k(A, f) = S_k(A', f')$, the diagonal restriction is also identical $S_k^{T_d}(A, f) = S_k^{T_d}(A', f')$. By this equality, we can marginalize the P-distributions, retaining the equality

$$\begin{aligned} &\{\, D(u_1, \ldots, u_{k-d}, Au_1, \ldots, Au_{k-d}, Af_1, \ldots, Af_d, f_1, \ldots, f_d) \mid u_i : \Omega \to [-1, 1] \,\} = \\ &\{\, D(u'_1, \ldots, u'_{k-d}, Au'_1, \ldots, Au'_{k-d}, A'f'_1, \ldots, A'f'_d, f'_1, \ldots, f'_d) \mid u'_i : \Omega' \to [-1, 1] \,\}, \end{aligned}$$

or equivalently

$$S_{k-d}(A, (Af, f)) = S_{k-d}(A', (A'f', f')).$$

For the second part of the proof we have

$$\begin{aligned} \mathrm{DM}_d(S_k(A, f) &= \{\, \mathrm{DM}_d(D_{(A,f)}(v_1, \ldots, v_k) \mid D_{(A,f)}(v_1, \ldots, v_k) \in S_k^{T_d}(A, f) \,\} \\ &= \{\mathrm{DM}_d(D_{(A,f)}(v_1, \ldots, v_{k-d}, f_1, \ldots, f_d) \mid v_j : \Omega \to [-1, 1] \, \forall j \in [k - d] \,\} \\ &= \{p_* D_{(A,f)}(v_1, \ldots, v_{k-d}, f_1, \ldots, f_d) \mid v_j : \Omega \to [-1, 1] \, \forall j \in [k - d] \,\} \\ &= \{D_{(A,(Af,f))}(v_1, \ldots, v_{k-d}) \mid v_k : \Omega \to [-1, 1] \, \forall j \in [k - d] \,\} \\ &= S_{k-d}(A, (Af, f)) = S_{k-d}(\mathrm{Agg}(A, f)). \end{aligned}$$

$\square$

I.2.2. MPNNs ON PROFILES

In this section, we define the application of MPNNs on profiles using the above operations, i.e., pushforward on the signal and diagonal marginalization.

Recall from definition C.1 that a MPNN model with readout is a tuple $(\varphi, \psi)$ where $\varphi$ is a collection $(\varphi^{(l)})_{l=0}^{L}$ of Lipschtz continuous functions

$$\varphi^{(0)} : \mathbb{R}^d \to \mathbb{R}^{d_0}, \qquad \varphi^{(l)} : \mathbb{R}^{2d_{l-1}} \to \mathbb{R}^{d_l}, \quad 1 \le l \le L,$$

which are called update functions and $\psi : \mathbb{R}^{d_L} \to \mathbb{R}^p$, is a Lipschitz continuous function called the readout function.

**Definition I.8** (MPNN on $k$-Profiles). Let $(\varphi, \psi)$ be an $L$-layer MPNN model with readout, with hidden feature dimensions $d_0, \ldots, d_L \in \mathbb{N}$ and $p$ the output feature dimension. For $k \ge \sum_{j=0}^{L-1} d_j$, let $\mathrm{S}_{k,d}$ be a $k$-profile with input feature dimension $d \in \mathbb{N}$. The MPNN on $\mathrm{S}_{k,d}$ is defined as follows. Initialize $\mathfrak{s}_{k_0,d_0}^{(0)} := \varphi^{(0)} \tilde{\ast} \mathrm{S}_{k,d}$ (where $k_0 := k$) and compute the hidden profiles $\mathfrak{s}_{k_l,d_l}^{(l)}$ at layer $l$, with $1 \le l \le L$, by

$$\mathfrak{s}_{k_l,d_l}^{(l)} = \varphi^{(l)} \tilde{\ast} \mathrm{DM}_{d_{l-1}}(\mathfrak{s}_{k_{l-1},d_{l-1}}^{(l-1)}).$$

Let $p_{d_L}$ be the projection to the last $d_L$ coordinates. Then the profile level output $\mathfrak{S}$ is

$$\mathfrak{S}(\mathrm{S}_{k,d}) := \psi\Big( \int_{[-1,1]^{d_L}} \vec{x} \, d\nu(\vec{x}) \Big),$$

where $\nu$ is the unique P-distribution that remains in the last layer's profile $p_{d_L} \tilde{\ast} \mathfrak{s}_{k_L, d_L} = \{\nu\}$.

Since all $\mathfrak{s}_{k_l,d_l}^{(l)}$ are profiles, we abuse notation and when considering a representative to the input profile $\mathrm{S}_k(A, f)$, we will often denote by $\mathfrak{s}_{k_l,d_l}^{\varphi}(A, f)$ or $\mathfrak{s}_{k_l,d_l}(A, f)$ the $k_l$ profile at layer $l$, and by $\mathfrak{S}^{\varphi}(A, f)$ the profile level output.

## I.3. Commutation Property of MPNNs on P-Operator-Signals and Profiles

To conclude this section, we prove that the profile that is obtained by one message passing layer (MPL) of an MPNN on a profile $S_k(A, f)$ is equal to the profile of the P-operator-signal that is obtained from one MPL on the P-operator-signal $(A, f)$ that induces the profile $S_k(A, f)$. Before proving this important property, we give an example of a simple single-layer MPNN model to demonstrate this commutation property between MPNN on P-operator-signal and MPNN on its $k$-profile. The full formulation of this idea is given below in Theorem I.9.

Let $(\varphi, \psi)$ be a 1-layer MPNN model with readout, that is, we have an initialization function $\varphi^{(0)} : [-1, 1]^d \to [-1, 1]^{d_0}$, update function $\varphi^{(1)} : [-1, 1]^{d_0} \to [-1, 1]^{d_1}$ and a readout function $\psi : [-1, 1]^{d_1} \to [-1, 1]^p$. Let $(A, f) \in \mathcal{B}_d(\Omega)$ be a P-operator-signal with $k$-profile $\mathrm{S}_k(A, f)$, when we take $k > d_0$, specifically to demonstrate the importance of the projection in the last layer. Each layer will be implemented both on the P-operator-signal, and on the corresponding profile, thus showing the commutation property in practice.

**Initialization:** Let $\varphi : [-1, 1]^d \to [-1, 1]^{d_0}$. By definition I.1, we have $\mathfrak{h}^{(0)} = \varphi^{(0)} \circ f : \Omega \to [-1, 1]^{d_0}$, and the P-operator-signal $(A, \mathfrak{h}^{(0)}) \in \mathcal{B}_d(\Omega)$. For the $k$-profile $\mathrm{S}_k(A, f)$, initialization gives $\mathfrak{s}_{k,d_0}^{(0)} := \varphi^{(0)} \tilde{\ast} \mathrm{S}_k(A, f)$. We employ remark I.3, to have the initialized $k$-profile $\mathrm{S}_k(A, \varphi^{(0)} \circ f) = \mathrm{S}_k(A, \mathfrak{h}^{(0)})$.

**Aggregation**: When considering P-operator-signals, recall that aggregation is performed by applying the P-operator on the signal coordinate-wise to obtain $A\mathfrak{h}^{(0)} : \Omega \to \mathbb{R}^{d_0}$. Concatenated with the original signal, we get the P-operator-signal $(A, (A\mathfrak{h}^{(0)}, \mathfrak{h}^{(0)})) \in \mathcal{B}_{2d_0}(\Omega)$. Aggregation with the case of $k$-profiles consists of two parts: restriction to the diagonal and marginalization. When restricting to the diagonal we get the $k$-profile $\mathrm{S}_k^{T_{d_0}}(A, \mathfrak{h}^{(0)})$ in which P-distributions of the form $\mathrm{D}_{(A, \mathfrak{h}^{(0)})}(v_1, \ldots, v_{k-d_0}, \mathfrak{h}_1^{(0)}, \ldots, \mathfrak{h}_{d_0}^{(0)})$. Now, after marginalization, we get with $k_0$-profile where $k_0 = k - d_0$, $\mathrm{DM}_{d_0}(\mathfrak{s}_{k,d_0}^{(0)}) = \mathrm{S}_{k_0}(A, (A\mathfrak{h}^{(0)}, \mathfrak{h}^{(0)}))$ in which P-distributions of the form

$$\mathrm{D}(v_1, \ldots, v_{k_0}, Av_1, \ldots, Av_{k_0}, A\mathfrak{h}_1^{(0)}, \ldots, A\mathfrak{h}_{d_0}^{(0)}, \mathfrak{h}_1^{(0)}, \ldots, \mathfrak{h}_{d_0}^{(0)}).$$

**Update**: Update is similar to initialization. For update function $\varphi^{(1)}$ we get the next-layer signal $\mathfrak{h}^{(1)} := \varphi^{(1)}(\mathfrak{h}^{(0)}, A\mathfrak{h}^{(0)}) : \Omega \to [-1, 1]^{d_1}$, thus having the P-operator-signal $(A, \mathfrak{h}^{(1)})$. For our aggregated profile $\mathrm{DM}_{d_0}(\mathfrak{s}_{k,d_0}^{(0)})$, we obtain

$$\mathfrak{s}_{k_0,d_1}^{(1)} = \varphi^{(1)} \tilde{\ast} \mathrm{DM}_{d_0}(\mathfrak{s}_{k,d_0}^{(0)}) = \mathrm{S}_{k_0}(A, \varphi^{(1)} \circ (A\mathfrak{h}^{(0)}, \mathfrak{h}^{(0)})) = \mathrm{S}_{k_0}(A, \mathfrak{h}^{(1)}).$$

**Readout**: Readout in MPNN on P-operator-signals is the global aggregation of the last-layer signal. In our case $\mathfrak{H}_{(A,f)} = \psi\left(\int_\Omega \mathfrak{h}_x^{(1)} \, d\mu(x)\right) \in [-1,1]^p$. The use of the projection in the readout in case of profiles is now comes to play. Recall that after update, the profile consists of P-distributions of the form $D_{(A,\mathfrak{h}^{(1)})}(v_1, \ldots, v_{k_0})$. If we chose originally $k = d_0$, would remain with one unique measure $D(\mathfrak{h}^{(1)})$, but since we assume that $k > d_0$, there are infinite number of P-distributions in $\mathfrak{s}_{k_0,d_1}^{(1)}$, since we want to integrate the signal alone, we pushforward against the projection to the last $d_1$ coordinates $p_{d_1}$ thus giving the unique measure $D(\mathfrak{h}^1)$. Then, by definition I.8, we obtain that the profile level output is

$$\mathfrak{S}(S_k(A,f)) = \psi\left(\int_{[-1,1]^{d_1}} \vec{x} \, dD(\mathfrak{h}^{(1)})(\vec{x})\right) = \psi\left(\int_\Omega \mathfrak{h}_x^{(1)} \, d\mu(x)\right).$$

This demonstration showcases that our two implementations on both P-operator-signals and profiles coincide. This Leads us directly to the following theorem that formulates and generalizes this idea of commutations.

Let $\mathcal{S}_k$ be the operator $(A,f) \mapsto S_k(A,f)$, i.e. the operator that sends a P-operator-signal to its $k$ profile.

**Theorem I.9.** *Let $(\varphi,\psi)$ be an L-layers MPNN model with readout. Let $(A,f) \in \mathcal{B}_d(\Omega)$ be a P-operator-signal with $k$-profile $S_k(A,f)$, where $k \geq \sum_{l=0}^{L-1} d_l$. Then for every $l \in [L]$,*

$$\mathfrak{s}_{k_l,d_l}^\varphi(A,f) = \mathcal{S}_{k_l}(A, \mathfrak{h}_{\varphi,(A,f)}^{(l)}),$$

*and*

$$\mathfrak{S}^\varphi(A,f) = \mathfrak{H}_{(A,f)}^\varphi.$$

*Proof.* We prove by induction.

*Induction base*: Recall that for $l = 0$, we have $k_0 = k$ and $\mathfrak{h}^{(0)} = \varphi \circ f$. So,

$$\mathfrak{s}_{k_0,d_0}^\varphi = \varphi^{(0)} \tilde{\ast} S_k(A,f) = S_k(A, \mathfrak{h}^{(0)}) = \mathcal{S}_k(A, \mathfrak{h}^{(0)}).$$

*Induction step*: Assume that for $0 \leq l \leq L-1$, $\mathfrak{s}_{k_l,d_l}^\varphi(A,f) = \mathcal{S}_{k_l}(A, \mathfrak{h}_{\varphi,(A,f)}^{(l)})$. Then

$$\begin{aligned}
\mathfrak{s}_{k_{l+1},d_{l+1}}^\varphi(A,f) &= \varphi^{(l+1)} \tilde{\ast} \mathrm{DM}_{d_l}(\mathfrak{s}_{k_l,d_l}^{(l)}(A,f)) = \\
&\quad \varphi^{(l+1)} \tilde{\ast} \mathrm{DM}_{d_l}(S_{k_l}(A, \mathfrak{h}_{\varphi,(A,f)}^{(l)})) = \\
&\quad \varphi^{(l+1)} \tilde{\ast} S_{k_l-d_l}(A, (A\mathfrak{h}_{\varphi,(A,f)}^{(l)}, \mathfrak{h}_{\varphi,(A,f)}^{(l)})) = \\
&\quad S_{k_{l+1}}(A, \varphi^{(l+1)} \circ (A\mathfrak{h}_{\varphi,(A,f)}^{(l)}, \mathfrak{h}_{\varphi,(A,f)}^{(l)})) = \\
&\quad S_{k_{l+1}}(A, \mathfrak{h}_{\varphi,(A,f)}^{(l+1)}) = \mathcal{S}_{k_{l+1}}((A, \mathfrak{h}_{\varphi,(A,f)}^{(l+1)})).
\end{aligned}$$

Let $\nu$ be the unique measure remaining in $p_{d_L} \tilde{\ast} \mathfrak{s}_{k_L,d_L}(A,f)$. Then

$$\mathfrak{S}^\varphi(A,f) = \psi\left(\int_{[-1,1]^{d_L}} \vec{x} \, d\nu(\vec{x})\right) = \psi\left(\int_\Omega \mathfrak{h}_{\varphi,(A,f)}^{(L)} \, d\mu\right) = \mathfrak{H}_{(A,f)}^\varphi$$

$\square$

Our commutative relation is summarized in figure 3.

We finish this section with the following corollary, in which we state that given a P-operator-signal, a MPNN on its profile is equivalent to the profiles of MPNN applied on P-operator-signal.

**Corollary I.10.** *Let $\mathcal{S}$ be the operator that maps a P-operator-signal to its profile. Namely, $\mathcal{S} : (A,f) \mapsto S(A,f)$, then for every $(A,f)$,*

$$(A, f) \xrightarrow{\varphi} (A, \mathfrak{h}_{\varphi,(A,f)}^{(L)}) \xrightarrow{\text{Readout } \psi} \mathfrak{H}_{(A,f)}^{\varphi}$$

$$\mathcal{S}_k \downarrow \qquad\qquad \downarrow \mathcal{S}_{k_L} \qquad\qquad \|$$

$$S_k(A, f) \xrightarrow{\varphi} \mathfrak{s}_{k_L, d_L}^{\varphi}(A, f) \xrightarrow{\text{Readout } \psi} \mathfrak{S}^{\varphi}(A, f)$$

*Figure 3.* Commutative diagram demonstrating the equivalence of MPNN operations on P-operator-signals (top) and their corresponding profiles (bottom), as established in Theorem I.9. The vertical bar on the right denotes the equality of the final outputs.

## J. Hölder Continuity of MPNNs With Respect To Action Metric

In this section, we prove one of our main contributions of the paper. Namely, we prove that an MPNN applied on P-operator-signals is Hölder continuous function with respect to the action metric $d_M$.

**Theorem J.1.** *For $(p, q) \in [1, \infty) \times [1, \infty]$ and $r > 0$, let $(A_1, f_1) \in \mathcal{B}_{p,q,d}^r(\Omega_1)$ and $(A_2, f_2) \in \mathcal{B}_{p,q,d}^r(\Omega_2)$ be P-operator-signals over Borel probability spaces $(\Omega_1, \mu_1)$ and $(\Omega_2, \mu_2)$ respectively. Let $\varphi$ be an L-layer MPNN model. Then, there exists a constant $C_\varphi$ that depends on the number of layers $L$, the Lipschitz constant of the update functions, $p, q$ and $r$ such that*

$$d_M((A_1, \mathfrak{h}_1^{(L)}), (A_2, \mathfrak{h}_2^{(L)})) \leq C_\varphi d_M((A_1, f_1), (A_2, f_2))^{\frac{1}{p^d}}.$$

*If $(\varphi, \psi)$ is an L-layer MPNN model with readout, then there exists a constant $C_{(\varphi,\psi)}$ that depends on $C_\varphi$ and the Lipschitz constant of the readout function $\psi$ such that*

$$\|\mathfrak{H}(A_1, f_1) - \mathfrak{H}(A_2, f_2)\|_2 \leq C_{(\varphi,\psi)} d_M((A_1, f_1), (A_2, f_2))^{\frac{1}{p^d}}.$$

We will prove theorem J.1 using our definition of MPNN on profiles (Definition I.8). Indeed, by the commutation property (Theorem I.9) we have that MPNN on P-operator-signals is equivalent to MPNNs on $k$-profiles.

**Theorem J.2.** *For $(p, q) \in [1, \infty) \times [1, \infty]$ and $r > 0$, let $(A_1, f_1) \in \mathcal{B}_{p,q,d}^r(\Omega_1)$ and $(A_2, f_2) \in \mathcal{B}_{p,q,d}^r(\Omega_2)$ be P-operator-signals with $k$-profiles $S_k(A_1, f_1)$ an $S_k(A_2, f_2)$ respectively, where $k \geq \sum_{l=0}^{L-1} d_l$. Let $(\varphi, \psi)$ be an L-layer MPNN model with readout. Then, there exists a constant $C_{(\varphi,\psi)}$ that depends on the number of layers, the Lipschitz constant of the update and readout functions, $p, q$ and $r$ such that*

$$\|\mathfrak{S}(A_1, f_1) - \mathfrak{S}(A_2, f_2)\|_2 \leq C_{(\varphi,\psi)} d_H(S_k(A_1, f_1), S_k(A_2, f_2))^{\frac{1}{p^d}}.$$

In the following sections, we prove Theorem J.2 by showing that the two operations, "Pushforward on the signal", and "Diagonal marginalization" are Hölder continuous with respect to the Hausdorff distance. After we establish the Hölder continuity with respect to the Hausdorff distance, we use the bound to deduce Hölder continuity with respect to the action metric.

### J.1. Pushforward Of The Signal Is Contractive

We begin the proof by showing that the pushforward on the signal operation is Lipschitz continuous with respect to the Hausdorff distance. Specifically, the map that sends a $k$-profile to the $k$-profile obtained by applying pushforward on the signal is Lipschitz continuous. Since the update and readout functions are Lipschitz continuous, it follows that pushforward on the signal inherits this Lipschitz continuity.

We start with a general lemma regarding Wasserstein distance between pushforward measures. We show that when pushing forward two probability measures via the same Lipschitz continuous map, the distance between the pushforward measures is bounded by the Wasserstein distance of the original measures, up to the Lipschitz constant. For the exact definition of the Wasserstein distance and general background on optimal transport see Appendix G.

**Lemma J.3.** *Let $(X, d_X), (Y, d_Y)$ be two metric spaces, and $f : X \to Y$ be a Lipschitz continuous function with Lipschitz constant $L_f$. Let $\mu$ and $\nu$ be Borel probability measures over $X$. Then*

$$W(f_*\mu, f_*\nu) \leq L_f W(\mu, \nu).$$

*Proof.* Let $\gamma$ be a coupling of $\mu$ and $\nu$ for which the Wasserstein distance $W(\mu, \nu)$ is attained (Remark G.2). Then by definition

$$W(\mu, \nu) = \int_{X \times X} d_X(x_1, x_2) \, d\gamma(x_1, x_2).$$

Define the map $\mathbf{f} : X \times X \to Y \times Y$ by $\mathbf{f}(x, y) = (f(x), f(y))$, and define $\gamma' = \mathbf{f}_* \gamma$. Then $\gamma'$ is a coupling of $f_* \mu$ and $f_* \nu$. Therefore

$$\begin{aligned}
W(f_* \mu, f_* \nu) &\leq \int_{Y \times Y} d_Y(y_1, y_2) \, d\gamma'(y_1, y_2) \\
&= \int_{X \times X} d_Y(f(x_1), f(x_2)) \, d\gamma(x_1, x_2) \\
&\leq L_f \int_{X \times X} d_X(x_1, x_2) \, d\gamma(x_1, x_2) = L_f W(\mu, \nu).
\end{aligned}$$

$\square$

We use the lemma to conclude that when pushing the signal via a Lipschitz continuous map, the Hausdorff distance between the updated profiles is bounded by the Hausdorff distance of the profiles up to the Lipschitz constant.

**Lemma J.4.** *Let $(\Omega_1, \mu_1)$, $(\Omega_2, \mu_2)$ be two probability spaces with P-operator-signals $(A_1, f_1) \in \mathcal{B}_{p,q,d}(\Omega_1)$ and $(A_2, f_2) \in \mathcal{B}_{p,q,d}(\Omega_2)$ on $\Omega_1$ and $\Omega_2$ respectively. For $k \in \mathbb{N}$, let $S_k(A_1, f_1), S_k(A_2, f_2)$ be the $k$-profiles of $(A_1, f_1)$ and $(A_2, f_2)$. For every Lipschitz continuous map $\phi : \mathbb{R}^d \to \mathbb{R}^n$, with Lipschitz constant $L_\phi$, we have*

$$d_H\big(\phi \tilde{\ast} S_k(A_1, f_1), \phi \tilde{\ast} S_k(A_2, f_2)\big) \leq L_\phi d_H\big(S_k(A_1, f_1), S_k(A_2, f_2)\big).$$

*Proof.* Assume without loss of generality that

$$d_H\big(\phi \tilde{\ast} S_k(A_1, f_1), \phi \tilde{\ast} S_k(A_2, f_2)\big) = \sup_{\phi \tilde{\ast} \nu_1 \in S_k(A_1, \phi \circ f_1)} \inf_{\phi \tilde{\ast} \nu_2 \in S_k(A_2, \phi \circ f_2)} W(\phi \tilde{\ast} \nu_1, \phi \tilde{\ast} \nu_2).$$

From Lemma J.3 we have $W(\phi \tilde{\ast} \nu_1, \phi \tilde{\ast} \nu_2) \leq L_\phi W(\nu_1, \nu_2)$. Therefore

$$d_H\big(\phi \tilde{\ast} S_k(A_1, f_1), \phi \tilde{\ast} S_k(A_2, f_2)\big) \leq L_\phi \sup_{\nu_1 \in S_k(A_1, f_1)} \inf_{\nu_2 \in S_k(A_2, f_2)} W(\nu_1, \nu_2),$$

and thus

$$d_H\big(\psi \tilde{\ast} S_k(A_1, f_1), \psi \tilde{\ast} S_k(A_2, f_2)\big) \leq L_\psi d_H\big(S_k(A_1, f_1), S_k(A_2, f_2)\big).$$

$\square$

*Remark* J.5. Note that the proof holds for arbitrary pushforwards, and not only for pushforward on the signal. Therefore, pushing forward every P-distribution via a Lipschitz continuous map, produces sets of probability measures (not necessarily profiles) whose Hausdorff distance between them is bounded by the Hausdorff distance between the profiles, up to Lipschitz constant.

### J.2. Hölder Continuity of Diagonal Marginalization

We move to prove Hölder continuity of the diagonal marginalization operation between profiles.

Diagonal Marginalization involves restriction to the diagonal and marginalization. Since marginalization is projection which is a Lipschitz continuous map, than we have already established the continuity property with respect to marginalization in Lemma J.4. Therefore, we need to show that the restriction operation on profiles to the diagonal $T_d$ is Hölder continuous function on profiles. This is nontrivial, as by definition, Hausdorff distance between subsets can generally increase. However, in the case of profiles, the Hausdorff distance between the diagonally restricted profiles (the subsets of profiles that consists of measures that are supported on the diagonal $T_d$ as described in Definition I.6) is bounded by the Hausdorff distance of the profiles, up to a constant that depends on the bound of the operator norms, and exponent that depends on $p$ and $d$.

Recall from equation (11) the following notation

$$\tau_p(\nu) = \max_{1 \leq i \leq n} \int_{\mathbb{R}^n} |x_i|^p \, d\nu(x) \in [0, \infty]. \tag{16}$$

We have already established in Lemma H.16 that $S_k(A, f) \subset \mathcal{P}_{q,c}(\mathbb{R}^{2k+d})$ for $(p, q) \in [1, \infty] \times [1, \infty)$, and $(A, f) \in \mathcal{B}_{p,q,d}^r(\Omega)$, where $c = \max\{1, r^q\}$.

Now we can prove the Hölder continuity of restriction to the diagonal for uniformly bounded $(p, q)$ normed P-operators.

**Theorem J.6.** *Let $(\Omega_1, \mu_1)$ and $(\Omega_2, \mu_2)$ be Borel probability spaces. For $(p, q) \in [1, \infty) \times [1, \infty]$ and $r > 0$, let $(A_1, f_1) \in \mathcal{B}_{p,q,1}^r(\Omega_1), (A_2, f_2) \in \mathcal{B}_{p,q,1}^r(\Omega_2)$ be P-operator-signals over $\Omega_1$ and $\Omega_2$ respectively, where $f_i : \Omega_i \to [-1, 1]$, $i = 1, 2$. Then*

$$d_H(S_k^{T_1}(A_1, f_1), S_k^{T_1}(A_2, f_2)) \leq C_{p,k,c,r} \cdot d_H(S_k(A_1, f_1), S_k(A_2, f_2))^{\frac{1}{p}},$$

*where $C_{p,k,c,r} = (2(2k + 1)c)^{1 - \frac{1}{p}} + (1 + r)2^{1 - \frac{1}{p}}$.*

Note that the Hölder constant and the exponent depend only on $p$ and not on $q$. Specifically, for $p = 1$, we obtain Lipschitz continuity with the constant $2 + r$.

*Proof.* Let $D_1^T \in S_k^{T_1}(A_1, f_1) \subset S_k(A_1, f_1)$ be a P-distribution supported on the diagonal $T_1$ (recall definition I.6).

By definition of the Hausdorff distance, there exists a P-distribution

$$D_2 \in S_k(A_2, f_2)$$

such that

$$W(D_1^T, D_2) \leq d_H\big(S_k(A_1, f_1), S_k(A_2, f_2)\big).$$

If, in addition, $D_2$ is supported on the diagonal $T_1$, the proof is complete.

Indeed, recall that for two sets $A, B$ in a metric space $(X, d)$, the Hausdorff distance is defined as

$$d_H(A, B) = \max \left\{ \sup_{a \in A} \inf_{b \in B} d(a, b), \ \sup_{b \in B} \inf_{a \in A} d(a, b) \right\}.$$

Hence, for every point $a \in A$ there exists a point $b \in B$ minimizing $d(a, b)$, and conversely.

In our context, taking $A = S_k(A_1, f_1)$ and $B = S_k(A_2, f_2)$, this means that for each $D_1^T \in A$ there exists a "closest" P-distribution $D_2 \in B$ realizing the minimal Wasserstein distance to $D_1^T$. Therefore, if this closest distribution $D_2$ lies in the diagonal subset $S_k^T(A_2, f_2)$, the Hausdorff distance between the two sets is precisely the Wasserstein distance $W(D_1^T, D_2)$, and the claim follows.

Otherwise, $D_2$ is not supported on $T_1$, so there are $v_1, \ldots, v_k : \Omega_2 \to [-1, 1]$, with $v_k(\omega) \neq f_2(\omega)$ for $\omega$ in a set of positive measure in $\Omega_2$, such that $D_2 = \phi_* \mu_2$, where $\xi : \Omega_2 \to \mathbb{R}^{2k+1}$ is defined by

$$\xi(\omega) = \big(v_1(\omega), \ldots, v_k(\omega), A_2 v_1(\omega), \ldots, A_2 v_k(\omega), f_2(\omega)\big).$$

Define $\widetilde{D_2^T} = D_{(A_2, f_2)}(v_1, \ldots, v_{k-1}, f_2)$, and set $\phi : \Omega_2 \to \mathbb{R}^{2k+1}$ be the map such that $\widetilde{D_2^T} = \phi_* \mu_2$. That is

$$\phi(\omega) = \big(v_1(\omega), \ldots, v_{k-1}(\omega), f_2(\omega), A_2 v_1(\omega), \ldots, A_2 v_{k-1}(\omega), A_2 f_2(\omega), f_2(\omega)\big).$$

By the triangle inequality we have

$$W(D_1^T, \widetilde{D_2^T}) \leq W(D_1^T, D_2) + W(D_2, \widetilde{D_2^T}). \tag{17}$$

Consider the second term of 17. Clearly, the measure $\gamma = (\xi, \phi)_* \mu_2$ is a coupling of $D_2$ and $\widetilde{D_2^T}$. Therefore

$$
\begin{aligned}
W(D_2, \widetilde{D_2^T}) &\leq \int_{\mathbb{R}^{2k+1} \times \mathbb{R}^{2k+1}} \|x - y\|_1 \, d\gamma(x, y) \\
&= \int_{\Omega_2} \|\xi(\omega) - \phi(\omega)\|_1 \, d\mu_2(\omega) \\
&= \int_{\Omega_2} |v_k(\omega) - f_2(\omega)| + |A_2 v_k(\omega) - A_2 f_2(\omega)| \, d\mu_2(\omega) \\
&\leq \|v_k - f_2\|_1 + \|A_2 v_k - A_2 f_2\|_1 \leq \|v_k - f_2\|_p + \|A_2 v_k - A_2 f_2\|_q \\
&\leq (1 + r)\|v_k - f_2\|_p.
\end{aligned}
$$

Next, we prove that $\|v_k - f_2\|_1 \leq W(D_1^T, D_2)$ using the Kantorovich duality theorem (Theorem G.4). Define $h : \mathbb{R}^{2k+1} \to \mathbb{R}$ by $h(x) = |x_k - x_{2k+1}|$. Then $h$ is Lipschitz continuous function. Indeed

$$
\begin{aligned}
|h(x) - h(y)| &= \big| |x_k - x_{2k+1}| - |y_k - y_{2k+1}| \big| \\
&\leq \big| (x_k - x_{2k+1}) - (y_k - y_{2k-1}) \big| \leq |x_k - y_k| + |x_{2k+1} - y_{2k+1}| \\
&\leq \|x - y\|_1.
\end{aligned}
$$

Therefore

$$
W(D_1^T, D_2) \geq \int_{\mathbb{R}^{2k+1}} h(x) \, dD_2(x) - \int_{\mathbb{R}^{2k+1}} h(x) \, dD_1^T(x).
$$

Since $h(x) = 0$ for every $x \in T_1$ we have that the second integral vanishes. We have that

$$
\int_{\mathbb{R}^{2k+1}} h(x) \, dD_2(x) = \int_{\Omega_2} |v_k(\omega) - f_2(\omega)| \, d\mu_2(\omega) = \|v_k - f_2\|_1.
$$

For $p \geq 1$, $|v_k - f_2| = \frac{1}{|v_k - f_2|^{p-1}} \cdot |v_k - f_2|^p$, and since $\Omega_2$ is probability space and $v_k$ and $f_2$ are both bounded by 1, we get

$$
\|v_k - f_2\|_1 \geq \left(\frac{1}{2}\right)^{p-1} \cdot \|v_k - f_2\|_p^p.
$$

Therefore

$$
\|v_k - f_2\|_p \leq 2^{1 - \frac{1}{p}} \cdot \left(W(D_1^T, D_2)\right)^{\frac{1}{p}}
$$

We conclude

$$
\begin{aligned}
W(D_1^T, \widetilde{D_2^T}) &\leq W(D_1^T, D_2) + (1 + r) \cdot \|v_k - f_2\|_p \\
&\leq W(D_1^T, D_2) + (1 + r) \cdot 2^{1 - \frac{1}{p}} \cdot \left(W(D_1^T, D_2)\right)^{\frac{1}{p}} \\
&\leq \left(\left(W(D_1^T, D_2)\right)^{1 - \frac{1}{p}} + (1 + r) \cdot 2^{1 - \frac{1}{p}}\right) \cdot \left(W(D_1^T, D_2)\right)^{\frac{1}{p}} \\
&\leq C_{p,k,c,r} \cdot \left(W(D_1^T, D_2)\right)^{\frac{1}{p}} \qquad\qquad (18) \\
&\leq C_{p,k,c,r} \cdot \left(d_H(S_k(A_1, f_1), S_k(A_2, f_2))\right)^{\frac{1}{p}},
\end{aligned}
$$

where in (18) we applied lemma G.10. By definition of the Hausdorff distance

$$
d_H(D_1^T, S_k^{T_1}(A_2, f_2)) \leq C_{p,k,c,r} \cdot \left(d_H(S_k(A_1, f_1), S_k(A_2, f_2))\right)^{\frac{1}{p}}.
$$

Taking the supremum over all $D_1^T \in S_k^{T_1}(A_1, f_1)$ we have

$$
d_H(S_k^{T_1}(A_1, f_1), S_k^{T_1}(A_2, f_2)) \leq C_{p,k,c,r} \cdot \left(d_H(S_k(A_1, f_1), S_k(A_2, f_2))\right)^{\frac{1}{p}}.
$$

$\square$

Using Theorem J.6 inductively, we extend this result to $d$-channel signals.

**Corollary J.7.** *Let $(\Omega_1, \mu_1)$ and $(\Omega_2, \mu_2)$ be Borel probability spaces. For $(p, q) \in [1, \infty) \times [1, \infty]$ and $r > 0$, let $(A_1, f_1) \in \mathcal{B}^r_{p,q,d}(\Omega_1), (A_2, f_2) \in \mathcal{B}^r_{p,q,d}(\Omega_2)$ be P-operator-signals over $\Omega_1$ and $\Omega_2$ respectively. Then*

$$d_H(\mathrm{S}^{T_d}_k(A_1, f_1), \mathrm{S}^{T_d}_k(A_2, f_2)) \leq C_{p,k,c,r,d} \cdot d_H(\mathrm{S}_k(A_1, f_1), \mathrm{S}_k(A_2, f_2))^{\frac{1}{p^d}},$$

*where*

$$C_{p,k,c,r,d} = \left( (2(2k+d)c)^{1-\frac{1}{p}} + (1+r)2^{1-\frac{1}{p}} \right)^d. \tag{19}$$

## J.3. Hölder Continuity of a Message Passing Layer

We can now conclude Hölder continuity of a MPL with respect to Hausdorff distance.

**Theorem J.8.** *Let $(\Omega_1, \mu_1)$ and $(\Omega_2, \mu_2)$ be Borel probability spaces. For $(p, q) \in [1, \infty) \times [1, \infty]$ and $r > 0$, let $(A_1, f_1) \in \mathcal{B}^r_{p,q,d}(\Omega_1), (A_2, f_2) \in \mathcal{B}^r_{p,q,d}(\Omega_2)$ be P-operator-signals with $k$-profiles denoted as $\mathrm{S}_k$ and $\tilde{\mathrm{S}}_k$, where $k = \sum_{l=0}^L d_l$ respectively.. Let $\varphi^{(l)}$ be an MPNN layer of an L-layer MPNN model $\varphi$. Then*

$$d_H\left( \mathfrak{s}^{(l)}_{k_l, d_l}, \tilde{\mathfrak{s}}^{(l)}_{k_l, d_l} \right) \leq L_{\varphi^{(l)}} \cdot C_{p,k,c,r,d} \cdot d_H\left( \mathfrak{s}^{(l-1)}_{k_{l-1}, d_{l-1}}, \tilde{\mathfrak{s}}^{(l-1)}_{k_{l-1}, d_{l-1}} \right)^{1/p^{d_{l-1}}},$$

*where $C_{p,k,c,r,d}$ is the constant given in equation 19.*

*Proof.*

$$
\begin{aligned}
d_H\left( \mathfrak{s}^{(l)}_{k_l, d_l}, \tilde{\mathfrak{s}}^{(l)}_{k_l, d_l} \right) =& d_H\left( \varphi^{(l)} \tilde{*} \mathrm{DM}_{d_{l-1}}(\mathfrak{s}^{(l-1)}_{k_{l-1}, d_{l-1}}), \varphi^{(l)} \tilde{*} \mathrm{DM}_{d_{l-1}}(\tilde{\mathfrak{s}}^{(l-1)}_{k_{l-1}, d_{l-1}}) \right) \overset{(1)}{\leq} \\
& L_{\varphi^{(l)}} \cdot d_H\left( \mathrm{DM}_{d_{l-1}}(\mathfrak{s}^{(l-1)}_{k_{l-1}, d_{l-1}}), \mathrm{DM}_{d_{l-1}}(\tilde{\mathfrak{s}}^{(l-1)}_{k_{l-1}, d_{l-1}}) \right) \overset{(2)}{\leq} \\
& L_{\varphi^{(l)}} \cdot d_H\left( (\mathfrak{s}^{(l-1)}_{k_{l-1}, d_{l-1}})^{T_{d_{l-1}}}, (\tilde{\mathfrak{s}}^{(l-1)}_{k_{l-1}, d_{l-1}})^{T_{d_{l-1}}} \right) \overset{(3)}{\leq} \\
& L_{\varphi^{(l)}} \cdot C_{p,k,c,r,d} \cdot d_H\left( \mathfrak{s}^{(l-1)}_{k_{l-1}, d_{l-1}}, \tilde{\mathfrak{s}}^{(l-1)}_{k_{l-1}, d_{l-1}} \right)^{1/p^{d_{l-1}}},
\end{aligned}
$$

where inequality $(1)$ is by Lemma J.4, $(2)$ is by Remark J.5 and $(3)$ is by Corollary J.7.

$\square$

## J.4. Hölder Continuity of MPNNs

We now move to the proof of Theorem J.2.

*Proof.* Recall that the output of an L-layer MPNN model on profiles is given by $\mathfrak{S}(A, f) = \psi\left( \int_{[1,1]^{d_L}} \vec{x}\, d\nu(\vec{x}) \right)$, where $\nu$ is the unique measure corresponds to the 0-profile $\mathfrak{s}_{0, d_L}$. Let $\nu_i$ denote the unique measure remaining in the 0-profile obtained from MPNN on $\mathrm{S}_k(A_i, f_i)$. Then

$$
\begin{aligned}
\|\mathfrak{S}(A_1, f_1) - \mathfrak{S}(A_2, f_2)\|_2 \leq & \psi\left( \int_{[1,1]^{d_L}} \vec{x}\, d\nu_1(\vec{x}) \right) - \psi\left( \int_{[1,1]^{d_L}} \vec{x}\, d\nu_2(\vec{x}) \right) \leq \\
& L_\psi \cdot W(\nu_1, \nu_2) = L_\psi \cdot d_H\left( \mathfrak{s}_{0, d_L}(A_1, f_1), \mathfrak{s}_{0, d_L}(A_2, f_2) \right).
\end{aligned}
$$

By Theorem J.8 we have the inequality

$$L_\psi \cdot d_H\left( \mathfrak{s}_{0, d_L}(A_1, f_1), \mathfrak{s}_{0, d_L}(A_2, f_2) \right) \leq L_\psi \cdot L_{\varphi^{(L)}} \cdot C_{p,k,c,r,d} \cdot d_H\left( \mathfrak{s}^{(L)}_{k_l, d_l}, \tilde{\mathfrak{s}}^{(L)}_{k_L, d_L} \right)$$

The proof then follows from applying Theorem J.8 for every layer.

$\square$

Theorem J.1 now follows.

*Proof.* Recall that by the commutation property (Theorem I.9) we have that $\mathfrak{H}_{(A,f)} = \mathfrak{S}_{(A,f)}$. Then, by Theorem J.2

$$\|\mathfrak{H}(A_1, f_1) - \mathfrak{H}(A_2, f_2)\|_2 = \|\mathfrak{S}(A_1, f_1) - \mathfrak{S}(A_2, f_2)\|_2 \leq C_{(\varphi, \psi)} d_H(\mathrm{S}_k(A_1, f_1), \mathrm{S}_k(A_2, f_2))^{\frac{1}{p^d}}.$$

For every fixed $K \in \mathbb{N}$ we have by the definition of the action metric (Definition H.14) that

$$d_H(\mathrm{S}_K(A_1, f_1), \mathrm{S}_K(A_2, f_2)) \leq 2^K d_M(A_1, f_1), (A_2, f_2)),$$

we take $K = k_0 = \sum_{l=0}^{L-1} d_l$, and conclude that

$$\|\mathfrak{H}(A_1, f_1) - \mathfrak{H}(A_2, f_2)\|_2 \leq C_{(\varphi, \psi)} \left(2^K \cdot d_M(A_1, f_1), (A_2, f_2))\right)^{\frac{1}{p^d}}$$

$\square$

# K. The Computational Structure of DIDMs and MPNNs

Let $(\Omega, \mu)$ be a Borel probability space, and $(A, f) \in \mathcal{BF}_d^r(\Omega)$ a bofop-signal. We extend the 1-WL algorithm that we described in Appendix E to the setting of bofop-signals as follows.

## K.1. 1-WL on bofop-Signals

Recall the following construction from section E. Let $\mathcal{M}^0 = [-1, 1]^d$. For every $L \geq 0$, define $\mathcal{H}^L = \prod_{j \leq L} \mathcal{M}^j$ and $\mathcal{M}^{L+1} = \mathscr{M}_{\leq r}(\mathcal{H}^L)$, where the topologies of $\mathcal{H}^L$ and $\mathcal{M}^L$ are the product and weak* topologies respectively.

We define the bofop-IDMs and bofop-DIDMs for bofop-signals which extends the classic definitions for graphs and graphons that we defined in definition E.1.

**Definition K.1** (Graphop-IDMs and Graphop-DIDMs for Graphop-Signals). Let $(\Omega, \mu)$ be a Borel probability space and $(A, f) \in \mathcal{BF}_d^r(\Omega)$ be a bofop-signal. We define the map $\gamma_{(A,f),0} : \Omega \to \mathcal{H}^0$ to be the map $\gamma_{(A,f),0}(x) := f(x)$ for every $x \in \Omega$. Inductively $\gamma_{(A,f),L+1} : \Omega \to \mathcal{H}^{L+1}$ is defined by

1. $\gamma_{(A,f),L+1}(x)(j) = \gamma_{(A,f),L}(x)(j)$ for every $j \leq L$.

2. $\gamma_{(A,f),L+1}(x)(L+1)(E) = \left(A\mathbb{1}_{\gamma_{(A,f),L}^{-1}(E)}\right)(x)$, where $E \subset \mathcal{H}^L$ is a Borel set.

Finally, for every $L \geq 0$, let $\Gamma_{(A,f),L} = \gamma_{(A,f),L_*}\mu$, i.e. the pushforward of $\mu$ via $\gamma_{(A,f),L}$. We call $\gamma_{(A,f),L}$ a bofop-IDM of order $L$, and $\Gamma_{(A,f),L}$ a bofop-DIDM of order $L$.

It is easy to see that this definition extends several known settings. When $f$ is the constant signal $f \equiv 1$ and $A$ is the adjacency matrix of a graph $G$ we get the standard coloring that was defined originally in (Grebík & Rocha, 2022). When $A$ is a kernel operator, and in particular a graphon, the definition coincides with the coloring considered in (Böker et al., 2024). Finally, allowing non-constant signal, we get the setting of (Rauchwerger et al., 2025) that we described in Appendix E

## K.2. Hierarchy of DIDM Spaces - Dense Graphs and Graphons

Let us clarify the structural picture underlying DIDMs. For every fixed depth $L$, the space of DIDMs $\mathcal{P}(\mathcal{H}^L)$ without any requirement that they arise from any graph-structure is a compact space with respect to the product topology (Theorem E.5). This compact ambient space plays a purely structural role: it provides a closed and metrizable domain in which all later subspaces are defined.

However, as already established in Example E.7, the space of graphon-DIDMs, is a strict subset of the space of all DIDMs. In particular, there exist DIDMs that do not correspond to any graph.

When restricted to graphons, bofop-DIDMs are exactly graphon-DIDMs, as any graphon is in particular a bofop. The resulting space of graphon-DIDMs is again shown to be compact with respect to $\delta_{\text{DIDM}}^L$ (Appendix E).

To connect this limit theory to finite graphs, let $G = (V, E)$ be a graph with $|V| = N$ nodes. We assign each edge $e \in E$ with weight

$$a_e = \frac{w_e}{N}, \qquad w_e \in [0, 1].$$

This normalization is the standard dense-graph scaling that ensures convergence to a bounded graphon operator as $N \to \infty$ with respect to the cut distance. For such graphs, to the associated DIDMs we call dense-graph-DIDMs. As expected from graphon approximation theory, the space of dense-graph-DIDMs is dense in the space of graphon-DIDMs (Theorem 79 in (Rauchwerger et al., 2025)).

Sparse graphs, on the other hand, do not fit into the graphon framework: under scaling, they considered similar to the empty graph. As demonstrated in Example H.1.5, sparse-graph limits are more naturally represented by bofops rather than graphons. Definition K.1 allows us to extend the IDM/DIDM construction to bofops. Moreover, bofops with uniformly bounded $\infty \to \infty$ norm strictly generalize both graphons and sparse-graph limits (Theorem H.7). Consequently, the space of graphon-DIDMs is a strict subset of the space of graphop-DIDMs.

This naturally raises the question of whether compactness persists when the class of realizable DIDMs is extended from graphons to bofops. In the following section, specifically in Theorem M.2, we show that the answer is yes. The space of bofoop-DIDMs is compact as well, justifying the necessity of extending the analysis beyond graphons.

### K.3. Equivalency of MPNNs on DIDMs and Graphop-Signals

Recall definitions I.1 and E.8 for application of MPNN model on bofop-signals and computational DIDMs respectively. MPNN on bofop-IDMs and bofop-DIDMs is a canonical extension of MPNN on graphop-signals as follows. Let $(A, f)$ be a bofop-signal and $(\varphi, \psi)$ be an $L$-layer MPNN model with readout. Then, given the bofoop-IDMs $\{\gamma_{(A,f),t}^{(t)}\}_{t=0}^{L}$ and bofop-DIDMs $\Gamma_{(A,f),L}$ from definition K.1, we have that $\mathfrak{h}_x^{(t)} = \mathfrak{f}_{\gamma_{(A,f),t}(x)}^{(t)}$ for every $t \in [L]$ and $x \in \Omega$. Similarly $\mathfrak{H}_{(A,f)} = \mathfrak{F}_{\Gamma_{(A,f),L}}$.

To prove this result, we need the following Lemma which is an extension of Theorem 16.11 from (Billingsley, 2012) for bofops.

**Lemma K.2.** *Let $(\Omega, \mu)$ be a Borel probability space, and let $A$ be a bofop. For every $x \in \Omega$ consider the measure $\nu_{A,x}$, defined for measurable sets $E \subset \Omega$ by $\nu_{A,x}(E) = (A\mathbb{1}_E)(x)$. Then every measurable function $f : \Omega \to \mathbb{R}$ is integrable with respect to $\nu_{A,x}$, and for a.e. $x \in \Omega$, we have*

$$\int_\Omega f(y) \, d\nu_{A,x}(y) = Af(x).$$

*Proof.* First, assume that $f(x) = \mathbb{1}_E(x)$ for some measurable set $E \subset \Omega$. Then, by definition

$$\int_\Omega \mathbb{1}_E(y) \, d\nu_{A,x}(y) = \nu_{A,x}(E) = (A\mathbb{1}_E)(x).$$

By the linearity of the integral and $A$, a similar equality follows when $f$ is a simple function (i.e. linear combination of indicators). To see that, write $f(x) = \sum_{k=1}^n \mathbb{1}_{\mathcal{E}_k}(x)$, where $n \in \mathbb{N}$. Then

$$\int_\Omega f(y) \, d\nu_{A,x}(y) = \int_\Omega \sum_{k=1}^n \mathbb{1}_{E_k}(y) \, d\nu_{A,x}(y) =$$

$$\sum_{k=1}^n \int_\Omega \mathbb{1}_{E_k}(y) \, d\nu_{A,x}(y) =$$

$$\sum_{k=1}^n A\mathbb{1}_{E_k}(x) = A\Big(\sum_{k=1}^n \mathbb{1}_{E_k}(x)\Big) = Af(x).$$

Since each measurable nonnegative function $f$ can be approximated by simple functions, by the monotone convergence theorem we have that $\int_\Omega f(y) \, d\nu_{A,x}(y) = Af(x)$. If $f$ has negative values, write $f = f^+ - f^-$, where $f^+(x) = f(x)$ if

$f(x)$ is positive and 0 otherwise, and define $f^-(x) = -f(x)$ if $f(x)$ is negative and 0 otherwise. Since both $f^+$ and $f^-$ are nonnegative functions, we use the linearity of the integral again, to conclude our desired result. □

**Lemma K.3.** *Let $(A, f)$ be a bofop-signal and $\varphi$ be an L-layer MPNN model. Then, given the bofop-IDMs $\{\gamma_{(A,f),t}\}_{t=0}^L$ we have that $\mathfrak{h}_x^{(t)} = \mathfrak{f}_{\gamma_{(A,f),t}(x)}^{(t)}$ for every $t \in [L]$ and almost every $x \in \Omega$.*

*Proof.* We prove by induction.

*Induction Base:* By definition we have

$$\mathfrak{h}_x^{(0)} = \varphi \circ f(x) = \mathfrak{f}_{\gamma_{(A,f),0}(x)}^{(0)}.$$

Let $1 \le t \le L$, assume that for a.e. $x \in \Omega$,

$$\mathfrak{h}_x^{(t-1)} = \mathfrak{f}_{\gamma_{(A,f),t-1}(x)}^{(t-1)}.$$

*Induction Step:* We have

$$\mathfrak{h}_x^{(t)} = \varphi^{(t)}\left(\mathfrak{h}_x^{(t-1)}, (A\mathfrak{h}_-^{t-1})(x)\right).$$

Recall that for every measurable $E \subset \mathcal{H}^{t-1}$,

$$\gamma_{(A,f),t}(x)(t)(E) = \left(A\mathbb{1}_{\gamma_{(A,f),t-1}^{-1}(E)}\right)(x)$$

Consider the measure on $\Omega$,

$$\nu_{A,x}(E) = \left(A\mathbb{1}_E(y)\right)(x).$$

Then

$$\gamma_{(A,f),t}(x)(t) = \gamma_{(A,f),t-1}(x)_*\nu_{A,x}.$$

Therefore,

$$\begin{aligned}
\mathfrak{f}_{\gamma_{(A,f),t}(x)}^{(t)} &= \varphi^{(t)}\left(\mathfrak{f}_{p_{t,t-1}^{(t-1)}(\gamma_{(A,f),t})}, \int_{\mathcal{H}^{t-1}} \mathfrak{f}_-^{t-1}\, dp_t(\gamma_{(A,f),t}(x))\right) \\
&= \varphi^{(t)}\left(\mathfrak{f}_{\gamma_{(A,f),t-1}}^{(t-1)}, \int_{\mathcal{H}^{t-1}} \mathfrak{f}_-^{t-1}\, d\gamma_{(A,f),t}(x)(t)\right) \\
&= \varphi^{(t)}\left(\mathfrak{f}_{\gamma_{(A,f),t-1}}^{(t-1)}, \int_{\mathcal{H}^{t-1}} \mathfrak{f}_-^{t-1}\, d\gamma_{(A,f),t-1}(x)_*\nu_{A,x}\right) \\
&= \varphi^{(t)}\left(\mathfrak{f}_{\gamma_{(A,f),t-1}}^{(t-1)}, \int_{\Omega} \mathfrak{f}_-^{(t-1)} \circ \gamma_{(A,f),t-1}(y)\, d\nu_{A,x}(y)\right) \\
&= \varphi^{(t)}\left(\mathfrak{f}_{\gamma_{(A,f),t-1}}^{(t-1)}, A\mathfrak{f}_{\gamma_{(A,f),t-1}(-)}^{(t-1)}(x)\right),
\end{aligned}$$

where in the last equality we used lemma K.2. Hence by the induction assumption we obtain

$$\mathfrak{h}_x^{(t)} = \mathfrak{f}_{\gamma_{(A,f),t}(x)}^{(t)}.$$

□

**Lemma K.4.** *Let $(A, f)$ be a bofop-signal, and $(\varphi, \psi)$ be a L-layer model with readout. Then*

$$\mathfrak{F}_{\Gamma_{(A,f),L}} = \mathfrak{H}_{(A,f)}.$$

*Proof.* Recall that $\Gamma_{(A,f),L} = \mu_*\gamma_{(A,f),L}$. Then

$$\mathfrak{F}_{\Gamma_{(A,f),L}} = \psi\Big(\int_{\mathcal{H}^L} \mathfrak{f}_-^{(L)}\, d\Gamma_{(A,f),L}\Big)$$

$$= \psi\Big(\int_{\mathcal{H}^L} \mathfrak{f}_-^{(L)}\, d\gamma_{(A,f),L_*}\mu\Big)$$

$$= \psi\Big(\int_{\Omega} \mathfrak{f}_{\gamma_{(A,f),L}(x)}\, d\mu\Big)$$

$$= \psi\Big(\int_{\Omega} \mathfrak{h}_x^{(L)}\, d\mu\Big) = \mathfrak{H}_{(A,f)}.$$

$\square$

### K.4. DIDM Mover's Distance of Bofop-Signals

Next, we define a distance between bofop-signals, viewed as distance between their bofop-DIDMs, analog to the original setting that was described before definition E.4.

**Definition K.5** (DIDM Mover's Distance). Given two bofop-signals $(A_1, f_1), (A_2, f_2)$ and $L \geq 1$, the *DIDM Mover's Distance* between $(A_1, f_1), (A_2, f_2)$ is defined as

$$\delta_{\text{DIDM}}^L\big((A_1, f_1), (A_2, f_2)\big) = \mathbf{OT}_{\mathbf{d}_{\text{IDM}}^{\mathbf{L}}}(\Gamma_{(A_1,f_1),L}, \Gamma_{(A_2,f_2),L}).$$

## L. Density of Graphs in the Space of Bofop-DIDMs

In this section, we prove that the space of DIDMs induced by graphs via 1-WL is dense in the space of bofop-DIDMs.

Throughout this section, we assume that the standard Borel probability space $(\Omega, \mu)$ is atomless.

To approximate bofop-signal by graphs, we partition the space $\Omega$ into subsets with equal measure, each representing a node of the graph, similarly to Definition D.4 inspired by (Levie, 2024). The atomless assumption is necessary for the existence of the equipartition. If $\Omega$ contains atoms, we replace it by the atomless standard probability space $\Omega' = \Omega \times [0, 1]$ with the product measure $\mu \times \lambda$ where $\lambda$ is the Lebesgue measure on $[0, 1]$.

Any bofop signal $(A, f) \in \mathcal{BF}_d^r(\Omega)$ can be lifted to a bofop-signal $(A', f') \in \mathcal{BF}_d^r(\Omega')$ by defining

$$f'(x, t) = f(x) \qquad \text{and} \qquad A'g(x, t) = A\tilde{g}(x), \tag{20}$$

where $\tilde{g}(x) = \int_{[0,1]} g(x, t)\, d\lambda(t)$. We begin by proving that this lifting is invisible to the 1-WL test. Namely, we show that applying the 1-WL algorithm to the bofop-signal $(A, f)$ and its lifted counterpart $(A', f')$ produces the same Bofop-DIDM.

**Proposition L.1.** *Let $(A, f) \in \mathcal{BF}_d^r(\Omega)$ be a bofop-signal and let $(A', f') \in \mathcal{BF}_d^r(\Omega')$ be the bofop-signal defined by equation (20). Then, for every $L \geq 0$,*

$$\Gamma_{(A,f),L} = \Gamma_{(A',f'),L}.$$

*Proof.* We prove by induction on $L$ that for $\mu'$-a.e. $(x, t) \in \Omega'$,

$$\gamma_{(A',f'),L}(x, t) = \gamma_{(A,f),L}(x).$$

*Induction Base:* By definition 5.1 we have

$$\gamma_{(A',f'),0}(x, t) = f'(x, t) = f(x) = \gamma_{(A,f),0}(x).$$

*Induction Step:* Assume that for $L \geq 0$, $\gamma_{(A',f'),L}(x, t) = \gamma_{(A,f),L}(x)$, we prove that $\gamma_{(A',f'),L+1}(x, t) = \gamma_{(A,f),L+1}(x)$. By the induction assumption and Definition 5.1, for every $j \leq L$,

$$\gamma_{(A',f'),L+1}(x, t)(j) = \gamma_{(A',f'),L}(x, t)(j) = \gamma_{(A,f),L}(x)(j) = \gamma_{(A,f),L+1}(x)(j).$$

It is sufficient to prove that for the last coordinates we have equality between $\gamma_{(A',f'),L+1}(x,t)(L+1) = \gamma_{(A,f),L+1}(x)(L+1)$ as measures over $\mathcal{H}^L$. Let $E \subset \mathcal{H}^L$ be a measurable set. Then,

$$\gamma_{(A',f'),L+1}(x,t)(L+1)(E) = \left( A' \mathbb{1}_{\gamma_{(A',f'),L}^{-1}(E)} \right)(x,t) = A \left( \int_{[0,1]} \mathbb{1}_{\gamma_{(A',f'),L}^{-1}(E)}(-,s) \, d\lambda(s) \right)(x).$$

By our induction assumption, for $(x,t) \in \Omega'$, $(x,t) \in \gamma_{(A',f'),L}^{-1}(E)$ if and only if $x \in \gamma_{(A,f),L}^{-1}(E)$. Therefore,

$$A \left( \int_{[0,1]} \mathbb{1}_{\gamma_{(A',f'),L}^{-1}(E)}(-,s) \, d\lambda(s) \right)(x) = A \mathbb{1}_{\gamma_{(A,f),L}^{-1}(E)}(x) = \gamma_{(A,f),L+1}(x)(L+1)(E).$$

We obtain that

$$\gamma_{(A',f'),L+1}(x,t) = \gamma_{(A,f),L+1}(x).$$

To prove equality for the bofop-DIDMs set $E \subset \mathcal{H}^L$. Then

$$\Gamma_{(A',f'),L}(E) = \mu'(\{(x,t) \in \Omega \times [0,1] \,|\, \gamma_{(A',f'),L}(x,t) \in E\}) = \mu(\{x \in \Omega \,|\, \gamma_{(A,f),L}(x) \in E \}) = \Gamma_{(A,f),L}(E).$$

$\square$

For $n \in \mathbb{N}$ Let $\mathcal{P} = \{I_k\}_{k=1}^n$ be an equipartition of $\Omega$ into pairwise disjoint sets such that $\mu(I_k) = \frac{1}{n}$ for every $k \in [n]$.

**Definition L.2** (Projection Bofop). Let $(\Omega, \mu)$ be an atomless standard probability space, $n \in \mathbb{N}$ and let $\mathcal{P} = \{I_k\}_{k=1}^n$ be an equipartition of $\Omega$. We define the *projection bofop* $P_\mathcal{P}$ by

$$P_\mathcal{P} f(x) = n \sum_{k=1}^n \left( \int_{I_k} f(y) \, d\mu(y) \right) \mathbb{1}_{I_k}(x).$$

Note that indeed $P_\mathcal{P}$ is a bofop as it is self adjoint and positively preserving linear operator with operator norm $\|P_\mathcal{P}\|_{\infty \to \infty} = 1$.

Recall from Definition D.4 that given a graph-signal $(G, \mathbf{f})$ with node set $[n]$ and equipartition of $[0,1]$ into $n$ intervals $[(k-1)/n, k/n)$, the signal $\mathbf{f}$ induces a piecewise constant function $f_\mathbf{f}$ defined by $f_\mathbf{f}(x) = \sum_{k=1}^n \mathbf{f}_k \mathbb{1}_{[(k-1)/n,k/n)}(x)$ where $\mathbf{f}_k$ is the feature vector of the node $k \in [n]$. This notion can easily be extended to our more general bofop-signal analysis. Formally, given a graph signal $(G, \mathbf{f})$ and an equipartition $\mathcal{P} = \{I_k\}_{k=1}^n$ of $\Omega$, the induced signal $f_\mathbf{f}^\mathcal{P} \in \mathcal{L}^\infty(\Omega)$ is defined by

$$f_\mathbf{f}^\mathcal{P}(x) = \sum_{k=1}^n \mathbf{f}_k \mathbb{1}_{I_k}(x)$$

Observe that the image of the projection bofop $P_\mathcal{P}$ is precisely the subspace of such induced signals. Specifically, for every $f \in \mathcal{L}^\infty(\Omega)$, its projection under $P_\mathcal{P}$ is exactly the induced signal $f_\mathbf{f}^\mathcal{P}$, where the coordinates are obtained via local averaging $(f_\mathbf{f}^\mathcal{P})_k = n \int_{I_k} f \, d\mu$. The projection bofop allows us to map not only signals but also bofops into the discrete domain. Every bofop $A$ can be restricted into the equipartition $\mathcal{P}$ by applying the projection bofop before and after the bofop. We define the *projected bofop* $A_\mathcal{P}$ by

$$A_\mathcal{P} = P_\mathcal{P} A P_\mathcal{P}.$$

We show that given a bofop $A$, the projected bofop acts naturally as a kernel operator, and its action on induced signals exactly mirrors the discrete action of a graph shift operator on feature vectors.

**Lemma L.3.** *Let $(\Omega, \mu)$ be a standard atomless probability space, $n \in \mathbb{N}$ and let $\mathcal{P} = \{I_k\}_{k=1}^n$ be an equipartition of $\Omega$. For every bofop $A \in \mathcal{BF}^r(\Omega)$, the projected bofop $A_\mathcal{P}$ acts as a kernel operator. Namely, there exists a symmetric measurable function $K_\mathcal{P}$ over $\Omega^2$ such that*

$$(A_\mathcal{P} f)(x) = \int_\Omega K_\mathcal{P}(x,y) \cdot f(y) \, d\mu(y),$$

*for every $f \in \mathcal{L}^\infty(\Omega)$, where*

$$K_\mathcal{P}(x,y) = n^2 \sum_{j=1}^n \sum_{k=1}^n (\mathbb{1}_{I_k}, \mathbb{1}_{I_j})_A \cdot \mathbb{1}_{I_j}(x) \mathbb{1}_{I_k}(y)$$

*Proof.* Let $f \in \mathcal{L}^\infty(\Omega)$. We apply each bofop separately. By Definition L.2

$$(P_\mathcal{P} f)(x) = n \sum_{k=1}^n \left( \int_{I_k} f(y) \, d\mu(y) \right) \mathbb{1}_{I_k}(x).$$

Denote by $\alpha_k = n \int_{I_k} f \, d\mu$. Applying $A$ on both sides gives

$$(A P_\mathcal{P} f)(x) = \sum_{k=1}^n \alpha_k \cdot (A \mathbb{1}_{I_k})(x),$$

where we used the linearity of $A$ and the fact that $\alpha_k$ is constant for every $k \in [n]$. We apply the projection bofop $P_\mathcal{P}$ again and obtain

$$(A_\mathcal{P} f)(x) = n \sum_{k=1}^n \sum_{j=1}^n \alpha_k \left( \int_{I_j} (A \mathbb{1}_{I_k})(y) \, d\mu(y) \right) \mathbb{1}_{I_j}(x).$$

Let $K \in \mathbb{R}^{n \times n}$ be the matrix defined by

$$K_{j,k} = n^2 \int_{I_j} (A \mathbb{1}_{I_k})(z) \, d\mu(z) = n^2 (\mathbb{1}_{I_k}, \mathbb{1}_{I_j})_A, \tag{21}$$

where $(v, u)_A := \int_\Omega (Av) \cdot u \, d\mu$ for every $u, v \in \mathcal{L}^\infty(\Omega)$. Because $A$ is self-adjoint, the bilinear form is symmetric, implying $K_{j,k} = K_{k,j}$.

Substituting $\alpha_k = n \int_\Omega f(y) \mathbb{1}_{I_k}(y) \, d\mu(y)$ back into the equation for $(A_\mathcal{P} f)(x)$ yields:

$$(A_\mathcal{P} f)(x) = \sum_{j=1}^n \sum_{k=1}^n K_{j,k} \left( \int_\Omega f(y) \mathbb{1}_{I_k}(y) \, d\mu(y) \right) \mathbb{1}_{I_j}(x)$$

$$= \int_\Omega \left( \sum_{j=1}^n \sum_{k=1}^n K_{j,k} \mathbb{1}_{I_j}(x) \mathbb{1}_{I_k}(y) \right) f(y) \, d\mu(y).$$

Defining the symmetric measurable step-function $K_\mathcal{P}(x, y) := \sum_{j=1}^n \sum_{k=1}^n K_{j,k} \mathbb{1}_{I_j}(x) \mathbb{1}_{I_k}(y)$, we obtain the final kernel representation:

$$(A_\mathcal{P} f)(x) = \int_\Omega K_\mathcal{P}(x, y) f(y) \, d\mu(y).$$

$\square$

Every graph $G$ with $n$ nodes can be associated with some graph shift operator, denoted $A_G = (a_{i,j})_{i,j \in [n]}$ (such as the adjacency matrix or its normalized counterpart). Using the equipartition $\mathcal{P}$, we can induce a corresponding piecewise constant function $K_\mathcal{P}$ by $K_\mathcal{P}(x, y) = n \sum_{i,j=1}^n a_{i,j} \mathbb{1}_{I_i}(x) \mathbb{1}_{I_j}(y)$. Because this induced kernel operator acts entirely on the partitioned space, it corresponds to a projected bofop. The following formalizes this idea, proving that applying a projected bofop to an induced signal is exactly equivalent to applying the discrete graph shift operator to the feature vectors and then inducing the resulting signal.

**Lemma L.4.** *Let $(\Omega, \mu)$ be a standard atomless probability space, $n \in \mathbb{N}$ and $\mathcal{P} = \{I_k\}_{k=1}^n$ an equipartition of $\Omega$. Let $A \in \mathcal{BF}^r(\Omega)$ be a bofop and $A_\mathcal{P}$ the projected bofop. Let $K \in \mathbb{R}^{n \times n}$ be the symmetric matrix defined in equation (21), and denote $S = \frac{1}{n} K$. For every graph signal $(G, \mathbf{f})$ with node set $[n]$ and feature matrix $\mathbf{f} \in \mathbb{R}^{n \times d}$, where each row $k \in [n]$ is the feature vector of the node $k$, we have*

$$A_\mathcal{P} f_{\mathbf{f}}^\mathcal{P} = f_{S\mathbf{f}}^\mathcal{P}.$$

*Proof.* By Lemma L.3, the action of the projected bofop on the induced signal $f_{\mathbf{f}}^\mathcal{P}$ is given by the integral

$$(A_\mathcal{P} f_{\mathbf{f}}^\mathcal{P})(x) = \int_\Omega K_\mathcal{P}(x, y) f_{\mathbf{f}}^\mathcal{P}(y) \, d\mu(y),$$

where $K_{\mathcal{P}}(x, y) = \sum_{j=1}^{n} \sum_{k=1}^{n} K_{j,k} \mathbb{1}_{I_j}(x) \mathbb{1}_{I_k}(y)$.

Substituting the definitions of $K_{\mathcal{P}}$ and the induced signal $f_{\mathbf{f}}^{\mathcal{P}}(y) = \sum_{m=1}^{n} \mathbf{f}_m \mathbb{1}_{I_m}(y)$ into the integral, we obtain:

$$(A_{\mathcal{P}} f_{\mathbf{f}}^{\mathcal{P}})(x) = \int_{\Omega} \left( \sum_{j=1}^{n} \sum_{k=1}^{n} K_{j,k} \mathbb{1}_{I_j}(x) \mathbb{1}_{I_k}(y) \right) \left( \sum_{m=1}^{n} \mathbf{f}_m \mathbb{1}_{I_m}(y) \right) d\mu(y)$$

$$= \sum_{j=1}^{n} \sum_{k=1}^{n} \sum_{m=1}^{n} K_{j,k} \mathbf{f}_m \mathbb{1}_{I_j}(x) \int_{\Omega} \mathbb{1}_{I_k}(y) \mathbb{1}_{I_m}(y) \, d\mu(y).$$

Because the sets in the equipartition $\mathcal{P}$ are pairwise disjoint, the product of the indicator functions $\mathbb{1}_{I_k}(y) \mathbb{1}_{I_m}(y)$ is zero whenever $k \neq m$. When $k = m$, the integral evaluates to the measure of the interval, which is $\mu(I_k) = \frac{1}{n}$. Thus, the summation becomes

$$(A_{\mathcal{P}} f_{\mathbf{f}}^{\mathcal{P}})(x) = \sum_{j=1}^{n} \sum_{k=1}^{n} K_{j,k} \mathbf{f}_k \mathbb{1}_{I_j}(x) \left( \frac{1}{n} \right)$$

$$= \sum_{j=1}^{n} \left( \sum_{k=1}^{n} \left( \frac{1}{n} K_{j,k} \right) \mathbf{f}_k \right) \mathbb{1}_{I_j}(x).$$

Since $S = \frac{1}{n} K$, we substitute $S_{j,k} = \frac{1}{n} K_{j,k}$. The inner sum then becomes exactly the $j$-th row of the matrix product $S\mathbf{f}$, which we denote as $(S\mathbf{f})_j = \sum_{k=1}^{n} S_{j,k} \mathbf{f}_k$. Substituting this back yields

$$(A_{\mathcal{P}} f_{\mathbf{f}}^{\mathcal{P}})(x) = \sum_{j=1}^{n} (S\mathbf{f})_j \mathbb{1}_{I_j}(x).$$

Recognizing the right-hand side as the definition of the induced signal for the feature matrix $S\mathbf{f}$, we conclude that

$$(A_{\mathcal{P}} f_{\mathbf{f}}^{\mathcal{P}})(x) = f_{S\mathbf{f}}^{\mathcal{P}}(x).$$

$\square$

We summarize the construction in the following commutative diagram.

$$
\begin{array}{ccc}
\mathbf{f} & \xrightarrow{\ S\ } & S\mathbf{f} \\
{\scriptstyle \mathcal{P}} \downarrow & & \downarrow {\scriptstyle \mathcal{P}} \\
f_{\mathbf{f}}^{\mathcal{P}} & \xrightarrow{\ A_{\mathcal{P}}\ } & f_{S\mathbf{f}}^{\mathcal{P}}
\end{array}
$$

We next show that as we take finer equipartitions, the corresponding sequence of projected bofops converges to the original bofop in the strong topology (i.e. pointwise convergence) with respect to $L_1$ norm.

**Theorem L.5.** *Let $(\Omega, \mu)$ be a standard probability space and $(A, f) \in \mathcal{BF}_d^r(\Omega)$ be a bofop-signal. Let $\{\mathcal{P}_n\}_{n=1}^{\infty}$ be a sequence of equipartitions of $\Omega$ for $n$ sets, with corresponding sequence of projected bofops $\{A_{\mathcal{P}_n}\}_{n=1}^{\infty}$. Then*

$$\lim_{n \to \infty} \|P_{\mathcal{P}_n} f - f\|_1 = 0,$$

*and for every $g \in \mathcal{L}^1(\Omega)$,*

$$\lim_{n \to \infty} \|A_{\mathcal{P}_n} g - Ag\|_1 = 0.$$

*Proof.* We start with the convergence of the projected signal. Let $\varepsilon > 0$. Since piecewise constant functions are dense in $\mathcal{L}^1(\Omega)$, there exists a piecewise constant function $f_\varepsilon$ on finite number of sets, say $M > 0$ such that $\|f - f_\varepsilon\|_1 < \epsilon$. By the

triangle inequality we have

$$
\begin{aligned}
\|P_{\mathcal{P}_n} f - f\|_1 &\leq \|P_{\mathcal{P}_n} f - P_{\mathcal{P}_n} f_\varepsilon\|_1 + \|P_{\mathcal{P}_n} f_\varepsilon - f_\varepsilon\|_1 + \|f_\varepsilon - f\|_1 \\
&\leq \|P_{\mathcal{P}_n}\|_{1 \to 1} \cdot \|f_\varepsilon - f\|_1 + \|P_{\mathcal{P}_n} f_\varepsilon - f_\varepsilon\|_1 + \|f_\varepsilon - f\|_1 \\
&< 2\epsilon + \|P_{\mathcal{P}_n} f_\varepsilon - f_\varepsilon\|_1.
\end{aligned}
$$

Let $E_n \subset \Omega$ be the union of all sets in $\mathcal{P}_n$ that contain at least on discontinuity of $f_\varepsilon$. Since $\mathcal{P}_n$ is an equipartition, we have $\mu(E_n) \leq \frac{2M}{n}$. On the set $\Omega \setminus E_n$ the function $f_\varepsilon$ is constant on each set, then $P_{\mathcal{P}_n} f_\varepsilon = f_\varepsilon$. Therefore,

$$
\|P_{\mathcal{P}_n} f_\epsilon - f_\varepsilon\|_1 = \int_{E_n} |P_{\mathcal{P}_n} f_\epsilon - f_\varepsilon| \, d\mu \leq 2\|f_\varepsilon\|_\infty \mu(E_n) \leq \frac{4M\|f_\varepsilon\|}{n}.
$$

We obtain

$$
\|P_{\mathcal{P}_n} f - f\|_1 \leq 2\varepsilon + \frac{4M\|f_\varepsilon\|_\infty}{n}.
$$

As $n$ tends to infinity, the tern $\frac{4M\|f_\varepsilon\|_\infty}{n}$ tends to zero, hence there exists $N > 0$ such that for every $n > N$, $\frac{4M\|f_\varepsilon\|_\infty}{n} < \varepsilon$. Therefore, for every $n > N$,

$$
\|P_{\mathcal{P}_n} f - f\|_1 \leq 3\varepsilon,
$$

i.e.,

$$
\lim_{n \to \infty} \|P_{\mathcal{P}_n} f - f\|_1 = 0.
$$

For the second part of the theorem, let $g \in \mathcal{L}^1(\Omega)$. By the triangle inequality

$$
\|A_{\mathcal{P}_n} g - Ag\|_1 \leq \|P_{\mathcal{P}_n} A(P_{\mathcal{P}_n} g - g)\|_1 + \|P_{\mathcal{P}_n}(Ag) - Ag\|_1.
$$

By the first part of the proof, since $Ag \in \mathcal{L}^1(\Omega)$, we have $\|P_{\mathcal{P}_n}(Ag) - Ag\|_1 \to 0$ as $n \to \infty$. For the first tern, we have

$$
\|P_{\mathcal{P}_n} A(P_{\mathcal{P}_n} g - g)\|_1 \leq r\|P_{\mathcal{P}_n} g - g\|_1.
$$

Applying the first part of the proof again yields $\|P_{\mathcal{P}_n} g - g\|_1 \to 0$ and hence

$$
\lim_{n \to \infty} \|A_{\mathcal{P}_n} g - Ag\|_1 = 0.
$$

$\square$

Next, we prove the main result of this section: density of graph-signals in the space of bofop-signals with respect to the DIDM mover's distance. Since the distance between two bofop-signals is computed by the distance between their bofop-DIDMs, we show that given a bofop-signal, as we take finer equipartitions of the space $\Omega$, the sequence of bofop-DIDMs obtained from the corresponding projected bofop-signals converge in the DIDM Mover's distance.

The proof relies on the following technical lemma regarding convergence of pushforward measures with respect to a sequence of $\mathcal{L}^1$ functions that converge in norm.

**Lemma L.6.** *Let $(\Omega, \mu)$ be a probability space, and let $f_n, f \in \mathcal{L}^1(\Omega)$ such that $\lim_{n \to \infty} \|f_n - f\|_1 = 0$. Let $\nu_n = (f_n)_* \mu$ and $\nu = f_* \mu$ be their respective pushforward measures. Then $\nu_n$ converges weakly to $\nu$.*

*Proof.* Recall from Remark G.9 that convergence in Wasserstein distance implies weak convergence. Hence, it is sufficient to show that $W(\nu_n, \nu) \to 0$ (recall Definition G.1). Let $\gamma_n = (f_n, f)_* \mu$ be a sequence of couplings of $\nu_n$ and $\nu$. Then

$$
W(\nu_n, \nu) \leq \int_{\mathbb{R}^d \times \mathbb{R}^d} |x - y| \, d\gamma_n(x, y) = \int_\Omega |f_n(\omega) - f(\omega)| \, d\mu = \|f_n - f\|_1 \to 0.
$$

We obtain that $W(\nu_n, \nu) \to 0$. $\square$

**Theorem L.7.** *Let $(\Omega, \mu)$ be a standard atomless probability space and $(A, f) \in \mathcal{BF}_d^r(\Omega)$ be a bofop-signal. Let $\{\mathcal{P}_n\}_{n=1}^{\infty}$ be a sequence of equipartitions of $\Omega$ into $n$ sets, with the corresponding sequence of projected bofop-signals $(A_{\mathcal{P}_n}, P_{\mathcal{P}_n} f)$. For any integer $L \geq 0$, let $\Gamma_{(A_{\mathcal{P}_n}, P_{\mathcal{P}_n} f), L}$ and $\Gamma_{(A, f), L}$ be their respective bofop-DIDMs of order $L$, denoted by $\Gamma_{n, L}$ and $\Gamma_L$ respectively. Then, as $n$ tends to infinity, we have that $\lim_{n \to \infty} \mathbf{OT}_{d_{\mathrm{IDM}}^L}(\Gamma_{n, L}, \Gamma_L) = 0$.*

*Proof.* Denote by $\gamma_{n, L} = \gamma_{(A_{\mathcal{P}_n}, P_{\mathcal{P}_n} f), L}$ and $\gamma_L = \gamma_{(A, f), L}$. We prove by induction on $L$ that $\lim_{n \to \infty} \int_{\Omega} d_{\mathrm{IDM}}^L(\gamma_{n, L}(x), \gamma_L(x)) \, d\mu(x) = 0$, which implies $\lim_{n \to \infty} \mathbf{OT}_{d_{\mathrm{IDM}}^L}(\Gamma_{n, L}, \Gamma_L) = 0$.

*Induction Base:* For $L = 0$, recall from Definition 5.1 that $\gamma_{n, 0} = P_{\mathcal{P}_n} f$ and $\gamma_0 = f$. Then, by the first part of Theorem L.5, $\lim_{n \to \infty} \| P_{\mathcal{P}_n} f - f \|_1 = 0$. By Lemma L.6, this implies $\Gamma_{n, 0} \to \Gamma_0$ weakly, and thus,

$$\lim_{n \to \infty} \mathbf{OT}_{d_{\mathrm{IDM}}^0}(\Gamma_{n, 0}, \Gamma_0) = 0.$$

*Induction Step:* For $L \geq 0$, assume that $\lim_{n \to \infty} \int_{\Omega} d_{\mathrm{IDM}}^L(\gamma_{n, L}(x), \gamma_L(x)) \, d\mu(x) = 0$. We will prove that

$$\lim_{n \to \infty} \int_{\Omega} d_{\mathrm{IDM}}^{L+1}(\gamma_{n, L+1}(x), \gamma_{L+1}(x)) \, d\mu(x) = 0.$$

First, by the recursive nature of the metric $d_{\mathrm{IDM}}^{L+1}$, for every $x \in \Omega$ we have

$$d_{\mathrm{IDM}}^{L+1}(\gamma_{n, L+1}(x), \gamma_{L+1}(x)) = d_{\mathrm{IDM}}^L(\gamma_{n, L}(x), \gamma_L(x)) + \mathbf{OT}_{d_{\mathrm{IDM}}^L}(\nu_{n, x}, \nu_x) \tag{22}$$

where $\nu_{n, x}(E) = A_{\mathcal{P}_n} \mathbb{1}_{\gamma_{n, L}^{-1}(E)}(x)$ and $\nu_x(E) = A \mathbb{1}_{\gamma_L^{-1}(E)}(x)$. Integrating both sides of equation (22) with respect to $\mu$ yields:

$$\int_{\Omega} d_{\mathrm{IDM}}^{L+1}(\gamma_{n, L+1}(x), \gamma_{L+1}(x)) d\mu(x) = \int_{\Omega} d_{\mathrm{IDM}}^L(\gamma_{n, L}(x), \gamma_L(x)) d\mu(x)$$
$$+ \int_{\Omega} \mathbf{OT}_{d_{\mathrm{IDM}}^L}(\nu_{n, x}, \nu_x) \, d\mu(x). \tag{23}$$

By the induction hypothesis, the first term on the right-hand side has vanishing limit as $n \to \infty$. To bound the second term, we define a "mixed" measure $\tilde{\nu}_{n, x}$ on $\mathcal{H}^L$ by $\tilde{\nu}_{n, x}(E) = A_{\mathcal{P}_n}(\mathbb{1}_{\gamma_L^{-1}(E)})(x)$. By the triangle inequality

$$\int_{\Omega} \mathbf{OT}_{d_{\mathrm{IDM}}^L}(\nu_{n, x}, \nu_x) \, d\mu(x) \leq \int_{\Omega} \mathbf{OT}_{d_{\mathrm{IDM}}^L}(\nu_{n, x}, \tilde{\nu}_{n, x}) \, d\mu(x) + \int_{\Omega} \mathbf{OT}_{d_{\mathrm{IDM}}^L}(\tilde{\nu}_{n, x}, \nu_x) \, d\mu(x). \tag{24}$$

To bound the first term, we construct an explicit coupling between $\nu_{n, x}$ and $\tilde{\nu}_{n, x}$. Notice that the total masses are identical, as both are $A_{\mathcal{P}_n}(\mathbb{1}_{\Omega})(x)$. We define the joint measure $\pi_x$ on $\mathcal{H}^L \times \mathcal{H}^L$ as the joint pushforward mapping

$$\pi_x(E \times F) = A_{\mathcal{P}_n}\left(\mathbb{1}_{\gamma_{n, L}^{-1}(E) \cap \gamma_L^{-1}(F)}\right)(x). \tag{25}$$

Since the marginals of $\pi_x$ match $\nu_{n, x}$ and $\tilde{\nu}_{n, x}$ respectively, it is a coupling. Because the optimal transport distance is an infimum over all couplings, it is bounded from above by the integral cost of $\pi_x$. Then

$$\mathbf{OT}_{d_{\mathrm{IDM}}^L}(\nu_{n, x}, \tilde{\nu}_{n, x}) \leq \int_{\mathcal{H}^L \times \mathcal{H}^L} d_{\mathrm{IDM}}^L(u, v) d\pi_x(u, v)$$
$$= A_{\mathcal{P}_n}\left(d_{\mathrm{IDM}}^L(\gamma_{n, L}, \gamma_L)\right)(x),$$

where in the last equality we applied Lemma K.2. Integrating over $\Omega$ yields $\int_{\Omega} \mathbf{OT}_{d_{\mathrm{IDM}}^L}(\nu_{n, x}, \tilde{\nu}_{n, x}) \, d\mu(x) \leq r \int_{\Omega} d_{\mathrm{IDM}}^L(\gamma_{n, L}(y), \gamma_L(y)) d\mu(y)$, which goes to 0 as $n \to \infty$ by the induction assumption.

For the second term, $\mathbf{OT}_{d_{\mathrm{IDM}}^L}(\tilde{\nu}_{n, x}, \nu_x)$, the signal $\gamma_L$ is fixed, and the measures differ only by the operators $A_{\mathcal{P}_n}$ and $A$. Since $\mathbf{OT}_{d_{\mathrm{IDM}}^L}$ metrizes the weak topology on $\mathcal{M}_{\leq r}(\mathcal{H}^L)$, it is sufficient to prove that for every bounded continuous function $\phi : \mathcal{H}^L \to \mathbb{R}$ we have

$$\lim_{n \to \infty} \int_{\Omega} \left[ \int_{\mathcal{H}^L} \phi \, d\tilde{\nu}_{n, x} - \int_{\mathcal{H}^L} \phi \, d\nu_x \right] d\mu(x) = 0.$$

Using again Lemma K.2 we get

$$\lim_{n\to\infty} \int_\Omega |A_{\mathcal{P}_n}(\phi \circ \gamma_L)(x) - A(\phi \circ \gamma_L)(x)| \, d\mu(x) = \lim_{n\to\infty} \|A_{\mathcal{P}_n}(\phi \circ \gamma_L) - A(\phi \circ \gamma_L)\|_1.$$

Since $\phi \circ \gamma_L : \Omega \to \mathbb{R}$ is bounded, we apply the second part of Theorem L.5 and we obtain

$$\int_\Omega \mathbf{OT}_{d_{\mathrm{IDM}}^L}(\tilde{\nu}_{n,x}, \nu_x) \, d\mu(x) \to 0,$$

as $n \to \infty$.

Recall that

$$\mathbf{OT}_{d_{\mathrm{IDM}}^L}(\tilde{\nu}_{n,x}, \nu_x) = \inf_{\gamma_x \in \Gamma(\tilde{\nu}_{n,x}, \nu_x)} \int_{\mathcal{H}^L \times \mathcal{H}^L} d_{\mathrm{IDM}}^L(u,v) \, d\gamma_x(u,v) + |A_{\mathcal{P}_n} \mathbb{1}_\Omega(x) - A\mathbb{1}_\Omega(x)|.$$

Consider the tern $|A_{\mathcal{P}_n} \mathbb{1}_\Omega(x) - A\mathbb{1}_\Omega(x)|$, when integrating over $\Omega$ we get

$$\int_\Omega |A_{\mathcal{P}_n} \mathbb{1}_\Omega(x) - A\mathbb{1}_\Omega(x)| \, d(\mu(x) = \|A_{\mathcal{P}_n} \mathbb{1}_\Omega - A\mathbb{1}_\Omega\|_1 \to 0,$$

as $n \to \infty$ because of the second part of Theorem L.5.

Since both terms converge to zero, equation (23) implies:

$$\lim_{n\to\infty} \int_\Omega d_{\mathrm{IDM}}^{L+1}(\gamma_{n,L+1}(x), \gamma_{L+1}(x)) \, d\mu(x) = 0$$

which consequently implies $\lim_{n\to\infty} \mathbf{OT}_{d_{\mathrm{IDM}}^{L+1}}(\Gamma_{n,L+1}, \Gamma_{L+1}) = 0$, thus completing the proof. $\qquad \square$

We finish this section with the density of graph-signals in the space of bofop-signals with respect to the DIDM mover's distance. Actually, since the adjacency matrix can be normalized in various ways we consider graphs as GSOs and state the the result for weighted graph shift operators rather than for unweighted graphs alone.

Note that since in our setting, we consider bofops with uniformly bounded $\infty \to \infty$ (or equivalently $1 \to 1$) norms, when considering the subclass of GSOs, the norm is simply the maximal degree of the weighted graph obtained from the GSO. Therefore, the following result shows that the class of weighted graphs with maximal degree uniformly bounded by some $r > 0$ is dense in $\mathcal{BF}_d^r(\Omega)$.

**Corollary L.8.** *Let $r > 0$, $d \geq 0$ and $L \geq 0$. Let $(A, f) \in \mathcal{BF}_d^r(\Omega)$ be a bofop-signal over some Borel probability space $(\Omega, \mu)$. Then there exists a sequence of graph-signals $(G_n, \mathbf{f}_n)$ with a corresponding sequence of GSOs $S_n$ such that*

$$\lim_{n\to\infty} \delta_{\mathrm{DIDM}}^L((A,f), (S_n, \mathbf{f}_n)) = 0.$$

*Proof.* First, recall from Proposition L.1 that we can assume that $\Omega$ is atomless. Indeed, if $\Omega$ has atoms, we simply replace it by $\Omega' = \Omega \times [0,1]$ and we have $\delta_{\mathrm{DIDM}}^L((A,f), (A', f')) = 0$, and thus for every $n \in \mathbb{N}$,

$$\delta_{\mathrm{DIDM}}^L((A,f), (S_n, \mathbf{f}_n)) = \delta_{\mathrm{DIDM}}^L((A', f'), (S_n, \mathbf{f}_n)).$$

Let $\mathcal{P}_n$ be a equipartition of $\Omega$ into $n$ sets, and let $(A_{\mathcal{P}_n}, P_{\mathcal{P}_n} f)$ be the corresponding bofop-signals. From Lemma L.3 each projected bofop-signal acts as a graph-signal with the corresponding GSO $S_n$ with entries $(S_n)_{i,j} = n(\mathbb{1}_{I_k}, \mathbb{1}_{I_j})_A$. Let $\Gamma_{n,L}$ be the sequence of bofop-DIDMs obtained from applying the 1-WL algorithm (Definition 5.1) on the projected bofop-signals $(A_{\mathcal{P}_n}, P_{\mathcal{P}_n} f)$. Then, by Theorem L.7 we have

$$\lim_{n\to\infty} \mathbf{OT}_{d_{\mathrm{IDM}}^L}(\Gamma_{(A,f),L}, \Gamma_{n,L}) = 0.$$

Equivalently,

$$\lim_{n\to\infty} \delta_{\mathrm{DIDM}}^L((A,f), (A_{\mathcal{P}_n}, f_{\mathbf{f}}^{\mathcal{P}_n})) = 0,$$

whcih concludes the proof. $\qquad \square$

# M. Compactness of The Space of Bofop-DIDM

In this section, we will use the two topologies that we defined on the space of bofop-signals $\mathcal{BF}_d^r(\Omega)$, metrized by the metrics $d_M$ and $\delta_{\mathrm{DIDM}}^L$. We will prove that the topology that the action metric metrize is finer the topology metrized by the DIDM mover's distance. Since we proved in Corollary H.21 that the space $(\mathcal{BF}_d^r, d_M)$ is compact, it implies that the space $(\mathcal{BF}_d^r, \delta_{\mathrm{DIDM}}^L)$ is compact as well.

## M.1. Action Topology is Finer Than DIDM Mover's Topology

In this section, we show that for a sequence of bofop-signals, convergence in the action metric $d_M$ implies convergence in the DIDM mover's distance $\delta_{\mathrm{DIDM}}^L$. The proof relies on the continuity of MPNNs (Theorem J.1 together with Theorem E.11, which characterizes convergence of DIDMs in terms of convergence of MPNN outputs. Since all spaces under consideration are metric spaces, their topologies are completely determined by their metrics. Therefore, the topology induced by the action metric is finer than the topology induced by the DIDM mover's distance. We formalize this implication in the following theorem.

**Theorem M.1.** *Let $(A_i, f_i)_{i \in \mathbb{N}} \in \mathcal{BF}_d^r(\Omega_i)$ be a sequence of bofop-signals over a sequence of Borel probability spaces $(\Omega_i, \mu_i)$. Also, let $(A, f) \in \mathcal{BF}_d^r(\Omega)$ be a bofop-signal over a Borel probability space $(\Omega, \mu)$. If $\lim_{n \to \infty} d_M\left((A_n, f_n), (A, f)\right) = 0$, then $\lim_{n \to \infty} \delta_{DIDM}^L\left((A_n, f_n), (A, f)\right) = 0$.*

*Proof.* Recall from definition H.13 that a sequence of bofop-signal action converges if for every $k$ the sequence of $k$-profiles is a Cauchy sequence and therefore converges by the completeness of the spaces. Let $\mathrm{S}_k^{(n)}$ denote the $k$-profile of the bofop-signal $(A_n, f_n)$, and let $\mathrm{S}_k$ be the $k$-profile of $(A, f)$. By theorem J.2 MPNNs on profiles are Hölder continuous functions with respect to Hausdorff distance, in particular they are continuous, and therefore $\lim_{n \to \infty} \|\mathfrak{S}(\mathrm{S}_k^{(n)}) - \mathfrak{S}(\mathrm{S}_k)\|_2 = 0$. By our commutation property (theorem I.9) we have that application of MPNN on profiles is equivalent to the profiles of the MPNN on P-operator-signals. So we have for every $k \geq 0$

$$\lim_{n \to \infty} d_H\left(\mathrm{S}_k(A_n, \mathfrak{h}_{(A_n, f_n)}^{(L)}), \mathrm{S}_k(A, \mathfrak{h}_{(A, f)}^{(L)})\right) = 0,$$

and by definition of action convergence (definition H.13) we get $d_M((A_n, \mathfrak{h}_{(A_n, f_n)}^{(L)}), (A, \mathfrak{h}_{(A, f)}^{(L)})) = 0$. Since application of MPNN on P-operator-signal is equivalent to the application of the MPNN on the induced IDMs and DIDMs ( Lemma K.3), we have that

$$\mathscr{H}_{\Gamma_{(A_i, f_i)}, L} \to \mathscr{H}_{\Gamma_{(A, f)}, L}.$$

Lastly, since convergence of DIDMs is equivalent to the convergence of every MPNN on the DIDMs, we conclude

$$\lim_{n \to \infty} \delta_{\mathrm{DIDM}}^L\left((A_n, f_n), (A, f)\right) = 0.$$

$\square$

## M.2. Compactness of The Space of Bofop-DIDM

We now move to prove the compactness of bofop-DIDMs using the above discussion.

**Theorem M.2.** *For every $L \geq 0$, the space of bofop-DIDMs $\Gamma_L(\mathcal{BF}_d^r) \subsetneq \mathscr{P}(\mathcal{H}^L)$ is compact.*

*Proof.* By Theorem M.1, the action topology is finer than the P-DIDM mover's distance topology, and since $(\mathcal{BF}_d^r, d_M)$ is compact, it implies that $(\mathcal{BF}_d^r, \delta_{\mathrm{DIDM}}^L)$ is compact. By the definition of $\delta_{\mathrm{DIDM}}^L$, the map $\Gamma_{(A, f), L} : \mathcal{BF}_d^r \to \mathcal{P}(\mathcal{H}^L)$ is continuous. Therefore, the space $\Gamma_{(A, f), L}(\mathcal{BF}_d^r)$ is compact, because a continuous function between metric spaces maps compact sets to compact sets. $\square$

Similarly to the fact that graphons form a strict subset of the space of all DIDMs by Example E.7, an analogue construction for bofop-signals shows that the space of bofop-DIDMs is also a strict subset of the space of all DIDMs.

### M.3. Hierarchy of DIDM Spaces - Dense Graphs, Graphons, Sparse Graphs, and Graphops

We have proven the compactness of the space of bofop-DIDMs $\Gamma_L(\mathcal{BF}_d^r)$. Moreover, DIDMs obtained via 1-WL on graphon-signals, which we refer to as *graphon-DIDMs*, are already known to form a compact subspace of the space of all DIDMs (Corollary 65 in (Rauchwerger et al., 2025)). Therefore, the space of graphon-DIDMs is a strict compact subspace of the bofop-DIDM space, which is a strict subset of the space of all DIDMs, which is compact itself. This established a hierarchy of DIDM spaces, useful for theoretical results in graph machine learning. The results are summarized in figure 1.

## N. Universal Approximation and Generalization Bounds of MPNNs for Sparse Graphs

In this section we prove our main results of the paper - universal approximation theorem and generalization theorem for MPNNs on bofop-signals

### N.1. Universal Approximation of DIDM-Mover's Continuous Functions By MPNNs

Universal approximation theorem for continuous functions on the space of all DIDMs was established in (Rauchwerger et al., 2025). We extend this result to the setting of bofop-signals,, proving a universal approximation theorem specifically for continuous functions over the space of bofop-DIDMs.

A priori, it is not clear whether a continuous function defined only on the space of bofop-DIDMs admits a continuous extension to the entire space of DIDMs. Without such extension, the universal approximation theorem of (Rauchwerger et al., 2025) cannot be applied directly. Our analysis resolves this issue by employing the compactness of bofop-DIDMs. Since the space of bofop-DIDMs is compact, any continuous function defined on it admits a continuous extension to the full DIDM space.

Building on that, we extend the universal approximation theorem to the bofop setting. Specifically, we prove a universal approximation theorem for continuous functions on the space of bofop-DIDMs. This establishes that the expressive power known for DIDMs persists when restricting attention to bofop-DIDMs.

Recall from Appendix E the notation for the sets of L-layer MPNN models $\mathcal{N}_L^{d_L} \subseteq C(\mathcal{H}^L, \mathbb{R}^{d_L})$, where $C(\mathcal{H}^L, \mathbb{R}^{d_L})$ is the set of all continuous functions $\mathcal{H}^L \mapsto \mathbb{R}^{d_L}$, and the set of all L-layer MPNN with readout Similarly we define the set $\mathcal{NN}_L^p \subseteq C(\mathscr{P}(\mathcal{H}^L), \mathbb{R}^p)$ as the set of L-layer MPNN models with readout.

To restrict the discussion to bofop-DIDMs we define the set $\mathcal{NN}_L^p(\mathcal{BF}_d^r) := \{\mathfrak{H}^{(\varphi, \psi)}(-, -) : \mathcal{BF}_d^r \to \mathbb{R}^p | (\varphi, \psi)$ is an L-layer MPNN model with readout}. We can now prove universal approximation theorem for bofop-signals.

**Theorem N.1.** *Let $L \in \mathbb{N}_0$. Then, the set $\mathcal{NN}_L^1(\mathcal{BF}_d^r)$ is uniformly dense in $C(\mathcal{BF}_d^r, \mathbb{R})$.*

*Proof.* By Corollary 5.3 the space of bofop-DIDMs $(\Gamma_L(\mathcal{BF}_d^r), \mathbf{OT}_{d_{\mathrm{IDM}}^L})$ is compact. Since we can identify the space $(\mathcal{BF}_d^r, \delta_{\mathrm{DIDM}}^L)$ with $(\Gamma_L(\mathcal{BF}_d^r), \mathbf{OT}_{d_{\mathrm{IDM}}^L})$ (by Definition K.5), we have compactness of $(\mathcal{BF}_d^r, \delta_{\mathrm{DIDM}}^L)$. Continuity of MPNNs follows from Theorem E.10, as the space of Bofop-DIDMs is a subset of the space of all DIDMs. Since $\mathcal{NN}_L^1(\Gamma_L(\mathcal{BF}_d^r))$ is a subalgebra of $C(\mathcal{P}(\mathcal{H}^L), \mathbb{R})$, by Lemma 78 in (Rauchwerger et al., 2025), $\mathcal{NN}_L^1(\mathcal{BF}_d^r)$ satisfies the conditions of the Stone-Weierstrass Theorem C.4. $\square$

### N.2. Generalization Bound for MPNNs on Sparse Graph Data Distributions

In this section, we prove A generalization theorem for MPNN on the space of bofop-signals with respect to the DIDMs-mover's distance $\delta_{\mathrm{DIDM}}^L$. While a related generalization bound could also be derived using the action metric $d_M$, relying on the Hölder continuity of MPNNs and compactness of the space, our analysis deliberately focuses on $\delta_{\mathrm{DIDM}}^L$. As shown in theorem M.1, the topology induced by $d_{\mathrm{DIDM}}^L$ is coarser than the topology metrized by the action metric. Therefore, we can conclude that the covering number of the space $(\mathcal{BF}_d^r, \delta_{\mathrm{DIDM}}^L)$ is smaller than the one of $(\mathcal{BF}_d^r, d_M)$.

While compactness guarantee the existence of finite covering, an explicit bound on the covering number of the space of graphon-DIDMs remains unknown. Extending beyond graphons, it remains open to determine corresponding covering bounds for the larger space of bofop-DIDMs, which itself is smaller than the unknown bound of bofop-signals, with respect to the action metric.

### N.2.1. CLASSIFICATION SETTING

We extend the generalization analysis from Appendix G2 in(Levie, 2024) and Appendix K in(Rauchwerger et al., 2025), and adjust it to meet our definitions. While our focus is on the space of Bofop-DIDMs, where MPNNs are Lipschitz continuous, we formulate the setting more generally, and consider classes of Hölder continuous functions. This formulation extends the settings presented in (Levie, 2024; Rauchwerger et al., 2025), and is suited for the space of graphop-signals equipped with the action-metric, where MPNNs are Hölder continuous. Notably, this formulation accommodates the Lipschitz case, as Lipschitz continuity is a particular instance of Hölder continuity.

Let $(\mathcal{BF}_d^r, \delta_{\text{DIDM}}^L)$ be the space of bofop-signals with the DIDM mover's distance. We define ground truth classifier to $C$ classes as follows. Let $\mathcal{C} : \mathcal{BF}_d^r \to \mathbb{R}^C$ be a measurable piecewise constant function of the following form. Equipped with $\delta_{\text{DIDM}}^L$ the space $(\mathcal{BF}_d^r, \delta_{\text{DIDM}}^L)$ is compact. Hence, there is a partition of $\mathcal{BF}_d^r$ into disjoint measurable sets $B_1 \ldots, B_C \subset \mathcal{BF}_d^r$ such that $\cup_{i=1}^C B_i = \mathcal{BF}_d^r$, and for every $i \in [C]$ and $x \in B_i$,

$$\mathcal{C}(x) = e_i,$$

where $e_i \in \mathbb{R}^C$ is the standard basis element with entries $(e_i)_j = \delta_{i,j}$, where $\delta_{i,j}$ is the Kronecker delta.

We define an arbitrary data distribution as follows. Let $\mathcal{B}$ be the Borel $\sigma$-algebra of $\mathcal{BF}_d^r$ and let $\nu$ be any probability measure on the measurable space $(\mathcal{BF}_d^r, \mathcal{B})$. We may assume that we complete $\mathcal{B}$ with respect to $\nu$, obtaining $\sigma$-algebra $\Sigma$. If we do not complete the measure, we just denote $\Sigma = \mathcal{B}$. Whether $(\mathcal{BF}_d^r, \Sigma, \nu)$ is defined as a complete measure space or not does not affect our construction.

Note that by Theorem J.1, for every $i \in [C]$, the space of MPNN with Lipschitz update functions and a Lipschitz readout function, restricted to $B_i$, is a subset of $Hol^\alpha(B_i, L_1)$ which is a restriction of $Hol^\alpha(\mathcal{BF}_d^r, L_1)$ to $B_i \subset \mathcal{BF}_d^r$ for some $L_1 > 0$ and $0 < \alpha \leq 1$. Let $\mathcal{E}$ be a Lipschitz continuous loss function with Lipchitz constant $L_2$. Therefore, since $\mathcal{C} \in Lip(B_i, 0)$, for any $\Gamma \in Hol^\alpha(\mathcal{BF}_d^r, L_1)$, the function $\mathcal{E}(\Gamma|_{B_i}, \mathcal{C}|_{B_i})$ is in $Hol^\alpha(B_i, L_1 L_2)$.

### N.2.2. UNIFORM MONTE CARLO APPROXIMATION OF HÖLDER CONTINUOUS FUNCTIONS

The proof of the generalization theorem (Theorem N.5) is based on the following result, which establishes uniform Monte Carlo approximation bounds for Hölder continuous functions over metric spaces with finite covering. This extends previous results such as (Levie, 2024; Xu & Mannor, 2012; Rauchwerger et al., 2025), which were proved for the Lipschitz case.

**Definition N.2** (Covering Number). A metric space $\Omega$ is said to have a covering number $\kappa : (0, \infty) \to \mathbb{N}$ if for $\epsilon > 0$, $\Omega$ can be covered by $\kappa(\epsilon)$ balls with radius $\epsilon$.

**Theorem N.3** (Uniform Monte Carlo Approximation For Hölder Continuous Functions ). *Let $\mathcal{X}$ be a metric probability space, with probability measure $\mu$, and covering number $\kappa(\epsilon)$. Let $X_1, \ldots, X_N$ be drawn i.i.d. from $\mathcal{X}$. Then for every $p > 0$, there exists an event $\mathcal{E}_{\text{Hol}}^p \subset \mathcal{X}^N$, (regarding the choice of $(X_1, \ldots, X_N)$), with probability*

$$\mu^N(\mathcal{E}_{\text{Hol}}^p) \geq 1 - p,$$

*such that for every $(X_1, \ldots, X_N) \in \mathcal{E}_{\text{Hol}}^p$, for every bounded Hölder continuous function $F : \mathcal{X} \to \mathbb{R}^d$ with Hölder constant $L_F$ and exponent $\alpha > 0$, we have*

$$\left\| \int F(x) \, d\mu(x) - \frac{1}{N} \sum_{i=1}^N F(X_i) \right\|_\infty \leq$$

$$2(\xi^{-1}(N))^\alpha L_F + \frac{1}{\sqrt{2}} (\xi^{-1}(N))^\alpha \|F\|_\infty \left( 1 + \sqrt{\log(2/p)} \right),$$

*where $\xi(r) = \frac{\kappa(r)^2 \log(\kappa(r))}{r^{2\alpha}}$ and $\xi^{-1}$ is the inverse function of $\xi$.*

The proof will use the following well known *Hoffeding's inequality*.

**Theorem N.4** (Hoffeding's Inequality). *Let $X_1, \ldots, X_N$, be independent random variables such that $a \leq X_i \leq b$ for every $i \in [N]$ almost surely. Then, for every $k > 0$,*

$$\mathbb{P}\left( \frac{1}{N} \left| \sum_{i=1}^N (X_i - \mathbb{E}[X_i]) \right| \geq k \right) \leq 2 \cdot exp\left( -\frac{2k^2 N}{(b-a)^2} \right).$$

The following is a proof for theorem N.3

*Proof.* Let $r > 0$. There exists a covering of $\mathcal{X}$ by a set of $K := \kappa(r)$ balls $\{B_j\}_{j \in [K]}$, of radius $r$. Define $J_1 := B_1$ and for $j = 2, \ldots, K$, define $I_j := B_j \setminus \cup_{i<j} B_i$. Then, $\{I_j\}_{j \in [K]}$ is a pairwise disjoint family of measurable sets that covers $\mathcal{X}$, and for every $j \in [K]$, $\operatorname{diam}(I_j) \leq 2r$, (where $\operatorname{diam}(\emptyset) = 0$). For each $j \in [K]$ let $c_j$ denote the center of the ball $B_j$.

Next, we derive a uniform concentration bound on the error between the measure of $I_j$ and its Monte Carlo approximation, valid for all $j \in [K]$. Let $j \in [K]$ and $q \in (0, 1)$, by theorem N.4, there is an event $\mathcal{E}_j^q$ with probability $\mu(\mathcal{E}_j^q) \geq 1 - q$, in which

$$\left\| \frac{1}{N} \sum_{i=1}^N \mathbb{1}_{I_j}(X_i) - \mu(I_j) \right\|_\infty \leq \frac{1}{\sqrt{2}} \cdot \frac{\sqrt{\log(\frac{2}{q})}}{\sqrt{N}}. \tag{26}$$

Consider the event

$$\mathcal{E}_{\mathrm{Hol}}^{Kq} = \cap_{j=1}^K \mathcal{E}_j^q,$$

with probability $\mu^N(\mathcal{E}_{\mathrm{Hol}}^{Kq}) \geq 1 - Kq$. In this event, 26 holds for every $j \in [K]$. We write $p = Kq$ and denote $\mathcal{E}_{\mathrm{Hol}}^p = \mathcal{E}_{\mathrm{Hol}}^{Kq}$.

Next, we bound uniformly the Monte Carlo approximation error of the integral of bounded Hölder continuous functions $F : \mathcal{X} \to \mathbb{R}^d$. We define the step function

$$F^r(x) = \sum_{j=1}^K F(c_j) \mathbb{1}_{I_j}(x).$$

Then,

$$\begin{aligned}
&\left\| \frac{1}{N} \sum_{i=1}^N F(X_i) - \int_{\mathcal{X}} F(x)\, d\mu(x) \right\|_\infty \leq \\
&\left\| \frac{1}{N} \sum_{i=1}^N F(X_i) - \frac{1}{N} \sum_{i=1}^N F^r(X_i) \right\|_\infty + \\
&\left\| \sum_{i=1}^N F^r(X_i) - \int_{\mathcal{X}} F^r(x)\, d\mu(x) \right\|_\infty + \\
&\left\| \int_{\mathcal{X}} F^r(x)\, d\mu(x) - \int_{\mathcal{X}} F(x)\, d\mu(x) \right\|_\infty = \\
&(1) + (2) + (3).
\end{aligned} \tag{27}$$

To bound (1), we define for each $X_i$ the unique index $j_i \in [K]$ for which $X_i \in I_{j_i}$. We calculate

$$\begin{aligned}
\left\| \frac{1}{N} \sum_{i=1}^N F(X_i) - \frac{1}{N} \sum_{i=1}^N F^r(X_i) \right\|_\infty &\leq \frac{1}{N} \sum_{i=1}^N \left\| F(X_i) - \sum_{j \in [K]} F(c_j) \mathbb{1}_{I_{j_i}}(X_i) \right\|_\infty \\
&\leq \frac{1}{N} \sum_{i=1}^N \left\| F(X_i) - F(c_{j_i}) \right\|_\infty \\
&\leq \frac{L_F}{N} \sum_{i=1}^N \left\| X_i - c_{j_i} \right\|_\infty^\alpha \leq L_F \cdot r^\alpha.
\end{aligned}$$

We next bound (2). In the event $\mathcal{E}_{\mathrm{Hol}}^p$, which holds with probability $1 - p$, equation 27 holds for every $j \in [K]$. In this event,

we get

$$\left\|\frac{1}{N}\sum_{i=1}^{N}F^r(X_i) - \int_{\mathcal{X}}F^r(x)\,d\mu(x)\right\|_{\infty} = \left\|\sum_{j=1}^{K}\left(\frac{1}{N}\sum_{i=1}^{N}F(c_i)\mathbb{1}_{I_{j_i}}(X_i) - \int_{I_j}F(c_j)\,d\mu(x)\right)\right\|_{\infty}$$

$$\leq \sum_{j=1}^{K}\|F\|_{\infty}\left|\frac{1}{N}\sum_{j=1}^{N}\Bbbk_{I_j}(X_i) - \mu(I_j)\right|$$

$$\leq K\|F\|_{\infty}\frac{1}{\sqrt{2}}\frac{\sqrt{\log(2K/p)}}{\sqrt{N}}.$$

Then, with probability at least $1 - p$

$$\left\|\frac{1}{N}\sum_{i=1}^{N}F^r(X_i) - \int_{\mathcal{X}}F^r(x)\,d\mu(x)\right\|_{\infty} \leq \frac{K\|F\|_{\infty}}{\sqrt{2}}\frac{\sqrt{\log(K) + \log(2/p)}}{\sqrt{N}}.$$

To bound (3), we calculate

$$\left\|\int_{\mathcal{X}}F^r(x)\,d\mu(x) - \int_{\mathcal{X}}F(x)\,d\mu(x)\right\|_{\infty} = \left\|\int_{\mathcal{X}}\sum_{j=1}^{K}F(c_j)\mathbb{1}_{I_j}\,d\mu(x) - \int_{\mathcal{X}}F(x)\,d\mu(x)\right\|_{\infty}$$

$$\leq \sum_{j=1}^{K}\int_{I_j}\left\|F(c_j) - F(x)\right\|_{\infty}d\mu(x) \leq L_F \cdot r^{\alpha}.$$

By plugging everything to 27, we get

$$\left\|\frac{1}{N}\sum_{i=1}^{N}F(X_i) - \int_{\mathcal{X}}F(x)\,d\mu(x)\right\|_{\infty} \leq 2L_F \cdot r^{\alpha} + \frac{K\|F\|_{\infty}}{\sqrt{2}}\frac{\sqrt{\log(K) + \log(2/p)}}{\sqrt{N}}$$

$$\leq 2L_F \cdot r^{\alpha} + \frac{K\|F\|_{\infty}}{\sqrt{2}}\frac{\sqrt{\log(K)} + \sqrt{\log(2/p)}}{\sqrt{N}}$$

$$\leq 2L_F \cdot r^{\alpha} + \frac{K\|F\|_{\infty}}{\sqrt{2}}\frac{\sqrt{\log(K)}}{\sqrt{N}}\left(1 + \sqrt{\log(2/p)}\right).$$

Lastly, choosing $r = \xi^{-1}(N)$, for $\xi(r) = \frac{\kappa(r)^2\log(\kappa(r))}{r^{2\alpha}}$, gives $\sqrt{N} = \frac{\kappa(r)\sqrt{\log(\kappa(r))}}{r^{\alpha}}$, so

$$\left\|\int F(x)\,d\mu(x) - \frac{1}{N}\sum_{i=1}^{N}F(X_i)\right\|_{\infty} \leq$$

$$2(\xi^{-1}(N))^{\alpha}L_F + \frac{1}{\sqrt{2}}(\xi^{-1}(N))^{\alpha}\|F\|_{\infty}\left(1 + \sqrt{\log(2/p)}\right),$$

Since the event $\mathcal{E}^p_{\text{Hol}}$ is independent of the choice of $F$, the proof is finished.

$\square$

### N.3. Generalization Theorem For MPNNs

We can now prove a generalization theorem to MPNNs on profiles, using theorem N.3.

**Theorem N.5.** *Consider the classification setting from subsection N.2.1. Let $X_1, \ldots, X_N$ be independent random samples from $(\mathcal{BF}^r_d, \Sigma, \nu)$. Then, for every $p > 0$ there is an event $(\mathcal{BF}^r_d)^N$, regarding the choice of $(X_1, \ldots, X_N)$, with probability*

$$\nu^N(\mathcal{E}^p) \geq 1 - Cp - \frac{4C^2}{N},$$

*in which for every function $\Gamma$ in the hypothesis class $Hol^\alpha(\mathcal{BF}_d^r, L_1)$, we have*

$$|R_{\text{stat}}(\Gamma_X) - R_{\text{Emp}}(\Gamma_X)| \leq$$

$$2(\xi^{-1}(N/2C))^\alpha L_1 L_2 + \frac{1}{\sqrt{2}}(\xi^{-1}(N/2C))^\alpha \Big(L_1 L_2 + \mathcal{E}(0,0)\Big) L_1 L_2 \Big(1 + \sqrt{\log(2/p)}\Big).$$

*where, $\xi(r) = \frac{\kappa(r)^2 log(\kappa(r))}{r^{2\alpha}}$, $\kappa$ is the covering number of $\mathcal{BF}_d^r$ and $\xi^{-1}$ is the inverse function of $\xi$.*

Note that as the size of the training set $N$ tends to infinity, the tern $\xi^{-1}(N)$ decreases to 0.

*Proof.* For every $i \in [C]$, let $N_i$ be number of samples in $\mathcal{BF}_d^r$ that falls within $B_i$. Consider the multidimensional random variable $(N_1, \ldots, N_C)$. The expected value is $(N/C, \ldots, N/C)$, and the variance is $\left(\frac{N(C-1)}{N^2}, \ldots, \frac{N(C-1)}{C^2}\right) \leq (N/C, \ldots, N/C)$. By Chebyshev's inequality, for every $a > 0$,

$$P\Big(|N_i - N/C| > a\sqrt{N/C}\Big) < \frac{1}{a^2}.$$

We take $a = \frac{1}{2}\sqrt{N/C}$, so $a\sqrt{N/C} = \frac{N}{2C}$, and

$$P\Big(|N_i - N/C| > N/2C\Big) < \frac{4C}{N}.$$

Therefore,

$$P\Big(N_i > N/2C\Big) > 1 - \frac{4C}{N}.$$

We intersect these events an get an event, d denoted by $\mathcal{E}_{\text{int}}$, with probability greater than $1 - \frac{4C^2}{N}$, in which $N_i > N/2C$ for every $i \in [C]$. In the following, given a set $B_i$, we consider a realization $n := N_i$ and use the law of total probability. Using theorem N.3, for every $p > 0$, there exists an event $\mathcal{E}_{\text{Hol},i}^p \subset B_i^n$ regarding the choice of $(X_1, \ldots, X_n) \in B_i^n$ with probability

$$\nu^n(\mathcal{E}_{\text{Hol},i}^p) \geq 1 - p,$$

such that for every function $\Gamma \in Hol^\alpha(\mathcal{BF}_d^r, L_1)$, we have

$$\left| \int \mathcal{E}\big(\Gamma(x), \mathcal{C}(x)\big) \, d\nu(x) - \frac{1}{n}\sum_{i=1}^n \mathcal{E}\big(\Gamma(X_i), \mathcal{C}(X_i)\big) \right| \leq \tag{28}$$

$$2(\xi^{-1}(n))^\alpha L_1 L_2 + \frac{1}{\sqrt{2}}(\xi^{-1}(n))^\alpha \Big\| \mathcal{E}\big(\Gamma(\cdot), \mathcal{C}(\cdot)\big) \Big\|_\infty L_1 L_2 \Big(1 + \sqrt{\log(2/p)}\Big) \leq$$

$$2(\xi^{-1}(N/2C))^\alpha L_1 L_2 + \frac{1}{\sqrt{2}}(\xi^{-1}(N/2C))^\alpha \Big(L_1 L_2 + \mathcal{E}(0,0)\Big) L_1 L_2 \Big(1 + \sqrt{\log(2/p)}\Big),$$

*where, $\xi(r) = \frac{\kappa(r)^2 log(\kappa(r))}{r^{2\alpha}}$, $\kappa$ is the covering number of $\mathcal{BF}_d^r$, $\xi^{-1}$ is the inverse function of $\xi$. In the last inequality we used that for every $x \in \mathcal{BF}_d^r$,*

$$|\mathcal{E}(\Gamma(x), \mathcal{C}(x))| \leq |\mathcal{E}(\Gamma(x), \mathcal{C}(x)) - \mathcal{E}(0,0)| + |\mathcal{E}(0,0)| \leq L_2 L_1 + |\mathcal{E}(0,0)|.$$

Since 28 holds for every $\Gamma \in Hol^\alpha(\mathcal{BF}_d^r, L_1)$, it holds in particular to any realization of $X, \Gamma_X$. So

$$|R_{\text{stat}}(\Gamma_X) - R_{\text{Emp}}(\Gamma_X)| \leq$$

$$2(\xi^{-1}(N/2C))^\alpha L_1 L_2 + \frac{1}{\sqrt{2}}(\xi^{-1}(N/2C))^\alpha \Big(L_1 L_2 + \mathcal{E}(0,0)\Big) L_1 L_2 \Big(1 + \sqrt{\log(2/p)}\Big).$$

Lastly, denote

$$\mathcal{E}^p = \mathcal{E}_{\text{int}} \cap \Big( \bigcup_{i \in [C]} \mathcal{E}_{\text{Hol},i}^p \Big).$$

$\square$

*Additional Background*

# O. Other models of sparse graphs

Here, we present the foundational theory for three models of sparse graphs: stretched graphons, $\mathcal{L}^p$ graphons and graphings, all of which are examples of graphops.

### O.1. Stretched Graphons

Graphons have typically been used as limit objects for dense graph sequences, utilizing the cut distance as the metric for convergence. However, sparse graph sequences converge to the zero graphon $W \equiv 0$ under the conventional definition of cut distance, which make this framework inadequate for many practical applications (Ji et al., 2024; Jian et al., 2023). To address this, stretched graphons and the stretched cut distance have been introduced in (Borgs et al., 2018a), providing a more suitable framework for analyzing the limits and convergence of sparse graph sequences.

**Definition O.1.** For a graphon $W : [0,1]^2 \to [0,1]$, we define the stretched graphon $W^s$ by

$$W^s(x,y) = W(\|W\|_1^{\frac{1}{2}} x, \|W\|_1^{\frac{1}{2}} y).$$

The motivation behind this definition is that for sparse graphons, their $\mathcal{L}^1$ norm is very small, but the stretched graphon allows us to transform the sparse graphon into a denser variation with $\mathcal{L}^1$ norm equal to 1.

Recall the terns cut metric and cut distance defined in definitions D.2. Next, we define their stretched variations

**Definition O.2** (Stretched Cut-Distance). We define the stretched cut-metric by

$$d_{\square,s}(W_1, W_2) = d_\square(W_1^s, W_2^s),$$

and the stretched cut distance by

$$\delta_{\square,s}(W_1, W_2) = \delta_\square(W_1^s, W_2^s).$$

We will justify the use of these definitions with the following example that was given in (Jian et al., 2023; Ji et al., 2024)

*Example* O.3 ((Jian et al., 2023) Example 1). Consider a sequence of graphs $G_n = (V_n, E_n)$ with $|V_n| = n$, constructed as follows. Let $\alpha \in (0,1)$, we choose $\lfloor n^{\frac{1+\alpha}{2}} \rfloor$ vertices $V_n'$ to form a complete subgraph with the remaining vertices being isolated. We have $|E_n| = \frac{|V_n'| \cdot (|V_n'|-1)}{2}$, and therefore the edge density $d(G_n)$ is bounded by

$$d(G_n) = \frac{|E_n|}{n^2} \le \frac{|V_n'|^2}{2n^2},$$

which tends to 0 as $n$ tends to infinity because $\alpha \in (0,1)$. Hence, the induced graphon $W_{G_n}$ converges to the zero graphon in the cut distance.

let $m_n = \frac{|V_n'|}{\sqrt{|V_n'| \cdot (|V_n'|-1)}}$. The stretched graphon $W_{G_n}^s$ is 1 in $(0, m_n)^2 \setminus D_n$ where $D_n$ consists of $|V_n'|$ diagonal squares, each with length $\frac{m_n}{|V_n'|}$, and 0 elsewhere. As $n \to \infty$ we have $m_n \to 1$ and $\lambda(D_n) \to 0$. Therefore, $W_{G_n}^s$ converges to $I_{[0,1]^2}$ in cut distance. We summarize with that while the sequence of graphons $W_{G_n}$ converge to 0 in the cut distance, it converge to another, non-trivial graphon in the stretched cut distance.

### O.2. $\mathcal{L}^p$ Graphons

Here, we present the basic extension of graphon theory (as presented in section D.1) to the theory of $\mathcal{L}^p$ graphons that was introduced in (Borgs et al., 2019; 2018b). In this context, a graphon will be a symmetric measurable function $W : [0,1]^2 \to \mathbb{R}$.

**Definition O.4.** A graphon $W$ is called an $\mathcal{L}^p$ graphon, for $1 \le p < \infty$, if

$$\|W\|_p^p = \int_{[0,1]^2} |W(x,y)|^p \, dx \, dy < \infty.$$

Note that an $\mathcal{L}^q$ graphon is also an $\mathcal{L}^p$ graphon for $1 \leq p \leq q \leq \infty$.

(Borgs et al., 2019) presents a generalization of Szemerédi's regularity lemma and proves a weak regularity lemmas for $\mathcal{L}^p$ graphons. Equipped with this lemma, the space of uniformly bounded $\mathcal{L}^p$ graphons, endowed with the cut distance is proved to be compact in Theorem 2.13 in (Borgs et al., 2019).

Consider the following model of graphs $G_n$ that demonstrates the limitations of traditional $\mathcal{L}^\infty$ graphons.

*Example* O.5. Label the vertices of $G_n$ by $V_n = [n]$. For two vertices $i, j \in V_n$, the probability of the edge $(i, j)$ to appear in the graph is $\min\{1, n^\beta (ij)^{-\alpha}\}$, where $0 < \alpha < 1$ and $0 < \beta < 2\alpha$ [3]. This model is a straightforward way to generate a power law degree distribution, where the expected degree of vertex $i$ decreases as $i^{-\alpha}$ [4]. The expected number of the entire edges in the graph is on order of $n^{\beta - 2\alpha + 2}$ which is superlinear when $\beta > 2\alpha - 1$. However, even rescaling by the edge density $n^{\beta - 2\alpha}$ does not yield an $\mathcal{L}^\infty$ graphon. Instead, we obtain the graphon $W(x, y) = (xy)^{-\alpha}$ which is unbounded.

## O.3. Graphings

In this part we introduce infinite graphs that generalize finite bounded degree graphs as described in (Lovász, 2012). Their main role is to act as limit objects for sequences of graphs with bounded degrees, similar to how graphons serve in the context of dense graphs.

We begin with a more general term. Let $(\Omega, \mathcal{B})$ be a Borel $\sigma$-algebra. A graph $\mathbf{G}$, with $V(\mathbf{G}) = \Omega$, is said to be a Borel graph if its edge set is a Borel set in $\mathcal{B} \times \mathcal{B}$. We consider only graphs with all degrees bounded by $D > 0$. Lemma 18.2 in (Lovász, 2012) is a useful result that motivate the definition that follows.

**Lemma O.6.** *A graph $\mathbf{G}$ on a Borel space $(\Omega, \mathcal{B})$ is a Borel graph if and only if for every Borel set $B \in \mathcal{B}$, the neighborhood $N_{\mathbf{G}}(B)$ is Borel.*

Now, we are ready to introduce the notion of graphing (or measure preserving graphs). We first endow the measure space $(\Omega, \mathcal{B})$ with a probability measure $\mu$.

**Definition O.7.** We say that a graph $\mathbf{G}$, is a graphing, if it is a Borel graph, and for every measurable sets $A, B$, we have

$$\int_A deg_B(x) \, d\mu(x) = \int_B deg_A(x) \, d\mu(x). \tag{29}$$

In other words, the number of edges connecting $A$ to $B$ is the same as the number of edges connecting $B$ to $A$. To clarify, a graphing is a quadruple $\mathbf{G} = (\Omega, \mathcal{B}, \mu, E)$, where $\Omega = V(\mathbf{G})$, $\mathcal{B}$ is a Borel $\sigma$-algebra on $\Omega$, $\mu$ is a probability measure and $E \in \mathcal{B} \times \mathcal{B}$ is the edge set satisfying the measure preserving condition 29.

The name "graphing" was introduced in (Adams, 1990) and it refers to the representation of the classes of an equivalence relation as the connected components of a Borel graph. However, it appears that the usage of the term has evolved to cover a broader definition, similar to that of graphons, which are used for dense graphs.

*Example* O.8 ((Lovász, 2012), Example 18.15). If $D = 1$ then every graphing $\mathbf{G}$ is the graph of an involution $\phi : S \to S$ for some set $S \subset V(\mathbf{G})$ (an involution is a map that, when applied twice, returns to the original element, meaning it is its own inverse). Since $S$ cannot include isolated vertices, we have $S = N_{\mathbf{G}}(V(\mathbf{G}))$, and it is measurable from lemma O.6. Furthermore, for every measurable set $A \subset S$ we have

$$\mu(\phi^{-1}(A)) = \int_{\phi^{-1}(A)} deg_A(x) \, d\mu(x) = \int_A deg_{\phi^{-1}(A)}(x) \, d\mu(x) = \mu(A),$$

thus, $\phi$ is a measure preserving map.

---

[3]The inequalities $\alpha < 1$ and $\beta < 2\alpha$ can be understood as follows: the first inequality prevents the majority of the edges from being concentrated among a sublinear number of vertices, while the second ensures that the cut-off, which involves taking the minimum with 1, only impacts a negligible fraction of the edges.

[4]For further reading see (Barabási & Albert, 1999).

