# OpenReview forum: "A Graphop Analysis of Graph Neural Networks on Sparse Graphs: Generalization and Universal Approximation"
_ICML.cc/2026/Conference — ICML 2026 regular_

### Official Review · Reviewer_Lytk · 2026-03-11

**Soundness:** 4
**Presentation:** 4
**Significance:** 4
**Originality:** 4
**Overall Recommendation:** 6
**Confidence:** 2

**Summary:**

This work uses recent mathematical results to study universal approximation theorems and generalization results of message passing neural networks, which extends previous literature, which worked only on dense graphs.  The basic intuition is the following: instead of using graphons (functional descriptions of graphs as two variable functions obtained as limit objects of families of graphs) they use graphops (operators describing the action of graphs on functions).

I found very interesting the definition of bofop-signal, which puts together a bofop (bounded fiber operator) with a signal, where the signal is a function used to represent in this framework node attributes. In this way, they can study MPNN with node features. This had repercussions in other theoretical tools they generalized (e.g. Definition 3.4 of k-profile).

The paper is clear and well written and rigorous. Despite the mathematical complexity, I believe the appendices make it self sufficient information in the appendices to understand the main text.

This paper is out of my area of expertise, but I can understand the relevance of having a general approximation theorem for MPNN.

**Compliance With Llm Reviewing Policy:**

Affirmed.

**Key Questions For Authors:**

- Would it be possible to extend this work to NN on tensors or hyper-graphs?
- Can you please add references of textbook/papers when you are citing well-known definitions (i.e. in the "background" appendices)?
- A question that is perhaps obvious for a practitioner in your field, but not for a reader from other fields. Why one would want to base a universal approximation theorem on the Stone–Weierstrass theorem? What are the advantages?

**Limitations:**

yes

**Strengths And Weaknesses:**

It would be probably helpful to move some of the definitions in the main text to the appendices so to better highlight all the technical tools you need to develop to reach to the main result. For example, by moving the results section before appendix F into the main text, and perhaps adding an "informal statement" of what I understand to be your main result (Theorem M.4.).

---

> ### Author Rebuttal · Authors · 2026-03-29
>
> We thank the reviewer for the interest in our work.
> Below, we respond point by point to the reviewer's comments.
>
>
> > 1. It would be probably helpful to move some of the definitions in the main text to the appendices so to better highlight all the technical tools you need to develop to reach to the main result. For example, by moving the results section before appendix F into the main text, and perhaps adding an "informal statement" of what I understand to be your main result (Theorem M.4.).
>
> **Response.** Thank you for this comment. Since the theoretical construction in our paper is quite vast, it is indeed very challenging to condense the construction to the short ICML format. For such a paper, the appendix contains all of the details, and the main paper is a summary, often informal, of the appendix, focusing on the big picture and implications to the field.
> If accepted, we will make an effort to summarize more of the results from the appendix in the main paper.
>
> > 2. Would it be possible to extend this work to NN on tensors or hyper-graphs?
>
> **Response.**  An extension to hypergraphs may be possible by defining operators over tensor products of signals. This solution is seemed to be hinted within the question of the reviewer. We leave such extensions to future work.
>
> > 3. Can you please add references of textbook/papers when you are citing well-known definitions (i.e. in the "background" appendices)?
>
> **Response.** Thank you for the suggestion. We will add such references in the camera-ready version if accepted.
>
> >4. A question that is perhaps obvious for a practitioner in your field, but not for a reader from other fields. Why one would want to base a universal approximation theorem on the Stone–Weierstrass theorem? What are the advantages?
>
> **Response.**  The Stone–Weierstrass theorem is a general formulation of universal approximation. This theorem is formulated over general metric spaces, where the hypothesis class is a general algebra of functions. To derive specific universal approximation theorems, one should specify the metric space (which would be the space of data) and the algebra (which would be the neural networks), showing that they satisfy all of the assumptions in the Stone–Weierstrass theorem.
> We will add such a discussion to the paper if accepted.

---

> > ### Author Rebuttal · Reviewer_Lytk · 2026-04-04
> >
> > Thanks for the answers.
> >
> > I understand your answer on UAT. My question was on why specifically Stone-Weierstrass and not other approaches. What advantages you have with SW?

---

> > > ### Author Response · Authors · 2026-04-05
> > >
> > > Thank you for the clarification.
> > >
> > > Let us elaborate. Our reason for using the Stone–Weierstrass (SW) theorem is that its generality makes it well suited for our setting, where the input space is not a Euclidean space, but rather a “non-standard” compact metric space that we introduced in the paper. Other universal approximation approaches we know of are restricted to inputs from a Euclidean space. Moreover, SW reduces universal approximation to establishing continuity and separation power. The latter is a central question for GNNs on its own (called expressivity). Hence, SW is appealing here not only because it yields a universal approximation theorem, but also because its assumptions align directly with the key properties of interest to the GNN community.
> > >
> > > However, if you have a suggestion for another approach for universal approximation that is appropriate for our setting we will be more than happy to hear.
> > >
> > > Again, we appreciate your responses and kind suggestions.

---

### Official Review · Reviewer_vTpy · 2026-03-11

**Soundness:** 3
**Presentation:** 4
**Significance:** 3
**Originality:** 3
**Overall Recommendation:** 4
**Confidence:** 4

**Summary:**

This paper puts forward graphop analysis as a tool to analyze the limit properties of graph neural networks, with the objective of overcoming limitations of graphon-based analysis, namely the ability to analyze convergence to sparse limit objects. It establishes a framework where graphops are equipped with the typical notions of graph signal processing (adjacency, signals supported on graphops), and provide results on universal approximation, expressivity (extension of WL-test), and the existence of a convergence  bounds of MPNNs to bofop-DIDMs.

**Compliance With Llm Reviewing Policy:**

Affirmed.

**Final Justification:**

This work leverages graphop analysis as a theoretical tool to derive generalization and universal approximation results on Graph Neural Networks. While the results are powerful in the sense that they are general to sparse graphs of any size, the practical impact is somewhat limited, given that the results only show the existence of a generalization bound.

That being said, the work constitutes a meaningful contribution to the theoretical understanding of GNNs. The authors resolved most of my concerns during the discussion period. I recommend its acceptance.

**Key Questions For Authors:**

N/A, see weaknesses.

**Limitations:**

yes.

**Strengths And Weaknesses:**

### Strengths
1. The construction is indeed well-motivated, and the effort to unify the analysis on dense and sparse graphs is a valuable goal.
2. The exposition is clear and correct, as far as I could tell.
3. The theoretical results (universal approximation, expressivity, existence of a convergence bound) constitute a meaningful contribution.

### Weaknesses
1. Understanding that this is foundational work, it was not clear from the reading whether the value of graphop analysis comes only from unifying dense and sparse graph limit analysis, or whether it may enable a path for improved bounds over frameworks that treat sparse limit objects. For instance, it would have been nice to see explicit convergence rates of MPNNs to bofop-DIDMs. Given that the paper is centered around the premise that graphops enable analysis on sparse limit objects, it would have been interesting to see a contrast with bounds on previous works.
2. The abstract claims "(...) this leads to more powerful universal approximation theorems and generalization bounds than previous works", however, the authors only provide the *existence* of a generalization bound.
3. In the introduction, the paragraph labeled **Limitations in previous works**  should have citations. Particularly, the work does not mention the existing work [1] on graphop approximation and size transferability.
4. The related work sections would also be strengthened by including the existing literature on GNN limit analysis on graphons and manifolds, see below.
5. As admitted by the authors, the MPNN model requires additional structure for concrete applications.

While these weaknesses should be addressed, especially 1-3, I believe this is solid work that should be accepted.

###  References

**Graphop**

[1]  Le, Thien, and Stefanie Jegelka. "Limits, approximation and size transferability for GNNs on sparse graphs via graphops." _Advances in Neural Information Processing Systems_ 36 (2023): 41305-41342.


**Graphon neural networks**

[2]  Ruiz, Luana, Luiz Chamon, and Alejandro Ribeiro. "Graphon neural networks and the transferability of graph neural networks." _Advances in Neural Information Processing Systems_ 33 (2020): 1702-1712.

[3]  Ruiz, Luana, Luiz FO Chamon, and Alejandro Ribeiro. "Transferability properties of graph neural networks." _IEEE Transactions on Signal Processing_ 71 (2023): 3474-3489.

[4]  Cordonnier, Matthieu, et al. "Convergence of message-passing graph neural networks with generic aggregation on large random graphs." _Journal of Machine Learning Research_ 25.406 (2024): 1-49.

[5]  Herbst, Daniel, and Stefanie Jegelka. "Higher-order graphon neural networks: Approximation and cut distance." _arXiv preprint arXiv:2503.14338_ (2025).


**Manifold neural networks**

[6]  Levie, Ron, et al. "Transferability of spectral graph convolutional neural networks." _Journal of Machine Learning Research_ 22.272 (2021): 1-59.

[7]  Wang, Zhiyang, Luana Ruiz, and Alejandro Ribeiro. "Geometric graph filters and neural networks: Limit properties and discriminability trade-offs." _IEEE Transactions on Signal Processing_ 72 (2024): 2244-2259.

---

> ### Author Rebuttal · Authors · 2026-03-29
>
> We thank the reviewer for the comments and helpful suggestions.
> Below, we respond point by point to the reviewer's comments.
>
>
> > 1. Understanding that this is foundational work, it was not clear from the reading whether the value of graphop analysis comes only from unifying dense and sparse graph limit analysis, or whether it may enable a path for improved bounds over frameworks that treat sparse limit objects. For instance, it would have been nice to see explicit convergence rates of MPNNs to bofop-DIDMs. Given that the paper is centered around the premise that graphops enable analysis on sparse limit objects, it would have been interesting to see a contrast with bounds on previous works.
>
> **Response.** To the best of our knowledge, there are no previous works that proposed generalization bounds for general distributions of sparse graphs of arbitrary sizes. We are the first. Hence, our goal is not to improve any existing bound (as there aren't any), but just to prove such a bound exists (which vanishes when the size of the training set increases).
> Regarding explicit bounds, this is a very challenging question. To achieve such a thing in our analysis, we would need to develop explicit covering number bounds for the space of bofop-signals. While we proved in our paper that such a covering number is finite, obtaining explicit bounds is an open question also in the general graphop theory. Anyone who manages to do so will make a strong contribution to graph limit theory.
> If accepted, we would emphasize this point in the summary of the paper.
>
>
> > 2. The abstract claims "(...) this leads to more powerful universal approximation theorems and generalization bounds than previous works", however, the authors only provide the existence of a generalization bound.
>
> **Response.** You are right that the bound is not powerful in the sense that it is explicit and tight. What we meant is that the generalization theory is powerful in the sense that it accounts for very general graph distributions, more than any previous work.
> We will clarify this in the revised paper if accepted.
>
> > 3. In the introduction, the paragraph labeled Limitations in previous works should have citations. Particularly, the work does not mention the existing work [1] on graphop approximation and size transferability.
>
> **Response.** We will add citations to this section, and specifically mention [1] in our revised manuscript if accepted. The paper by [1] studies transferability, namely whether GNN outputs remain consistent across finite graphs of different sizes that arise from the same underlying graphop, which serves as a common generative model. Our focus is different. We focus on generalization, which refers to the ability of a GNN to perform well beyond the data on which it was trained. In this sense, generalization addresses a broader question than transferability. Note that the transferability analysis [1] is based on a very restrictive model of the data (all graphs are close to the same graphop), while in our analysis there is no assumption whatsoever on the data distribution. Accordingly, transferability analyses are much easier than generalization analyses.
>
> *References*
> [1] Le, Thien, and Stefanie Jegelka. "Limits, approximation and size transferability for GNNs on sparse graphs via graphops." Advances in Neural Information Processing Systems 36 (2023): 41305-41342.
>
>
> > 4. The related work sections would also be strengthened by including the existing literature on GNN limit analysis on graphons and manifolds, see below.
>
> **Response.** Thank you for your suggestions. We will include these citations and a brief discussion in the revised version if accepted.

---

> > ### Author Rebuttal · Reviewer_vTpy · 2026-04-03
> >
> > I thank the authors for their response and revisions. There was no response to (5), but I understand it to be beyond the scope of this work and difficult to discuss in the context of rebuttals. I will maintain my positive score.

---

> > > ### Author Response · Authors · 2026-04-04
> > >
> > > Thank you again for taking the time to thoroughly assess our paper.
> > >
> > > Note that in point 5 you repeated a weakness that we reported ourselves in the paper, where we also wrote a possible solution. Hence, we assumed you did not expect a response, as we supposed that you already knew our solution to the problem. This, especially as the instructions say that not every point has to be responded to.
> > >
> > > However, as this was a misunderstanding, we can offer a response now.
> > >
> > > In the paper we wrote:
> > >
> > > "**Limitations.** Our construction of MPNNs on profiles is purely theoretical. Since profiles are defined as *sets* of measures, with no additional structure, they are not directly amenable to computational mathematics. One possible future avenue is to endow additional structure on profiles, like a probability measure, that would allow using them as a data structure for computations."
> > >
> > > Let us elaborate on this. In computational mathematics, when using elements that come from an uncountable infinite set, one must endow the set with additional structure. For example, when doing numerical analysis with real numbers, since the set of all real numbers is uncountably infinite, almost none of the real numbers can be represented on a computer. To still be able to do computations, we need a *metric* on the set of numbers, i.e., the absolute difference $|x-y|$. Now that we have this additional structure, we can discuss sets of numbers that are close enough to all real numbers in some large interval of interest -- the numbers in floating point representation. With the metric, we can also discuss approximations of operations like differentiation and integration. Without a metric there would be no notion of approximation, and any finite set of numbers would just be a small fraction of all numbers.
> > >
> > > Now, in our case, a profile is a set of distributions. While we have a metric between profiles -- the Hausdorff metric -- it is very challenging and nonstandard to do computational mathematics with the Hausdorff metric on all profiles. To construct something like a floating point representation for profiles we need two levels of approximation - finding a finite set of measures close to all measures in Wasserstein distance, and then considering all subsets of these measures as the approximation of profiles. This is simply not feasible.
> > >
> > > However, another approach for converting a set into something computational is endowing it with a probability measure. This opens the door for Monte Carlo analysis, which does not require any metric, and does not suffer from the curse of dimensionally like metric methods. This is the viable path forward that we propose to develop in future work.
> > >
> > > **Regardless, in this paper we only need the construction of profiles as a theoretical tool to be used as an intermediate step towards understanding DIDMs better. The computational aspects of profiles do not affect our results whatsoever.**
> > >
> > > We would also like to stress the following point. You wrote “the MPNN model requires additional structure for concrete applications.” This is not accurate. The MPNN itself is the standard MPNN on graphs, and is fully computational. What we stated in the paper is that *profiles* are not currently computational entities. This comment is meant to address potential future work, which may want to adopt profiles as a computational structure and base novel graph machine learning models on them.
> > > Accordingly, the fact that profiles are not computational data structures has no effect on the present work.
> > >
> > >
> > > We will extend our discussion on this topic in the appendix if accepted.
> > >
> > > We hope this response is satisfactory, and would ask the reviewer to consider raising their score if so.

---

### Official Review · Reviewer_w1Qb · 2026-03-12

**Soundness:** 1
**Presentation:** 2
**Significance:** 3
**Originality:** 3
**Overall Recommendation:** 5
**Confidence:** 3

**Summary:**

This paper uses the recently developed theory for sparse graph limits called graphops to develop strong theoretical results (a novel universal function approximation theorem and generalization bounds) for MPNNs. They extend the line of work on graphons and GNNs and through the graphops theory bypass the main weakness of the original graphon based results (uniform convergence only worked for dense graphs or for graphs of bounded size, while here it works for sparse too and for graphs of unbounded size).

**Compliance With Llm Reviewing Policy:**

Affirmed.

**Final Justification:**

Post rebuttal: updated score. The authors provided the missing theorem and proof.

**Key Questions For Authors:**

See above.

**Limitations:**

yes

**Strengths And Weaknesses:**

This is an important and strong paper. It brings the theory of sparse graph limits (specifically graphops) to mpnns/expressivity and expands on the important line of work of mpnns and graphons. This new perspective allows to develop better universal function approximation (and generalization) results: uniform convergence results for graphs of unbounded size and handling sparse graphs. The previous graphon based results only allowed dense graphs.

However, the paper seems rather rushed. In particular, one of the two main theorems  (the universal function approximation result for bofops) and **its proof is simply missing**. Both are postponed until Appendix M and even announced as "In this section we prove our main results of the paper", but then the section just ends without the formal statement of the theorem and without a proof. This is why I currently indicated soundness as poor even though I do believe the statement is correct.

Further comments:
* There is another work on GNNs and graphops by Le & Jegelka [NeurIPS 2023]. Please discuss how your results differ from this paper. Otherwise it might seem like this submission is the first using graphop theory in the context of GNN.


Smaller typos:
* "inour"
* "above two theorem"
* "For more details on covering number generalization bounds see Appendix C.4". Appendix C.4 just contains two very vague sentences about generalization in general not about covering numbers.
* "sprase" -> "sparse"
* " A prior" -> "priori"

---

> ### Author Rebuttal · Authors · 2026-03-29
>
> We thank the reviewer for the detailed comments and the careful attention to details.
> Below, we respond point by point to the reviewer's comments.
>
> > 1. However, the paper seems rather rushed. In particular, one of the two main theorems (the universal function approximation result for bofops) and its proof is simply missing.
>
> **Response.** We apologize for the omission, which occurred accidentally when we moved the results from our draft version to the conference format.
> The proof of the universal approximation theorem is quite short, and we present it below, as all of the challenging parts of the proof were already proven in the appendices before Appendix M. We will include the missing theorem statement and the proof in the final version.
>
> Before the proof, we introduce some notation for the convenience of the reviewer. Recall that the space $\Gamma_L(\mathcal{BF}^r_d)$ is the space of all bofop-DIDMs of order $L$. Let $\mathcal{NN}_L^{p}(\mathcal{BF}^r_d)\subset C(\mathcal{BF}^r_d,\mathbb{R}^p)$ be the space of all $L$-layer MPNN models with readout, where $C(\mathcal{BF}^r_d,\mathbb{R}^p)$ is the space of all continuous functions $\mathcal{BF}^r_d\to\mathbb{R}^p$. Also, recall from Appendix E.8 the set $\mathcal{N}^{d_L}_L$ of $L$-layer MPNN models from the space of IDMs $\mathcal{H}^L$, to $\mathbb{R}^{d_L}$, and the set $\mathcal{N}\mathcal{N}^p_L$ of all $L$-layer MPNN models with readout from the space of DIDMs $\mathcal{P}(\mathcal{H}^L)$, to $\mathbb{R}^p$.
>
> *Theorem.* Let $L\in\mathbb{N}_0$. Then, the set $\mathcal{NN}_L^1(\mathcal{BF}^r_d)$ is uniformly dense in $C(\mathcal{BF}^r_d,\mathbb{R})$.
>
> The proof relies on the Stone-Weierstrass Theorem. To apply it, we use Lemma 78 from [2].
>
> *Lemma 1.*
> Let $0 \leq t < \infty$. The set
> $\mathcal{N}^1_t$
> is closed under multiplication and linear combinations, contains the constant function $\mathbb{1}_{\mathcal{H}^t}$, and separates points of $\mathcal{H}^t$.
>
> Lemma 1 can also be formulated for the set $\mathcal{NN}_t^1$ as mentioned in the proof of Theorem 15 in [2].
>
> *Proof.*
>
> By Corollary 5.3 the space
>  $(\Gamma_L(BF_d^r),OT_{d^L_{IDM}})$
> is compact.  Since we can identify the space $(BF_d^r,\delta_{DIDM}^L)$ with $(\Gamma_L(BF_d^r),OT_{d^L_{IDM}})$ (by Definition K.5), we also have compactness of $(\mathcal{BF}^r_d,\delta_{\text{DIDM}}^L)$. Since $\mathcal{N}\mathcal{N}_L^1(\Gamma_L(\mathcal{BF}_d^r))$ is a subalgebra of $C(\mathcal{P}(\mathcal{H}^L),\mathbb{R})$, by Lemma 1, $\mathcal{NN}_L^1(\mathcal{BF}_d^r)$ satisfies the conditions of the Stone-Weierstrass Theorem (Theorem C.4). $\quad\square$
>
>
>
> > 2. There is another work on GNNs and graphops by Le & Jegelka [NeurIPS 2023]. Please discuss how your results differ from this paper. Otherwise it might seem like this submission is the first using graphop theory in the context of GNN.
>
> **Response.** We will mention this paper in our revised manuscript if accepted. The paper by [1] studies transferability, namely whether GNN outputs remain consistent across finite graphs of different sizes that arise from the same underlying graphop, which serves as a common generative model. Our focus is different. We focus on generalization, which refers to the ability of a GNN to perform well beyond the data on which it was trained. In this sense, generalization addresses a broader question than transferability. Note that the transferability analysis [1] is based on a very restrictive model of the data (all graphs are close to the same graphop), while in our analysis there is no assumption whatsoever on the data distribution. Accordingly, transferability analyses are much easier than generalization analyses.
>
> *References*
> [1] Le, Thien, and Stefanie Jegelka. "Limits, approximation and size transferability for GNNs on sparse graphs via graphops." Advances in Neural Information Processing Systems 36, 2023.
>
> [2] Rauchwerger, L., Jegelka, S., and Levie, R. "Generalization, expressivity, and universality of graph neural networks on attributed graphs." International Conference on Learning Representations, 2025.
>
> > 3. Smaller typos:
>
> **Response.** Thank you, we will correct these typos.

---

> > ### Author Rebuttal · Reviewer_w1Qb · 2026-04-01
> >
> > Thanks.

---

> > > ### Author Response · Authors · 2026-04-05
> > >
> > > Thank you very much for your thoughtful review and for reconsidering your rating. We appreciate your careful reading, constructive engagement, and the time you devoted to our work.

---

### Official Review · Reviewer_6WbH · 2026-03-15

**Soundness:** 3
**Presentation:** 3
**Significance:** 2
**Originality:** 3
**Overall Recommendation:** 4
**Confidence:** 4

**Summary:**

This work presents a unified mathematical framework for analyzing Message Passing Neural Networks (MPNNs) on both sparse and dense graphs. It extends the graphop analysis of Backhausz and Szegedy (2022). The main contributions are (1) incorporating node features into the framework, leading to graphop-signal analysis; (2) focusing on a subclass of graphop-signals and establishing that MPNNs are Lipschitz continuous with respect to the action metric; and (3) connecting graphop theory with the theory of distributions of iterated degree measures (DIDMs), showing that MPNNs are uniformly equicontinuous under the DIDM-mover’s distance. The construction of MPNNs on profiles in this work is purely theoretical.

**Compliance With Llm Reviewing Policy:**

Affirmed.

**Key Questions For Authors:**

I like the mathematics, but it would be helpful to illustrate it with a numerical example, if possible. Thanks.

**Limitations:**

It would be nice to include a numerical examples.

**Strengths And Weaknesses:**

I like the authors’ mathematics, but it would be helpful to illustrate their extensions with a numerical example. The practical limitations are not clear to me.

---

> ### Author Rebuttal · Authors · 2026-03-29
>
> We thank the reviewer for the positive assessment of the mathematics and for the suggestions.
> Below, we respond point by point to the reviewer's comments.
>
>
> > 1. I like the authors’ mathematics, but it would be helpful to illustrate their extensions with a numerical example.
>
> **Response.** As explained in the paper, profiles are not computational structures, so we cannot perform experiments on them (see also the response to point 2). In practice, experiments can only be conducted on DIDMs, or on synthetic and standard graph benchmarks, using evaluation metrics such as accuracy or loss. Such experiments were already conducted in two previous papers by [1] and [2]. In those papers, while the theory was developed for dense graphs, the experiments were carried out on standard graph benchmarks, which are sparse. Hence, those experiments were not fully appropriate for [1] and [2], but are in fact appropriate for our paper. Since the same experimental pipeline transfers directly to the present setting, adding such experiments here would duplicate prior empirical evidence. For this reason, we chose to focus on the theoretical contribution.
> However, if the reviewer insists, we can duplicate the experiments of [1,2] in our paper. We of course expect to get the same results. Alternatively, if the reviewer agrees, we can cite those experiments in our paper, explaining their outcomes and giving full credit to [1,2].
>
> *References*
>
> [1] Böker, J., Levie, R., Huang, N., Villar, S., and Morris, C. Fine-grained expressivity of graph neural networks. Advances in Neural Information Processing Systems, 36, 2024.
>
> [2] Rauchwerger, L., Jegelka, S., and Levie, R. Generalization, expressivity, and universality of graph neural networks on attributed graphs. International Conference on Learning Representations, 2025.
>
>
>
>
>
> > 2. The practical limitations are not clear to me.
>
> **Response.** In the paper we wrote:
> *"Our construction of MPNNs on profiles is purely theoretical. Since profiles are defined as *sets* of measures, with no additional structure, they are not directly amenable to computational mathematics."*
>
> Let us elaborate on this. In computational mathematics, when using elements that come from an uncountable ifinite set, one must endow the set with additional structure. For example, when doing numerical analysis with real numbers, since the set of all real numbers is uncountably infinite, almost none of the real numbers can be represented on a computer. To still be able to do computations, we need a *metric* on the set of numbers, i.e., the absolute difference $|x-y|$. Now that we have this additional structure, we can discuss sets of numbers that are close enough to all real numbers in some large interval of interest -- the numbers in floating point representation. With the metric, we can also discuss approximations of operations like differentiation and integration. Without a metric there would be no notion of approximation, and any finite set of numbers would just be a small fraction of all numbers.
> Now, in our case, a profile is a set of distributions. While we have a metric between profiles -- the Hausdorff metric -- it is very challenging and nonstandard to do computational mathematics with the Hausdorff metric on all profiles. To construct something like a floating point representation for profiles we need two levels of approximation - finding a finite set of measures close to all measures in Wasserstein distance, and then considering all subsets of these measures as the approximation of profiles. This is simply not feasible.
> Hence, we only treat the construction of profiles as a theoretical tool and an intermediate step towards understanding DIDMs better.
>
> We will add an extended discussion on this computational challenge in the appendix.

---

### Decision · Program_Chairs · 2026-04-30

**Decision:**

Accept (regular)

**Comment:**

The paper relies on graphop analysis to present a unified framework to study generalization and universal approximation of message-passing graph neural networks, which is applicable to both dense and sparse graphs. The reviewers were unanimously positive about this paper saying the theoretical contribution is novel and significant. Some issues on presentation were raised, but they seem to have been addressed in the rebuttal.